# Hypothesis Testing for Differentially Private Linear Regression

Daniel Alabi [*1] and Salil Vadhan [†2]

[1]Department of Computer Science and Data Science Institute, Columbia University
[2]Harvard School of Engineering and Applied Sciences

## Abstract

In this work, we design differentially private hypothesis tests for the following problems in the general linear model: testing a linear relationship and testing for the presence of mixtures. The majority of our hypothesis tests are based on differentially private versions of the $F$-statistic for the general linear model framework, which are uniformly most powerful unbiased in the non-private setting. We also present another test for testing mixtures, based on the differentially private nonparametric tests of Couch, Kazan, Shi, Bray, and Groce (CCS 2019), which is especially suited for the small dataset regime. We show that the differentially private $F$-statistic converges to the asymptotic distribution of its non-private counterpart. As a corollary, the statistical power of the differentially private $F$-statistic converges to the statistical power of the non-private $F$-statistic. Through a suite of Monte Carlo based experiments, we show that our tests achieve desired *significance levels* and have a high *power* that approaches the power of the non-private tests as we increase sample sizes or the privacy-loss parameter. We also show when our tests outperform existing methods in the literature.

## 1 Introduction

Linear regression is one of the most fundamental statistical techniques available to social scientists and economists (especially econometricians) [43, 20]. A graduate student successfully completes her classes: what is the effect on her future earnings? A country implements tough penalties for concealed carry permit violations: what is the effect on gun-related deaths? A school district cuts the size of the average class size for high school classes: what is the effect on the standardized test scores of the minority students?

All three questions are about the (unknown) effect of changing one (independent) variable $X$ on another variable $Y$. Under reasonable assumptions, the relationship between $X$ and $Y$ can be modeled as a linear regression testing problem. One of the goals of performing regression analysis is for use in decision-making via point estimation (i.e., getting a single predicted value for the dependent variable). To increase the confidence of decision-makers and analysts in such estimates, it is often important to also release accompanying uncertainty estimates for the point estimates [18, 19, 6].

In this work, we aim to provide differentially private linear regression *uncertainty quantification* via the use of hypothesis tests. Given the realistic possibility of reconstruction, membership, and inference attacks [46, 25], we can rely on Differential Privacy (DP), a rigorous approach to quantifying

---

[*]Email: `alabid@cs.columbia.edu`. D.A. was supported by a Fellowship from Meta AI and Cooperative Agreement CB20ADR0160001 with the Census Bureau. Work done while a Ph.D. student at Harvard University.

[†]Email: `salil_vadhan@harvard.edu`. S.V. was supported by Cooperative Agreement CB20ADR0160001 and a Simons Investigator Award.

privacy loss [24, 22]. The task of DP linear regression is to, given datapoints $\{(x_i, y_i)\}_{i=1}^n$, estimate point or uncertainty estimates for linear regression while satisfying differential privacy. The majority of our tests will rely on generalized likelihood ratio test $F$-statistics.

In previous works [41, 49, 13, 3], differentially private estimators for linear regression are explored and key factors (such as sample size and variance of the independent variable) that affect the accuracy of these estimators are identified. The focus of these previous works is for point estimate prediction. The predictive accuracy of such estimators can be measured in terms of a confidence bound or mean-squared error. We continue the study of the utility of such estimators for use in uncertainty quantification via hypothesis testing [41]. (See Section B for a more detailed background on hypothesis testing.)

Earlier work on uncertainty quantification for linear regression was done by Sheffet [41], who constructed confidence intervals and hypothesis tests based on the $t$-test statistic, and can be used to test a linear relationship. The random projection routine in [41], based on the Johnson–Lindenstrauss (JL) transform, only starts to correctly reject the null hypothesis when the sample size is very large (or the variables have a large spread). This observation is also supported by the work of [21]. Furthermore, the random projection routine requires extra parameters (e.g., for specifying the dimensions of the random matrix). In our work, we use the $F$-statistic and our framework can be used to test mixture models, amongst other tests. We provide a general framework for DP tests based on the $F$-statistic. In addition, we will consider hypothesis testing for linear regression coefficients on both small and large datasets. For the mixture model tester, we additionally adapt and evaluate a nonparametric method, a DP analogue of the Kruskal-Wallis test due to Couch, Kazan, Shi, Bray, and Groce [21], which is especially suited for the small dataset regime (See Section C.3.2 for full design, implementation, and proof details for the Kruskal-Wallis test.). To the best of our knowledge, our tests are the first to differentially privately detect mixtures in linear regression models, with accompanying experimental validation.

## 1.1 Our Contributions

In this work, we show that for the problem of differentially private linear regression, we can perform hypothesis testing for two problems in the general linear model: (1) **Testing a Linear Relationship**: is the slope of the linear model equal to some constant (e.g., slope is 0)? (2) **Testing for Mixtures**: does the population consist of one or more sub-populations with different regression coefficients?

We provide a differentially private analogue of the $F$-statistic which we, under the general linear model, show converges in distribution to the asymptotic distribution of the $F$-statistic (Theorem 2.2). Furthermore, the DP regression coefficients converge, in probability, to the true coefficients (Lemma D.11). In particular, in Lemma D.11, we show a $1/\sqrt{n}$ statistical rate of convergence for the DP regression coefficients used for our hypothesis tests. This matches the optimal rate [38]. We then use our DP $F$-statistic and parametric estimates to obtain DP hypothesis tests using a Monte Carlo parametric bootstrap, following Gaboardi, Lim, Rogers, and Vadhan [30]. The Monte Carlo parametric bootstrap is used to ensure that our tests achieve a target significance level of $\alpha$ (i.e., data generated under the null hypothesis is rejected with probability $\alpha$). We experimentally compare these tests to their non-private counterparts for univariate linear regression (i.e., one independent variable and one dependent variable). To the best of our knowledge, our tests are the first that use the $F$-statistic to perform tests on the problem of linear regression while ensuring privacy of the data subjects. In addition, our $F$-statistic based tests can be adapted to work on design matrices in any dimension i.e., the design matrix can be cast in the form $X \in \mathbb{R}^{n \times p}$ for any integer $p \geq 2$ where $n$ represents the number of individuals in the dataset and $p$ represents the number of features per individual; we leave experimental evaluation of the multivariate case for future work.

Experimental evaluation of our hypothesis tests is done on: (1) **Synthetic Data**: We generate synthetic datasets with different distributions on the independent (or explanatory) variables. Specifically, we consider uniform, normal, and exponential distributions on the independent variables. We also vary the noise distribution of the dependent variable. (2) **Opportunity Insights (OI) Data**: We use a simulated version of the data used by the Opportunity Insights team (an economics research lab) to release the Opportunity Atlas tool, primarily used to predict social and economic mobility. We chose to use these datasets since they come from a real deployment of privacy-preserving statistics.

[3] The census tract-level datasets from these states can have a very small number of datapoints. (3) **Bike Sharing Dataset**: We use a real-world dataset publicly available in the UCI machine learning repository. The dataset consists of daily and hourly counts (with other information such as seasonal and weather information) of bike rentals in the Capital bikeshare system in years 2011 and 2012.

Our experimental findings are as follows:

1. **Significance and Power**: Across a variety of experimental settings, the significance is below the target significance level of 0.05. Thus, we have a high confidence that we will not falsely reject the null hypothesis, should the null hypothesis be true. Our tests are designed to be conservative in the sense that they err on the side of failing to reject the null (e.g., when the DP estimate of a variance is negative). Consistently, the power of our tests increase as we increase sample sizes (from hundreds up to tens of thousands) or as we relax the privacy parameter. The behavior of our DP tests tends toward that of the non-DP tests. The power of the DP OLS linear relationship tester increases as the slope of the model increases and as the noise in the dependent variable decreases. But when the DP estimate of the noise in the dependent variable is negative, the tests err on the side of failing to reject the null, leading to a lower power. The power of the mixture model tests also increases as the difference between the slopes in the two groups increases. And the more uneven the group sizes are, the lower the power.

2. **Alternative Methods**: (a) We compare our DP linear relationship tester, based on the $F$-statistic, to a test that builds on a DP parametric bootstrap method for confidence interval estimation. Ferrando, Wang, and Sheldon [29] prove that these intervals are consistent (in the asymptotic regime) and experimentally show that these intervals have good coverage. We can rely on such parametric bootstrap methods to decide to reject or fail to reject the null hypothesis. Such methods achieve the desired significance levels. However, we observe that the method is less powerful than the $F$-statistic approach. This behavior could be attributed to the differences in the bootstrap process: whereas we use estimates of sufficient statistics under the null in the bootstrap procedure, [29] uses the entirety of the sufficient statistics estimated for the parametric model. (b) The DP nonparametric method we provided, adapting the work of Couch, Kazan, Shi, Bray, and Groce [21], has a higher power in the small dataset regime than the DP $F$-statistic method. As the dataset size increases or the difference in slopes between the groups increases, the gap closes. In addition, as the variance of the independent variable increases, the $F$-statistic method outperforms the nonparametric method. This observation echos findings in previous work [3] for the case of prediction rather than hypothesis testing. In that work, it is shown that parametric methods for linear regression perform well in prediction as $n$, the variance of the independent variable, or the privacy loss parameter increase.

## 1.2 Overview of Techniques

We provide Algorithm 1, a generic framework for DP Monte Carlo tests via a parametric bootstrap routine for estimating sufficient statistics. Algorithm 1 crucially relies on DPStats, a procedure that uses statistics of the independent and dependent variables to produce DP statistics. These statistics can be used to decide to reject or fail to reject the null hypothesis. DPStats must satisfy DP—in our case $\rho$-zCDP (Zero-Concentrated Differential Privacy) [12]. Algorithms 2 and 5 are SSP (Sufficient Statistic Perturbation) implementations of DPStats for the linear relationship tester and mixture model tester, respectively. (See Section C.2 for more details.) Algorithm 6 is a DP nonparametric test framework based on the Kruskal-Wallis test statistic for testing mixtures. We use the standard Gaussian mechanism to make these algorithms satisfy zCDP although the Laplace mechanism could be used instead (i.e., for pure DP). To make Algorithm 6 DP, we randomly pair datapoints so that the resulting transformation is 1-Lipschitz (See Section C.3 for more details.). Theorem 2.2 shows that the DP $F$-statistic converges to the asymptotic distribution of its non-private counterpart, the chi-squared distribution. As a corollary, the statistical power of the DP $F$-statistic converges to the statistical power of the non-private $F$-statistic. To prove this result, we first show the $1/\sqrt{n}$ statistical rate of convergence for the DP coefficients. Then we reformulate both the $F$-statistic and the DP $F$-statistic in terms of the $1/\sqrt{n}$ convergent quantities, including convergent functions of the Gram

---

[3] See [19, 20] for a more detailed description of the use of the Opportunity Atlas tool in predicting social and economic mobility. [18, 3] evaluate privacy-protection methods on Opportunity Atlas data.

Matrix of the independent variable. Next, we construct random variables whose $\ell_2$-norm squared is distributed as the (non-central) chi-squared distribution. Then we prove that DP analogues of the squared $\ell_2$-norm of such random variables converge to the (non-central) chi-squared distribution. Finally, using such random variables, the reformulated DP $F$-statistic, reformulated $F$-statistic, and the continuous mapping theorem, we combine the convergent quantities to prove Theorem 2.2. See Section D for more details.

## 1.3  Other Related Work

**Differentially Private Linear Regression**    Sheffet [41] considers hypothesis testing for ordinary least squares for a specific test: testing for a linear relationship under the assumption that the independent variable is drawn from a normal distribution. Wang [49] focuses on using adaptive algorithms for linear regression prediction. $M$-estimators [34], motivated by the field of robust statistics [35], are a simple class of statistical estimators that present a general approach to parametric inference. Dwork and Lei [23, 38] and Chaudhuri and Hsu [17] present differentially private $M$-estimators with near-optimal statistical rates of convergence. Avella-Medina [7] generalizes the $M$-estimator approach to differentially private statistical inference using an empirical notion of influence functions to calibrate the Gaussian mechanism. Alabi, McMillan, Sarathy, Smith, and Vadhan [3] proposed median-based estimators for linear regression and evaluated their performance for prediction. All of these previous works show connections between robust statistics, $M$-estimators, and differential privacy. The ordinary least squares estimator is a classical $M$-estimator for prediction. Other examples include sample quantiles and the maximum likelihood estimation (MLE) objective. However, for differentially private hypothesis testing, as we show in this work, the optimal test statistic for DP linear regression depends on statistical properties of the dataset (such as variance and sample size). We also present novel differentially private test statistics that converge in distribution to the asymptotic distribution of the $F$-statistic. Bernstein and Sheldon [10] take a Bayesian approach to linear regression prediction and credible interval estimation. Through the Bayesian lens, there is also work on how to approximately bias-correct some DP estimators while providing some uncertainty estimates in terms of (private) standard errors [27]. As a motivation for differentially private simple linear regression point and uncertainty estimation, Bleninger, Dreschsler, Ronning [11] show how an attacker could use background information to reveal sensitive attributes about data subjects used in a simple linear regression analysis.

**General Differentially Private Hypothesis Testing**    Gaboardi, Lim, Rogers, and Vadhan [30] study hypothesis testing subject to differential privacy constraints. The tests they consider are: (1) *goodness-of-fit* tests on multinomial data to determine if data was drawn from a multinomial distribution with a certain probability vector, and (2) *independence* tests for checking whether two categorical random variables are independent of each other. Both tests use the chi-squared test statistic. Rogers and Kifer [40] develop new test statistics for differentially private hypothesis testing on categorical data while maintaining a given Type I error. Through the use of the subsample-and-aggregate framework, Barrientos et al. [9] compute univariate $t$-statistics by first partitioning the data into $M$ disjoint subsets, estimating the statistic on each subset, truncating the statistic at some threshold $a$, and then adding noise from a Laplace distribution to the average of the truncated $t$-statistics. Our framework requires a clipping parameter (similar to $a$) but does not require any others (e.g., the number of subsets). As noted in that work, the parameter $M$ plays a significant role in the performance—and tuning—of their tests while our tests require no such parameter tuning on partitioning of the data. A subset of previous work [47, 14, 45, 21] focus on differentially private independence tests between a categorical and continuous variables. Some of these works produce nonparametric tests which require little or no distributional assumptions on the data generation process. Specifically, Couch, Kazan, Shi, Bray, and Groce [21] develop DP analogues of rank-based nonparametric tests such as Kruskal-Wallis and Mann-Whitney signed-rank tests. The Kruskal-Wallis test, for example, can be used to determine whether the medians of two or more groups are the same. We adapt their test to the setting of linear regression by using it to compare the distributions of slopes between the two groups. Wang, Lee, and Kifer [50] develop DP versions of likelihood ratio and chi-squared tests, showing a modified equivalence between chi-squared and likelihood ratio tests. In the space of differentially private hypothesis testing, previous work introduce methods for differentially private identity and equivalence testing of discrete distributions [1, 2, 4, 5]. A differentially private version of the log-likelihood ratio test for the Neyman-Pearson lemma has also been shown to exist [15]. Furthermore, Awan and Slavkovic [8] derive uniformly most powerful DP

tests for simple hypotheses for binomial data. Suresh [44] proposes a hypothesis test, which can be made to satisfy differential privacy, that is robust to distribution perturbations under Hellinger distance. Sheffet and Kairouz, Oh, Viswanath [42, 36] also consider hypothesis testing, although in the local setting of differential privacy.

## 1.4 Differential Privacy

For the definitions below, we say that two databases $\mathbf{x}$ and $\mathbf{x}'$ are neighboring, expressed notationally as $d(\mathbf{x}, \mathbf{x}') = 1$ for any $\mathbf{x}, \mathbf{x}' \in \mathcal{X}^n$, if $\mathbf{x}$ differs from $\mathbf{x}'$ in exactly one row.

**Definition 1.1** (Differential Privacy [24, 22]). Let $\mathcal{M} : \mathcal{X}^n \to \mathcal{R}$ be a (randomized) mechanism. For any $\epsilon \geq 0, \delta \in [0, 1]$, we say $\mathcal{M}$ is $(\epsilon, \delta)$-**differentially private** if for all neighboring databases $\mathbf{x}, \mathbf{x}' \in \mathcal{X}^n, d(\mathbf{x}, \mathbf{x}') = 1$ and every $S \subseteq \mathcal{R}$,

$$\mathbb{P}[\mathcal{M}(\mathbf{x}) \in S] \leq e^\epsilon \cdot \mathbb{P}[\mathcal{M}(\mathbf{x}') \in S] + \delta.$$

If $\delta = 0$, then we say $\mathcal{M}$ is $\epsilon$-DP, sometimes referred to as **pure** differential privacy. Typically, $\epsilon$ is a small constant (e.g., $\epsilon \in [0.1, 1]$) and $\delta \leq 1/\operatorname{poly}(n)$ is cryptographically small.

**Definition 1.2** (Zero-Concentrated Differential Privacy (zCDP) [12]). Let $\mathcal{M} : \mathcal{X}^n \to \mathcal{R}$ be a (randomized) mechanism. For any neighboring databases $\mathbf{x}, \mathbf{x}' \in \mathcal{X}^n, d(\mathbf{x}, \mathbf{x}') = 1$, we say $\mathcal{M}$ satisfies $\rho$-**zCDP** if for all $\alpha \in (1, \infty)$,

$$D_\alpha(\mathcal{M}(\mathbf{x}) \| \mathcal{M}(\mathbf{x}')) \leq \rho \cdot \alpha,$$

where $D_\alpha(\mathcal{M}(\mathbf{x}) \| \mathcal{M}(\mathbf{x}'))$ is the Rényi divergence of order $\alpha$ between the distribution of $\mathcal{M}(\mathbf{x})$ and the distribution of $\mathcal{M}(\mathbf{x}')$. [4]

In this paper, we will primarily use $\rho$-zCDP as our definition of differential privacy, adding noise from a Gaussian distribution to ensure zCDP.

## 2 Differentially Private Linear Regression Testing

The goal of hypothesis testing is to infer, based on data, which of two hypothesis, $H_0$ (the null hypothesis) or $H_1$ (the alternative hypothesis), should be rejected. Let $P_\theta$ be a family of probability distributions parameterized by $\theta \in \Omega$. For some unknown parameter $\theta \in \Omega$, let $Z \sim P_\theta$ be the observed data. Then the two competing hypothesis are: $H_0 : \theta \in \Omega_0$ vs. $H_1 : \theta \in \Omega_1$, where $(\Omega_0, \Omega_1)$ form a partition of $\Omega$. We will consider hypothesis testing in the linear model $Y = X\beta + \mathbf{e}$, where $X \in \mathbb{R}^{n \times p}$ is a matrix of known constants, $\beta \in \mathbb{R}^p$ is the parameter vector that determines the linear relationship between $X$ and the dependent variable $Y$, and $\mathbf{e}$ is a random vector such that for all $i \in [n], \mathbb{E}[e_i] = 0, \operatorname{var}[e_i] = \sigma_e^2$. Furthermore, for all $i \neq j \in [n], \operatorname{cov}(e_i, e_j) = 0$. Note that the simple linear regression model, $y_i = \beta_2 + \beta_1 \cdot x_i + e_i$ for scalars $x_i, y_i$ and $e_i \, \forall i \in [n]$, can be cast as a linear model as follows: $X \in \mathbb{R}^{n \times 2}$ where the first column is an all-ones vector and the second column $(x_1, \ldots, x_n)^T$.

We will consider the general linear model: $Y \sim \mathcal{N}(X\beta, \sigma_e^2 I_{n \times n})$, where $I_{n \times n}$ is the $n \times n$ identity matrix. Let $\omega$ be an $r$-dimensional linear subspace of $\mathbb{R}^p$ and $\omega_0$ be a $q$-dimensional linear subspace of $\omega$ such that $0 \leq q < r$. We will consider hypothesis tests of the form: (i) $H_0$: $\beta \in \omega_0$; (ii) $H_1$: $\beta \in \omega \setminus \omega_0$.

Let $\hat\beta$ and $\hat\beta^N$ denote the least squares estimates under the alternative and null hypothesis respectively. In other words, $\hat\beta^N = \operatorname{argmin}_{z \in \omega_0} \|Xz - Y\|^2, \quad \hat\beta = \operatorname{argmin}_{z \in \omega} \|Xz - Y\|^2$.

---

[4]A related differential privacy notion, in terms of the Rényi divergence, is given in [39].

The **test statistic** (the $F$-statistic) we will use is equivalent to the generalized likelihood ratio test statistic

$$T = \left(\frac{n-r}{r-q}\right) \cdot \frac{\|Y - X\hat{\beta}^N\|^2 - \|Y - X\hat{\beta}\|^2}{\|Y - X\hat{\beta}\|^2} \tag{1}$$

$$= \left(\frac{n-r}{r-q}\right) \cdot \frac{\|X\hat{\beta} - X\hat{\beta}^N\|^2}{\|Y - X\hat{\beta}\|^2} \tag{2}$$

$$= \frac{1}{r-q} \cdot \frac{\|X\hat{\beta} - X\hat{\beta}^N\|^2}{S^2}, \tag{3}$$

where $S^2 = \|Y - X\hat{\beta}\|^2/(n-r)$. The vectors $Y - X\hat{\beta}$ and $X\hat{\beta} - X\hat{\beta}^N$ can be shown to be orthogonal, so that $\|Y - X\hat{\beta}^N\|^2 = \|Y - X\hat{\beta}\|^2 + \|X\hat{\beta} - X\hat{\beta}^N\|^2$ by the Pythagorean theorem [37]. When $r - q = 1$, this test is *uniformly most powerful unbiased* and for $r - q > 1$, the test is most powerful amongst all tests that satisfy certain symmetry restrictions [37].

To design a Monte Carlo hypothesis test, we follow a similar route to Gaboardi, Lim, Rogers, and Vadhan [30]. In Algorithm 1, we provide a framework to perform DP Monte Carlo tests using a parametric bootstrap based on a test statistic.

Let DPStats be a procedure that uses one or more statistics of $X, Y$ to produce DP statistics that can be used to reject or fail to reject the null hypothesis. In this paper, DPStats will satisfy $\rho$-zCDP (Zero-Concentrated Differential Privacy). $T$ is the test statistic computation procedure. As done in [30], for example, we will assume the dataset sizes are public information.

Let $T = T(\hat{\theta}_1)$ be the non-private test statistic procedure given $\hat{\theta}_1 = \hat{\theta}_1(X, Y)$. The goal is to compute $T(\hat{\theta}_1)$ where $\tilde{\theta}_1$ is an approximation of $\hat{\theta}_1$. DPStats returns $\tilde{\theta}_0$ and $\tilde{\theta}_1$. If $\tilde{\theta}_0$ and $\tilde{\theta}_1$ is not $\perp$ ($\perp$ is returned whenever the perturbed statistics cannot be used to simulate the null distributions), then we use $\tilde{\theta}_1$ to compute the DP test statistic and $\tilde{\theta}_0$ to simulate the null. $P_{\tilde{\theta}_0}$ represents the distribution from which we will sample from to simulate the null distribution. When $(X, y) \sim P_{\tilde{\theta}_0}$ for $\theta_0 \in \Omega_0$ and we set $\tilde{\theta}_1 = \tilde{\theta}_1(X, Y)$ and sample $(X', y') \sim P_{\tilde{\theta}_0}$, then $\hat{\theta}_1((X', y'))$ has approximately the same distribution as $\hat{\theta}_1((X, y))$. $\tilde{T}$ will denote the test statistic applied to the DP sufficient statistics.

---

**Algorithm 1** Monte Carlo DP Test Framework.

---

 1: **Data**: $X \in \mathbb{R}^{n \times p}; Y \in \mathbb{R}^n$
 2: **Input**: $n$ (dataset size); $\rho$ (privacy-loss parameter); $\alpha$ (target significance); $T$ (test statistic)
 3: $(\tilde{\theta}_0, \tilde{\theta}_1) = \text{DPStats}(X, Y, n, \rho)$
 4: **if** $\tilde{\theta}_0 = \tilde{\theta}_1 = \perp$ **then**
 5:     **return** Fail to Reject the null
 6: **end if**
 7: // non-DP test statistic applied to DP sufficient statistics
 8: $\tilde{t} = T(\tilde{\theta}_1)$
 9: Select $K > 1/\alpha$
10: **for** $k = 1 \ldots K$ **do**
11:     $\forall i \in [n], {}^k X_i, {}^k y_i \sim P_{\tilde{\theta}_0}$
12:     ${}^k \tilde{\theta}_1, {}^k \tilde{\theta}_0 = \text{DPStats}({}^k X_i, {}^k y_i, n, \rho)$
13:     Obtain $t_k$ from $T({}^k \tilde{\theta}_1)$
14: **end for**
15: Sort $t_{(1)} \leq \cdots \leq t_{(K)}$
16: Compute threshold $t_{(r)}$ where $r = \lceil (K+1)(1-\alpha) \rceil$
17: **if** $\tilde{t} > t_{(r)}$ **then**
18:     **return** Reject the null
19: **else**
20:     **return** Fail to Reject the null
21: **end if**

---

## 2.1 Differentially Private Algorithms

**Notation** For any constants $A, B \in \mathbb{R}$, we use $Y|_B^A$ to mean that the (random) variable $Y$ will be truncated to have an upper bound of $A$ and a lower bound of $B$. $\to$, $\xrightarrow{P}$, and $\xrightarrow{D}$ denotes convergence toward a limit, convergence in probability, and convergence in distribution, respectively. See Section B in the Appendix for more details.

---

**Algorithm 2** $\rho$-zCDP procedure $\texttt{DPStats}_L$

---

1: **Data**: $X \in \mathbb{R}^{n \times 2}; Y \in \mathbb{R}^n$
2: **Input**: integer $n \geq 2$; $r, q$; $\rho > 0, \Delta > 0$
3: Set $\rho = \rho/5$ and compute the following:

- $\tilde{x} = \frac{1}{n} \sum_{i=1}^n x_i]_{-\Delta}^{\Delta} + \mathcal{N}(0, \frac{2\Delta^2}{\rho n^2})$. $\quad \tilde{y} = \frac{1}{n} \sum_{i=1}^n y_i]_{-\Delta}^{\Delta} + \mathcal{N}(0, \frac{2\Delta^2}{\rho n^2})$.
- $\widetilde{x^2} = \frac{1}{n} \sum_{i=1}^n x_i^2]_0^{\Delta^2} + \mathcal{N}(0, \frac{\Delta^4}{2\rho n^2})$. $\quad \widetilde{xy} = \frac{1}{n} \sum_{i=1}^n x_i y_i]_{-\Delta^2}^{\Delta^2} + \mathcal{N}(0, \frac{2\Delta^4}{\rho n^2})$.
- $\widetilde{y^2} = \frac{1}{n} \sum_{i=1}^n y_i^2]_0^{\Delta^2} + \mathcal{N}(0, \frac{\Delta^4}{2\rho n^2})$.
- 

$$\tilde{\beta}_1 = \frac{\widetilde{xy} - \tilde{x}\tilde{y}}{\widetilde{x^2} - \tilde{x}^2}, \quad \tilde{\beta}_2 = \frac{\tilde{y} \cdot \widetilde{x^2} - \tilde{x} \cdot \widetilde{xy}}{\widetilde{x^2} - \tilde{x}^2}.$$

- 

$$\widetilde{S_0^2} = \frac{n\widetilde{y^2} - 2\tilde{\beta}_2 n\tilde{y} + n\tilde{\beta}_2^2}{n - r}.$$

$$\widetilde{S^2} = \frac{n\widetilde{y^2} - 2\tilde{\beta}_2 n\tilde{y} - 2\tilde{\beta}_1 n\widetilde{xy} + n\tilde{\beta}_2^2 + 2\tilde{\beta}_1\tilde{\beta}_2 n\tilde{x} + \tilde{\beta}_1^2 n\widetilde{xy}}{n - r}.$$

4: $(\tilde{\theta}_0, \tilde{\theta}_1) = (\perp, \perp)$
5: **if** $\min(\widetilde{S_0^2}, (n\widetilde{x^2} - n\tilde{x}^2)/(n - 1)) > 0$ **then**
6: $\quad \tilde{\theta}_0 = (\tilde{\beta}_2, \tilde{x}, \widetilde{x^2}, \widetilde{S_0^2}, n)$
7: $\quad \tilde{\theta}_1 = (\tilde{\beta}_1, \tilde{x}, \widetilde{x^2}, \widetilde{S^2}, n)$
8: **end if**
9: **return** $(\tilde{\theta}_0, \tilde{\theta}_1)$

---

**Lemma 2.1** (Lemma C.1). *For any $\rho, \Delta > 0$, Algorithm 2 satisfies $\rho$-zCDP.*

**Instantiating Algorithm 1 for the Linear Tester:** If the procedure $\texttt{DPStats}_L$ returns $(\perp, \perp)$, then we fail to reject the null. Otherwise, we use the returned statistics $\tilde{\theta}_1 = (\tilde{\beta}_1, \tilde{x}, \widetilde{x^2}, \widetilde{S^2}, n)$ to create the test statistic $T(\tilde{\theta}_1) = \frac{\tilde{\beta}_1^2 \cdot n \cdot (\widetilde{x^2} - \tilde{x}^2)}{\widetilde{S^2}}$ and use $\tilde{\theta}_0 = (\tilde{y}, \tilde{x}, \widetilde{x^2}, \widetilde{S_0^2}, n)$ to simulate the null distributions (to decide to reject or fail to reject the null hypothesis). $P_{\tilde{\theta}_0}$ is instantiated as a normal distribution and used to generate $^k x_i$ distributed as $\mathcal{N}(\tilde{x}, (n\widetilde{x^2} - n\tilde{x}^2)/(n - 1))$ and $^k y_i$ as $\tilde{\beta}_2 + e_i$, $e_i \sim \mathcal{N}(0, \widetilde{S_0^2})$ for all $i \in [n]$.

**Instantiating Algorithm 1 for the $F$-statistic Mixture Tester:** If the procedure $\texttt{DPStats}_M$ returns $(\perp, \perp)$, then we fail to reject the null. Otherwise, we use the returned statistics $\tilde{\theta}_1 = (\tilde{\beta}_1, \tilde{\beta}_2, \widetilde{x^2}_1, \widetilde{x^2}_2, \widetilde{x^2}, \widetilde{S^2}, n_1, n_2, n)$ to create the test statistic and use $\tilde{\theta}_0 = (\tilde{\beta}_1, \tilde{x}, \widetilde{x^2}, \widetilde{S_0^2}, n_1, n_2, n)$ to simulate the null distributions. $P_{\tilde{\theta}_0}$ is instantiated as a normal distribution and used to generate $^k x_i$ distributed as $\mathcal{N}(\tilde{x}, (n\widetilde{x^2} - n\tilde{x}_1^2)/(n - 1))$ and to generate $^k y_i$ distributed as $\tilde{\beta}_1 {}^k x_i + e_i$, $e_i \sim \mathcal{N}(0, \widetilde{S_0^2})$ for either group 1 with size $n_1$ or group 2 with size $n - n_1$.

## 2.2 Convergence

We now state and interpret our main theoretical result. For more details on the proof and consequences of the theorem, see Section D.

**Theorem 2.2** (Theorem D.1). *Let $\sigma_e > 0$, $r = p = 2, q = 1$, and $\beta \in \mathbb{R}^p$. For every $n \in \mathbb{N}$ with $n > r$, let $X_n \in \mathbb{R}^{n \times p}$ be the design matrix where the first column is an all-ones vector and the second column is $(x_1, \ldots, x_n)^T$. Let $\Delta = \Delta_n > 0$ be a sequence of clipping bounds, $\rho = \rho_n > 0$ be a sequence of privacy parameters, and $\eta_n^2 = \frac{\|X_n\beta - X_n\beta^N\|^2}{\sigma_e^2}$. Under the general linear model (GLM), $Y_n \sim \mathcal{N}(X_n\beta, \sigma_e^2 I_{n \times n})$. Let $\tilde{\beta}$ and $\tilde{\beta}^N$ be the DP least-squares estimate of $\beta$, obtained in Algorithm 2, under the alternative and null hypotheses, respectively. Let $\tilde{T} = \tilde{T}_n$ be the DP F-statistic computed from DP sufficient statistics via Algorithm 2 and Equation (3). Suppose the following conditions hold:*

1. *$\exists c_x, c_{x^2} \in \mathbb{R}$ such that $\bar{x} \to c_x$, $\overline{x^2} \to c_{x^2}$, and $c_{x^2} > c_x^2$,*

2. *$\exists \eta \in \mathbb{R}$ such that $\eta_n^2 \to \eta^2$,*

3. *$\frac{\Delta_n^2}{\rho_n n}, \frac{\Delta_n^4}{\rho_n n} \to 0$,*

4. *$\mathbb{P}[\exists i \in [n], y_i \notin [-\Delta_n, \Delta_n]] \to 0$ and $\forall i \in [n], x_i \in [-\Delta_n, \Delta_n]$.*

*Then we obtain the following results: Under the null hypothesis: $\tilde{\beta}^N = \tilde{\beta}_n^N \xrightarrow{P} \beta$. Under the alternative hypothesis: $\tilde{\beta} = \tilde{\beta}_n \xrightarrow{P} \beta$. $\tilde{T} = \tilde{T}_n \xrightarrow{D} \frac{\chi_{r-q}^2(\eta^2)}{r-q}$.*

The condition that $\mathbb{P}[\exists i \in [n], y_i \notin [-\Delta_n, \Delta_n]] \to 0$ (Condition 4 in Theorem 2.2), holds by a Gaussian tail bound, if $\exists k > 0$ such that for all $i \in [n]$, $\Delta_n \geq |\beta_1 x_i + \beta_2| + \sigma_e \sqrt{\log 2n^k}$.

$T_n$ is the non-private F-statistic while $\tilde{T}_n$ is the DP F-statistic constructed from DP sufficient statistics obtained via Algorithm 2. The main theorem in this section is Theorem 2.2, which shows the convergence, in distribution, of $\tilde{T}_n$ to the asymptotic distribution of $T_n$, the chi-squared distribution. As a corollary, the statistical power of $\tilde{T}_n$ converges to the statistical power of $T_n$. While Theorem 2.2 is specialized to the simple linear regression setting (i.e., $p = 2$), it can easily be extended to multiple linear regression.

Without privacy, we show that $T_n$ follows an F-distribution and prove convergence for its components:

**Theorem 2.3** (Theorem F.2). *For every $n \in \mathbb{N}$ with $n > r$, let $X = X_n \in \mathbb{R}^{n \times p}$ be the design matrix. Under the general linear model $Y = Y_n \sim \mathcal{N}(X_n\beta, \sigma_e^2 I_{n \times n})$,*

$$T = T_n \sim F_{r-q,n-r}(\eta_n^2), \quad \eta_n^2 = \frac{\|X_n\beta - X_n\beta^N\|^2}{\sigma_e^2},$$

*where $F_{n,m}$ is the F-distribution with parameters $n$, $m$, $\beta^N = \mathbb{E}[\hat{\beta}^N]$, $q$ is the dimension of $\omega_0$, and $r$ is the dimension of $\omega$ with $0 \leq q < r$.*

*Furthermore,*

1.
$$\|Y_n - X_n\hat{\beta}\|^2 \sim \mathcal{X}_{n-r}^2 \sigma_e^2, \quad \|X_n\hat{\beta} - X_n\hat{\beta}^N\|^2 \sim \mathcal{X}_{r-q}^2(\eta_n^2)\sigma_e^2.$$

2. *If there exists $\eta \in \mathbb{R}$ such that $\frac{\|X_n\beta - X_n\beta^N\|^2}{\sigma_e^2} \to \eta^2$, then*

$$T = T_n \sim F_{r-q,n-r}(\eta_n^2) \xrightarrow{D} \frac{\chi_{r-q}^2(\eta^2)}{r-q}.$$

3. *We have*

$$\frac{\|Y_n - X_n\hat{\beta}\|^2}{n-r} \xrightarrow{P} \sigma_e^2.$$

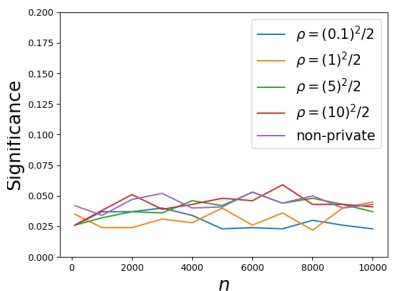

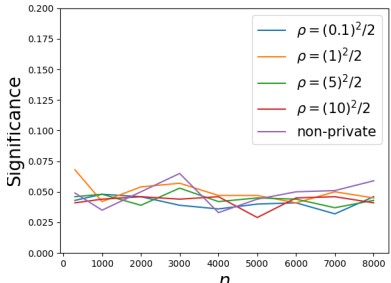

(a) Significance for testing a linear relationship. $x_i \sim \mathcal{N}(0.5, 1)$, $y_i \sim 0 \cdot x_i + \mathcal{N}(0, 0.001^2)$. $\Delta = 2$.

(b) Significance for testing a linear relationship. $x_i \sim \mathcal{N}(0.5, 1)$, $y_i \sim 0 \cdot x_i + \mathcal{N}(0, 1^2)$. $\Delta = 2$.

Figure 1

*The values $\beta = \mathbb{E}[\hat{\beta}]$, $\beta^N = \mathbb{E}[\hat{\beta}^N]$ are the expected values of our parameter estimates under the alternative and null hypotheses respectively.*

Above, the **noncentral $F$-distribution** $F_{n,m}(\lambda)$, with parameters $n, m$ and noncentrality parameter $\lambda$ is the distribution of $\frac{\chi_n^2(\lambda)/n}{\chi_m^2/m}$, the ratio of two scaled chi-squared random variables.

# 3 Experimental Evaluation

In the appendix (Section E), we provide a more thorough suite of experiments to support our claims and fully discuss how we compute the empirical power.

**General Parameter Setup for Synthetic Data:** For experimental evaluation on synthetic datasets, we generated datasets with sizes between $n = 100$ and $n = 10,000$. For both the linear relationship and mixture model tests on synthetic data below, we consider the following values of $\rho$: $\{0.1^2/2, 1^2/2, 5^2/2, 10^2/2\}$. We draw the independent variables $x_1, \ldots, x_n$ according to a few different distributions: Normal, Uniform, Exponential. For all tests below, the clipping parameter is either set to $\Delta = 2$ or $\Delta = 3$. For the synthetic data, the dependent variable $Y$ is generated using the linear or mixture model specification described in previous sections and by fixing or varying parameters (such as $\sigma_e$).

## 3.1 Testing a Linear Relationship on Synthetic Data

**Evaluating the Significance for Normally Distributed Independent Variables**: Generally, we see that the significance remains below the target significance level, on average, for all values of $\rho$. For the linear relationship tester, when the standard deviation of the dependent variable ($\sigma_e$) is small (Figure 1a), we see that the true significance level is well below the target significance of 0.05, which is fine (but conservative). We conjecture that this happens because when $\sigma_e$ is small: (i) we fail to reject when the noisy estimate of $\sigma_e$ is $\leq 0$; or (ii) the test statistic under the null distribution will be almost always 0 since under the null (even without privacy), the standard deviation of the test statistic is proportional to $\sigma_e$. In Figures 1a and 1b, we see the significance of the linear tester attains the target (of 0.05) as we vary the noise in the dependent variable $\sigma_e$.

## 3.2 Testing Mixture Models on Synthetic Data

**Evaluating the Power as we Increase the Variance of the Independent Variable**: In Figures 2a and 2b, we see that the $F$-statistic method outperforms the Kruskal-Wallis (KW) method when the variance of the independent variable is much larger (10x) than previously.

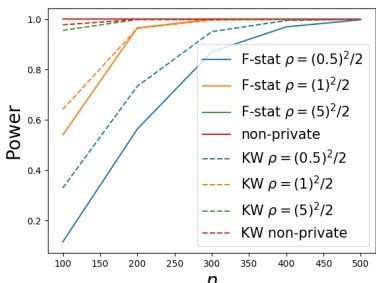

(a) Power for Kruskal-Wallis versus the $F$-statistic. $x_i \sim \mathcal{N}(0.5, 1)$, $y_i \sim -1 \cdot x_i + \mathcal{N}(0, 1)$ for Group 1. $y_i \sim 1 \cdot x_i + \mathcal{N}(0, 1)$ for Group 2. $\Delta = 3$.

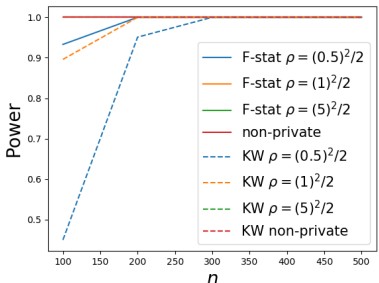

(b) Power for Kruskal-Wallis versus the $F$-statistic. $x_i \sim \mathcal{N}(0.5, 10)$, $y_i \sim -1 \cdot x_i + \mathcal{N}(0, 1)$ for Group 1. $y_i \sim 1 \cdot x_i + \mathcal{N}(0, 1)$ for Group 2. $\Delta = 3$.

Figure 2

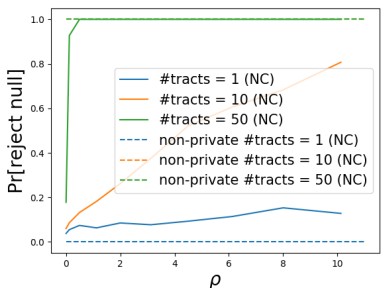

(a) Probability of rejecting null for testing a linear relationship in NC. $\Delta = 2$.

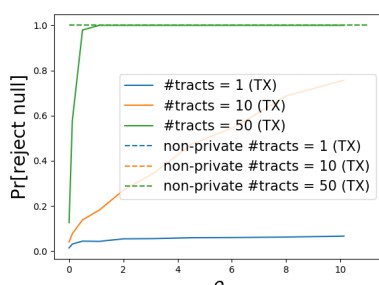

(b) Probability of rejecting null for testing a linear relationship in TX. $\Delta = 2$.

Figure 3

### 3.3 Testing on Opportunity Insights Data

The Opportunity Insights (OI) team gave us simulated data for census tracts from the following states in the United States: Idaho, Illinois, New York, North Carolina, Texas, and Tennessee. The dependent and independent variables are the child and parent national income percentiles, respectively. Figures 3a, and 3b show the probability of rejecting the null as we increase the parameter $\rho$ when using the DP linear tester.

## 4 Conclusion

We have developed differentially private hypothesis tests for testing a linear relationship in data and for testing for mixtures in linear regression models. We also show that the DP $F$-statistic converges to the asymptotic distribution of the non-private $F$-statistic. Through experiments, we show that our Monte Carlo tests achieve significance that is less than the target significance level across a wide variety of experiments. Furthermore, our tests generally have a high power, getting higher as we increase the dataset size and/or relax the privacy parameter. Even on small datasets (in the hundreds) with small slopes, our tests retain the small significance while having a good power. We have provided formal statements for the DP $F$-statistic in the asymptotic regime. We leave to future work the task of theoretically analyzing the procedures in the non-asymptotic regime. Experimental evaluation is done on simulated data for the Opportunity Atlas tool, UCI datasets, and on synthetic datasets of varying distributions on the independent variable (normal, exponential, and uniform). We also identify regimes where the power of our tests are low (e.g., when the variance of the dependent variable is very small).

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
