# Supplementary Material for "Hypothesis Testing for Differentially Private Linear Regression"

## A  Hypothesis Testing for Linear Regression

In this section, we review the theory of (non-private) hypothesis testing in the general linear model. We will consider hypothesis testing in the linear model

$$Y = X\beta + \mathbf{e},$$

where $X \in \mathbb{R}^{n \times p}$ is a matrix of known constants, $\beta \in \mathbb{R}^p$ is the parameter vector that determines the linear relationship between $X$ and the dependent variable $Y$, and $\mathbf{e}$ is a random vector such that for all $i \in [n]$, $\mathbb{E}[e_i] = 0$, $\text{var}[e_i] = \sigma_e^2$. Furthermore, for all $i \neq j \in [n]$, $\text{cov}(e_i, e_j) = 0$.

Note that the simple linear regression model, $y_i = \beta_2 + \beta_1 \cdot x_i + e_i$ for scalars $x_i, y_i$ and $e_i\ \forall i \in [n]$, can be cast as a linear model as follows: $X \in \mathbb{R}^{n \times 2}$ where

$$X = \begin{pmatrix} 1 & x_1 \\ 1 & x_2 \\ \cdots & \cdots \\ 1 & x_{n-1} \\ 1 & x_n \end{pmatrix}. \tag{4}$$

We will consider the general linear model: $Y \sim \mathcal{N}(X\beta, \sigma_e^2 I_{n \times n})$, where $I_{n \times n}$ is the $n \times n$ identity matrix. Let $\omega$ be an $r$-dimensional linear subspace of $\mathbb{R}^p$ and $\omega_0$ be a $q$-dimensional linear subspace of $\omega$ such that $0 \leq q < r$. We will consider hypothesis tests of the form:

1. $H_0$: $\beta \in \omega_0$.
2. $H_1$: $\beta \in \omega \setminus \omega_0$.

Let $\hat{\beta}$ and $\hat{\beta}^N$ denote the least squares estimates under the alternative and null hypothesis respectively. In other words,

$$\hat{\beta}^N = \operatorname*{argmin}_{z \in \omega_0} \|Xz - Y\|^2, \quad \hat{\beta} = \operatorname*{argmin}_{z \in \omega} \|Xz - Y\|^2.$$

The **test statistic** we will use is equivalent to the generalized likelihood ratio test statistic

$$T = \left( \frac{n - r}{r - q} \right) \cdot \frac{\|Y - X\hat{\beta}^N\|^2 - \|Y - X\hat{\beta}\|^2}{\|Y - X\hat{\beta}\|^2} \tag{5}$$

$$= \left( \frac{n - r}{r - q} \right) \cdot \frac{\|X\hat{\beta} - X\hat{\beta}^N\|^2}{\|Y - X\hat{\beta}\|^2} \tag{6}$$

$$= \frac{1}{r - q} \cdot \frac{\|X\hat{\beta} - X\hat{\beta}^N\|^2}{S^2}, \tag{7}$$

where $S^2 = \|Y - X\hat{\beta}\|^2/(n - r)$. The vectors $Y - X\hat{\beta}$ and $X\hat{\beta} - X\hat{\beta}^N$ can be shown to be orthogonal, so that $\|Y - X\hat{\beta}^N\|^2 = \|Y - X\hat{\beta}\|^2 + \|X\hat{\beta} - X\hat{\beta}^N\|^2$ by the Pythagorean theorem [37].

When $r - q = 1$, this test is *uniformly most powerful unbiased* and for $r - q > 1$, the test is most powerful amongst all tests that satisfy certain symmetry restrictions [37].

**Theorem A.1.** *For every $n \in \mathbb{N}$ with $n > r$, let $X = X_n \in \mathbb{R}^{n \times p}$ be the design matrix. Under the general linear model $Y = Y_n \sim \mathcal{N}(X_n\beta, \sigma_e^2 I_{n \times n})$,*

$$T = T_n \sim F_{r-q,n-r}(\eta_n^2), \quad \eta_n^2 = \frac{\|X_n\beta - X_n\beta^N\|^2}{\sigma_e^2},$$

*where $F_{n,m}$ is the F-distribution with parameters $n, m$, $\beta^N = \mathbb{E}[\hat{\beta}^N]$, $q$ is the dimension of $\omega_0$, and $r$ is the dimension of $\omega$ with $0 \leq q < r$.*

*Furthermore,*

1.

$$\|Y_n - X_n\hat{\beta}\|^2 \sim \mathcal{X}_{n-r}^2 \sigma_e^2, \quad \|X_n\hat{\beta} - X_n\hat{\beta}^N\|^2 \sim \mathcal{X}_{r-q}^2(\eta_n^2)\sigma_e^2.$$

2. *If there exists $\eta \in \mathbb{R}$ such that $\frac{\|X_n\beta - X_n\beta^N\|^2}{\sigma_e^2} \to \eta^2$, then*

$$T = T_n \sim F_{r-q,n-r}(\eta_n^2) \xrightarrow{D} \frac{\chi_{r-q}^2(\eta^2)}{r - q}.$$

3. *We have*

$$\frac{\|Y_n - X_n\hat{\beta}\|^2}{n - r} \xrightarrow{P} \sigma_e^2.$$

*The values $\beta = \mathbb{E}[\hat{\beta}], \beta^N = \mathbb{E}[\hat{\beta}^N]$ are the expected values of our parameter estimates under the alternative and null hypotheses respectively.*

The full statement and proof of Theorem A.1 appears as Theorem F.2 (Section F). Above, the **noncentral $F$-distribution** $F_{n,m}(\lambda)$, with parameters $n, m$ and noncentrality parameter $\lambda$ is the distribution of $\frac{\chi_n^2(\lambda)/n}{\chi_m^2/m}$, the ratio of two scaled chi-squared random variables. $\chi_K^2(\lambda)$ is a random variable distributed according to a chi-squared distribution with $K$ degrees of freedom and noncentrality parameter $\lambda$. That is, $\chi_K^2(\lambda)$ is distributed as the squared length of a $\mathcal{N}(v, I_{K \times K})$ vector where $v \in \mathbb{R}^K$ has length $\lambda$. Also, $\chi_K^2 \sim \chi_K^2(0)$.

## A.1 Testing a Linear Relationship in Simple Linear Regression Models

Consider the model: $y_i = \beta_2 + \beta_1 \cdot x_i + e_i$, where $e_i \sim \mathcal{N}(0, \sigma_e^2)$ are i.i.d. random variables and $x_1, \ldots, x_n$ are constants that form the following design matrix for our problem

$$X = \begin{pmatrix} 1 & x_1 \\ 1 & x_2 \\ \cdots & \cdots \\ 1 & x_{n-1} \\ 1 & x_n \end{pmatrix}.$$

In this case, $\omega = \mathbb{R}^2$ and $\omega_0 = \{\beta \in \mathbb{R}^2 : \beta_1 = 0\}$. As a result, our hypothesis is:

1. $H_0$: $\beta_1 = 0$.
2. $H_1$: $\beta_1 \neq 0$.

Note that $r = p = 2$ and $q = 1$.

Furthermore, let

$$\beta = \begin{pmatrix} \beta_2 \\ \beta_1 \end{pmatrix}, \quad \hat{\beta} = \begin{pmatrix} \hat{\beta}_2 \\ \hat{\beta}_1 \end{pmatrix}.$$

We use $\hat{\beta}^N$ to refer to the estimate of $\beta$ when the null hypothesis is true (i.e., $\beta_1 = 0$) and $\hat{\beta}$ be the estimate of $\beta$ when the alternative hypothesis holds.

For the calculations below, let

1. $\mathbf{x} \stackrel{\text{def}}{=} (x_1, x_2, \ldots, x_n)^T$, $\mathbf{y} \stackrel{\text{def}}{=} (y_1, y_2, \ldots, y_n)^T$,

2. $\bar{x} \stackrel{\text{def}}{=} \frac{1}{n} \sum_{i=1}^{n} x_i$, $\bar{y} \stackrel{\text{def}}{=} \frac{1}{n} \sum_{i=1}^{n} y_i$,

3. $\overline{x^2} \stackrel{\text{def}}{=} \frac{1}{n} \sum_{i=1}^{n} x_i^2$, $\overline{xy} \stackrel{\text{def}}{=} \frac{1}{n} \sum_{i=1}^{n} x_i y_i$,

4. $\widehat{\sigma_{xy}^2} \stackrel{\text{def}}{=} \overline{xy} - \bar{x} \cdot \bar{y}$, and $\widehat{\sigma_x^2} \stackrel{\text{def}}{=} \overline{x^2} - \bar{x}^2$.

We can then obtain the sufficient statistics

$$X^T X = \begin{pmatrix} n & n\bar{x} \\ n\bar{x} & n\overline{x^2} \end{pmatrix}, \quad X^T Y = \begin{pmatrix} n\bar{y} \\ n\overline{xy} \end{pmatrix}, \tag{8}$$

so that under the alternative hypothesis, the least squares estimate is

$$\hat{\beta} = \operatorname*{argmin}_{\beta \in \omega} \|Y - X\beta\|^2 = (X^T X)^{-1} X^T Y,$$

assuming that $X^T X$ is invertible which happens iff $\mathbf{x}$ is not the constant vector (so that $\det(X^T X) = n^2 \overline{x^2} - n^2 \bar{x}^2 = n^2 \cdot \widehat{\sigma_x^2} > 0$). Assuming invertibility, we have

$$(X^T X)^{-1} = \frac{1}{n^2 \overline{x^2} - n^2 \bar{x}^2} \begin{pmatrix} n\overline{x^2} & -n\bar{x} \\ -n\bar{x} & n \end{pmatrix}. \tag{9}$$

Thus, the least squares estimate under the alternative hypothesis is

$$\hat{\beta} = (X^T X)^{-1} X^T Y$$

$$= \frac{1}{n^2 \cdot \overline{x^2} - n^2 \cdot \bar{x}^2} \begin{pmatrix} n^2 \cdot \overline{x^2} \cdot \bar{y} - n^2 \cdot \bar{x} \cdot \overline{xy} \\ -n^2 \cdot \bar{x} \cdot \bar{y} + n^2 \cdot \overline{xy} \end{pmatrix} = \begin{pmatrix} \hat{\beta}_2 \\ \hat{\beta}_1 \end{pmatrix},$$

and further simplification results in the following slope and intercept estimates:

$$\hat{\beta}_1 = \frac{\overline{xy} - \bar{x} \cdot \bar{y}}{\overline{x^2} - \bar{x}^2} = \frac{\widehat{\sigma_{xy}^2}}{\widehat{\sigma_x^2}},$$

$$\hat{\beta}_2 = \bar{y} - \hat{\beta}_1 \bar{x} = \frac{\bar{y} \cdot \overline{x^2} - \bar{x} \cdot \overline{xy}}{\widehat{\sigma_x^2}}.$$

The square of residuals is $\|Y - X\hat{\beta}\|^2$ and an (unbiased) estimate of $\sigma_e^2$ is $S^2 = \frac{\|Y - X\hat{\beta}\|^2}{n-2}$.

Also, we can derive $\hat{\beta}^N$ as follows

$$\hat{\beta}^N = \operatorname*{argmin}_{\beta \in \omega_0} \|Y - X\beta\|^2 = \begin{pmatrix} \bar{y} \\ 0 \end{pmatrix} = \begin{pmatrix} \hat{\beta}_2^N \\ 0 \end{pmatrix}$$

so that

$$\|X\hat{\beta} - X\hat{\beta}^N\|^2 = \sum_{i=1}^{n} (\hat{\beta}_2 + \hat{\beta}_1 x_i - \hat{\beta}_2^N)^2$$

$$= \sum_{i=1}^{n} (\bar{y} - \hat{\beta}_1 \bar{x} + \hat{\beta}_1 x_i - \bar{y})^2 = \hat{\beta}_1^2 \sum_{i=1}^{n} (x_i - \bar{x})^2$$

$$= \hat{\beta}_1^2 \cdot n \cdot \widehat{\sigma_x^2}.$$

As a result, the test statistic $T$ is

$$T = \left(\frac{n-r}{r-q}\right)\frac{\|X\hat{\beta} - X\hat{\beta}^N\|^2}{\|Y - X\hat{\beta}\|^2} = \frac{\hat{\beta}_1^2}{S^2}\cdot n \cdot \widehat{\sigma_x^2}.$$

**Level-$\alpha$ Test**: Under $H_0$, $T \sim F_{1,n-2}$ since by Theorem A.1, the centrality parameter is $\eta^2 = 0$. The level-$\alpha$ test will then reject the null if $T$ is greater than the upper $\alpha$th quantile of this $F$-distribution, $F_{\alpha,1,n-2}$.

In other words, we will reject the null if

$$\frac{\hat{\beta}_1^2}{S^2}\cdot n \cdot \widehat{\sigma_x^2} > F_{\alpha,1,n-2}.$$

We see that the chance of rejecting the null increases as:

1. $\hat{\beta}_1^2$, the square of the slope estimate, increases.
2. $\widehat{\sigma_x^2}$ increases.
3. $n$ increases.
4. $S^2$ decreases.

**Power**: The power is the chance that a random variable distributed as $F_{1,n-2}(\eta^2)$ exceeds $F_{\alpha,1,n-2}$ where the centrality parameter is $\eta^2 = \frac{\beta_1^2}{\sigma_e^2}\cdot n \cdot \sigma_x^2$.

In other words, the power of the test is

$$\mathbb{P}\left[F_{1,n-2}\left(\frac{\beta_1^2}{\sigma_e^2}\cdot n \cdot \widehat{\sigma_x^2}\right) > F_{\alpha,1,n-2}\right],$$

where the probability is over the draws of the $F$-distribution.

## A.2   Testing for Mixtures in Simple Linear Regression Models

The goal of testing mixtures is to detect the presence of sub-populations. Consider the model where $n = n_1 + n_2, n_1, n_2 > 0, \beta_1, \beta_2 \in \mathbb{R}$ with the following generation model:

- $y_i = \beta_1 \cdot x_i + e_i$ for $i \in [n_1]$.
- $y_i = \beta_2 \cdot x_i + e_i$ for $i \in \{n_1 + 1, \ldots, n\}$.

where $e_i \sim \mathcal{N}(0, \sigma_e^2)$ are i.i.d. random variables and $x_1, \ldots, x_n$ are constants that form the following design matrix for our problem

$$X = \begin{pmatrix} x_1 & 0 \\ \cdots & \cdots \\ x_{n_1} & 0 \\ 0 & x_{n_1+1} \\ \cdots & \cdots \\ 0 & x_n \end{pmatrix}.$$

Note that $X$ is of full rank (except if all the $x_i$'s are 0 either for all $i \in [n_1]$ or for all $i \in \{n_1 + 1, \ldots, n\}$). Furthermore, $r = p = 2$.

In this case, $\omega = \mathbb{R}^2$ and $\omega_0 = \{\beta \in \mathbb{R}^2 : \beta_1 = \beta_2\}$. As a result, our hypothesis is:

1. $H_0$: $\beta_1 = \beta_2$.
2. $H_1$: $\beta_1 \neq \beta_2$.

Furthermore, let

$$\beta = \begin{pmatrix} \beta_1 \\ \beta_2 \end{pmatrix}, \quad \hat{\beta} = \begin{pmatrix} \hat{\beta}_1 \\ \hat{\beta}_2 \end{pmatrix}.$$

We use $\hat{\beta}^N$ to refer to the estimate of $\beta$ when the null hypothesis is true (i.e., $\beta_1 = \beta_2$) and $\hat{\beta}$ be the estimate of $\beta$ when the alternative hypothesis holds.

For the calculations below, let $n_2 = n - n_1$ and

- $\overline{x^2}_1 = \frac{1}{n_1}\sum_{i=1}^{n_1} x_i^2, \overline{x^2}_2 = \frac{1}{n_2}\sum_{i=n_1+1}^{n} x_i^2, \overline{x^2} = \frac{1}{n}\sum_{i=1}^{n} x_i^2$.
- $\overline{xy}_1 = \frac{1}{n_1}\sum_{i=1}^{n_1} x_i y_i, \overline{xy}_2 = \frac{1}{n_2}\sum_{i=n_1+1}^{n} x_i y_i, \overline{xy} = \frac{1}{n}\sum_{i=1}^{n} x_i y_i$.

We can then obtain

$$X^T X = \begin{pmatrix} n_1\overline{x^2}_1 & 0 \\ 0 & n_2\overline{x^2}_2 \end{pmatrix}, X^T Y = \begin{pmatrix} n_1\overline{xy}_1 \\ n_2\overline{xy}_2 \end{pmatrix},$$

so that, assuming $\overline{x^2}_1, \overline{x^2}_2 > 0$, we have

$$\hat{\beta} = (X^T X)^{-1} X^T Y = \begin{pmatrix} \overline{xy}_1/\overline{x^2}_1 \\ \overline{xy}_2/\overline{x^2}_2 \end{pmatrix}.$$

Furthermore,

$$\hat{\beta}^N = \begin{pmatrix} \overline{xy}/\overline{x^2} \\ \overline{xy}/\overline{x^2} \end{pmatrix},$$

since under the null hypothesis ($\beta_1 = \beta_2$), the design matrix "collapses" to

$$X_0 = \begin{pmatrix} x_1 \\ \cdots \\ x_{n_1} \\ x_{n_1+1} \\ \cdots \\ x_n \end{pmatrix},$$

so that $X_0^T X_0 = \sum_{i=1}^{n} x_i^2 = n\overline{x^2}$ and $X_0^T Y = \sum_{i=1}^{n} x_i y_i = n\overline{xy}$.

The squares of residuals is $\|Y - X\hat{\beta}\|^2$ and an (unbiased) estimate of $\sigma_e^2$ is $S^2 = \frac{\|Y - X\hat{\beta}\|^2}{n-2}$.

**Lemma A.2.**

$$\|X\hat{\beta} - X\hat{\beta}^N\|^2 = \frac{n_1\overline{x^2}_1 n_2\overline{x^2}_2}{n\overline{x^2}}(\hat{\beta}_1 - \hat{\beta}_2)^2,$$

where $X$ is the design matrix, $\hat{\beta}, \hat{\beta}^N$ are the least squares estimates under the alternative and null hypothesis, respectively.

*Proof.* First, from previous calculations, we obtained

$$\hat{\beta} = \begin{pmatrix} \overline{xy}_1/\overline{x^2}_1 \\ \overline{xy}_2/\overline{x^2}_2 \end{pmatrix}, \quad \hat{\beta}^N = \begin{pmatrix} \overline{xy}/\overline{x^2} \\ \overline{xy}/\overline{x^2} \end{pmatrix}.$$

Then $\hat{\beta}_1^N = \hat{\beta}_2^N$ is a weighted average of $\hat{\beta}_1$ and $\hat{\beta}_2$:

$$\hat{\beta}_1^N = \frac{n_1\overline{x^2}_1}{n\overline{x^2}}\hat{\beta}_1 + \frac{n_2\overline{x^2}_2}{n\overline{x^2}}\hat{\beta}_2.$$

Using this calculation, we can obtain that

$$\hat{\beta}_1 - \hat{\beta}_1^N = \frac{n_2\overline{x^2}_2(\hat{\beta}_1 - \hat{\beta}_2)}{n\overline{x^2}}, \quad \hat{\beta}_2 - \hat{\beta}_1^N = \frac{n_1\overline{x^2}_1(\hat{\beta}_2 - \hat{\beta}_1)}{n\overline{x^2}},$$

so that

$$\begin{aligned}
\|X\hat{\beta} - X\hat{\beta}^N\|^2 &= \sum_{i=1}^{n_1}(x_i\hat{\beta}_1 - x_i\hat{\beta}_1^N)^2 + \sum_{i=n_1+1}^{n}(x_i\hat{\beta}_2 - x_i\hat{\beta}_1^N)^2 \\
&= (\hat{\beta}_1 - \hat{\beta}_1^N)^2 n_1\overline{x^2}_1 + (\hat{\beta}_2 - \hat{\beta}_1^N)^2 n_2\overline{x^2}_2 \\
&= \frac{n_1\overline{x^2}_1 n_2\overline{x^2}_2}{n\overline{x^2}}(\hat{\beta}_1 - \hat{\beta}_2)^2.
\end{aligned}$$

This completes the proof.

$\square$

By Lemma A.2, our test statistic $T$ is

$$T = \left(\frac{n-r}{r-q}\right)\frac{\|X\hat{\beta} - X\hat{\beta}^N\|^2}{\|Y - X\hat{\beta}\|^2} = \frac{n_1\overline{x^2}_1 n_2\overline{x^2}_2}{S^2 n\overline{x^2}}(\hat{\beta}_1 - \hat{\beta}_2)^2.$$

**Level-$\alpha$ Test**: Under $H_0$, $T \sim F_{1,n-2}$ since by Theorem A.1, the centrality parameter is $\eta^2 = 0$. The level-$\alpha$ test will then reject the null if $T$ is greater than the upper $\alpha$th quantile of this $F$-distribution, $F_{\alpha,1,n-2}$.

In other words, we will reject the null if

$$\frac{n_1\overline{x^2}_1 n_2\overline{x^2}_2}{S^2 n\overline{x^2}}(\hat{\beta}_1 - \hat{\beta}_2)^2 > F_{\alpha,1,n-2}.$$

We see that the chance of rejecting the null increases as:

1. $|\hat{\beta}_1 - \hat{\beta}_2|$ increases.
2. $S^2$ decreases.
3. The ratio $\frac{n_1\overline{x^2}_1 n_2\overline{x^2}_2}{n\overline{x^2}}$ increases, which is more likely to occur when $n_1\overline{x^2}_1$ is close to $n_2\overline{x^2}_2$.

**Power**: The power is the chance that a random variable distributed as $F_{1,n-2}(\eta^2)$ exceeds $F_{\alpha,1,n-2}$ where the centrality parameter is $\eta^2 = \frac{n_1\overline{x^2}_1 n_2\overline{x^2}_2}{\sigma_e^2 n\overline{x^2}}(\beta_1 - \beta_2)^2$.

In other words, the power of the test is

$$\mathbb{P}\left[F_{1,n-2}\left(\frac{n_1\overline{x^2}_1 n_2\overline{x^2}_2}{\sigma_e^2 n\overline{x^2}}(\beta_1 - \beta_2)^2\right) > F_{\alpha,1,n-2}\right],$$

where the probability is over the draws of the $F$-distribution.

### A.3 Generalization to Higher Dimensions

Testing for a linear relationship and for mixtures using the $F$-statistic can be done in the multiple linear regression model as well. The main changes that will need to be made are:

1. We can use any general design matrix $X \in \mathbb{R}^{n \times p}$; and
2. The parameter to be estimated lives in $\mathbb{R}^p$ instead i.e., $\beta \in \mathbb{R}^p$, for any $p \geq 2$.

## B   More Preliminaries and Notation

### B.1   General Hypothesis Testing

The goal of hypothesis testing is to infer, based on data, which of two hypothesis, $H_0$ (the null hypothesis) or $H_1$ (the alternative hypothesis), should be rejected.

Let $P_\theta$ be a family of probability distributions parameterized by $\theta \in \Omega$. For some unknown parameter $\theta \in \Omega$, let $Z \sim P_\theta$ be the observed data. Then the two competing hypothesis are:

$$H_0 : \theta \in \Omega_0 \text{ vs. } H_1 : \theta \in \Omega_1,$$

where $(\Omega_0, \Omega_1)$ form a partition of $\Omega$.

A **test statistic** $T$ is random variable that is a function of the observed data $Z \sim P_\theta$. $T$ can be used to decide whether to reject or fail to reject the null hypothesis. A **critical region** $S$ is the set of values for the test statistic $T$ (or correspondingly for the observed data) for which the null hypothesis will be rejected. It can be used to completely determine a test of $H_0$ versus $H_1$ as follows: We reject $H_0$ if $Y \in S$ and fail to reject $H_0$ if $Y \notin S$.

Sometimes, *external randomization* might help with choosing between hypothesis $H_0$ and $H_1$ [26, 37]. By external randomness, we mean randomness not inherent in the sample or data collection process. In order to discriminate between hypothesis $H_0$ and $H_1$, we can define a notion of a critical function that can indicate the degree to which a test statistic is within a critical region. A critical function $\phi$ with range in $[0, 1]$ characterize randomized hypothesis tests. A nonrandomized test with critical region $S$ can thus be specified as $\phi = 1_S$. Conversely, if $\phi(y)$ is always 0 or 1 for all $y$ then the critical region is $S = \{y : \phi(y) = 1\}$ for this nonrandomized test. An advantage of allowing randomization (even without DP constraints) is that convex combinations of nonrandomized tests are not possible, but convex combinations of randomized tests are possible. i.e., if $\phi_1, \phi_2$ are critical functions and $t \in (0, 1)$, then $t\phi_1 + (1 - t)\phi_2$ is also a critical function so that the set of all critical functions form a convex set. Furthermore, nontrivial differentially private tests must be randomized.

For any $\theta \in \Omega$, the ideal test would tell us when $\theta \in \Omega_0$ and when $\theta \in \Omega_1$. This can be described by a **power function** $R(\cdot)$, which gives the chance of rejecting $H_0$ as a function of $\theta \in \Omega$:

$$R(\theta) = P_\theta(Y \in S),$$

for any critical region $S$.

A "perfect" hypothesis test would have $R(\theta) = 0$ for every $\theta \in \Omega_0$ and $R(\theta) = 1$ for every $\theta \in \Omega_1$. But this is generally impossible given only the "noisy" observed data $Z \sim P_\theta$.

A **significance level** $\alpha$ can be defined as

$$\alpha = \sup_{\theta \in \Omega_0} P_\theta(Y \in S).$$

In other words, the level $\alpha$ is the worst chance of incorrectly rejecting $H_0$. Ideally, we want tests that have a small chance of error when $H_0$ should not be rejected. The *p*-**value** is the probability of finding, based on observed data, test statistics at least as extreme as when the null hypothesis holds. That is, if $T$ is the test statistic function and $t$ is the observed test statistic, then the (one-sided) *p*-value is $\mathbb{P}[T \geq t \mid H_0]$.

## B.2 Convergence and Limits

**Definition B.1** (Limit of Sequence). A sequence $\{x_n\}$ **converges** toward the limit $x$, denoted $x_n \to x$, if

$$\forall \epsilon > 0, \lim_{n \to \infty} \mathbb{P}\left[|x_n - x| > \epsilon\right] = 0.$$

We will use Definition B.1 to show convergence of random variables (in probability or distribution).

**Definition B.2** (Convergence in Probability). A sequence of random variables $\{X_n\}$ **converges in probability** toward random variable $X$, denoted $X_n \xrightarrow{P} X$, if

$$\forall \epsilon > 0, \lim_{n \to \infty} \mathbb{P}\left[|X_n - X| > \epsilon\right] = 0.$$

Convergence in Probability (Definition B.2) for a sequence of random variables $X_1, X_2, \ldots$ toward random variable $X$ can be shown if for all $\epsilon > 0, \delta > 0$, there exists $N(\epsilon, \delta) = N$ such that for all $n \geq N, \mathbb{P}[|X_n - X| > \epsilon] < \delta$.

**Definition B.3** (Convergence in Distribution). A sequence of random variables $\{X_n\}$ **converges in distribution** toward random variable $X$, denoted $X_n \xrightarrow{D} X$, if

$$\lim_{n \to \infty} F_n(x) = F(x),$$

for all $x \in \mathbb{R}$ at which $F$ is continuous. $F_n, F$ are the cumulative distribution functions for $X_n, X$ respectively.

We can generalize Definitions B.2 and B.3 to random vectors and matrices (beyond scalars) as follows: if $A \in \mathbb{R}^{K \times L}$ is a random vector, then $A_n \xrightarrow{P} A$, $A_n \xrightarrow{D} A$ denotes entry-wise convergence in probability and distribution, respectively. Also, for any distribution $\mathcal{D}$ (e.g., $\mathcal{D} = \chi_k^2$), we write $A_n \xrightarrow{D} \mathcal{D}$ as a shorthand to mean that $A_n \xrightarrow{D} A$ for random variables $A_n, A$ such that $A$ follows $\mathcal{D}$ (i.e., $A \sim \mathcal{D}$). Similarly, $\mathcal{D}_n \xrightarrow{D} \mathcal{D}$ implies that $A_n \xrightarrow{D} A$ for $A_n \sim \mathcal{D}_n$ and $A \sim \mathcal{D}$.

**Lemma B.4.** *Let $\{X_n\}$ and $\{Y_n\}$ be a sequence of random vectors and $X$ be a random vector. Then:*

1. *If $X_n \xrightarrow{D} X$ and $X_n - Y_n \xrightarrow{P} 0$, then $Y_n \xrightarrow{D} X$.*

2. *If $X_n \xrightarrow{P} X$, then $X_n \xrightarrow{D} X$.*

3. *For a constant $c \in \mathbb{R}$, if $X_n \xrightarrow{D} c$, then $X_n \xrightarrow{P} c$.*

*Proof.* Follows from Theorem 2.7 in [48]. $\qquad\square$

Lemma B.4 is a helper lemma that is useful for proving convergence results, especially on DP estimates.

In later sections, we will show that the differentially private $F$-statistic converges, in distribution, to a chi-squared distribution (as does the non-DP $F$-statistic). This convergence result holds under certain conditions.

## C   Differentially Private Monte Carlo Tests

Since the private test statistic differs from the non-private version, we have to create new statistics to account for the level-$\alpha$ Monte Carlo differentially private testing. The majority of our tests will be based on DP sufficient statistics. In the statistics literature, a statistic is considered **sufficient**, with respect to a particular model, if it provides at least as much information for the value of an unknown parameter as any other statistic that can be calculated on a given sample [37].

Previous work [41, 49, 3] perturb the sufficient statistics for ordinary least squares and use the result to compute a slope and intercept in a DP way. To add noise to ensure privacy, we typically have to truncate certain random variables. We use $Y|_B^A$ to mean that the random variable $Y$ will be truncated to have an upper bound of $A$ and a lower bound of $B$.

For all our DP OLS Monte Carlo tests that sample from a continuous Gaussian, we can instead use discrete variants (e.g., [16]). The DP OLS Monte Carlo tests below use the zero-concentrated differential privacy definition [12].

### C.1   Monte Carlo Hypothesis Testing

We now proceed to discuss our general approach for designing a Level-$\alpha$ test for the task of linear regression estimation based on sufficient statistic perturbation. We rely on a sub-routine DPStats that can produce DP statistics, given a dataset and privacy parameters, when testing. Algorithm 3 can then be specialized to test for the presence of a linear relationship and for mixture models.

To design a Monte Carlo hypothesis test, we follow a similar route to Gaboardi, Lim, Rogers, and Vadhan [30]. In Algorithm 3, we provide a framework to perform DP Monte Carlo tests using a parametric bootstrap based on a test statistic. Let DPStats be a procedure that uses one or more statistics of $X, Y$ to produce DP statistics that can be used to reject or fail to reject the null hypothesis. In this paper, DPStats will satisfy $\rho$-zCDP (Zero-Concentrated Differential Privacy). [5] $T$ is the test statistic computation procedure. As done in [30], for example, we will assume the dataset sizes are public information.

Let $T = T(\hat{\theta}_1)$ be the non-private test statistic procedure given $\hat{\theta}_1 = \hat{\theta}_1(X, Y)$. The goal is to compute $T(\tilde{\theta}_1)$ where $\tilde{\theta}_1$ is an approximation of $\hat{\theta}_1$. DPStats returns $\tilde{\theta}_0$ and $\tilde{\theta}_1$. If $\tilde{\theta}_0$ and $\tilde{\theta}_1$ is not $\perp$ ($\perp$ is returned whenever the perturbed statistics cannot be used to simulate the null distributions), then we use $\tilde{\theta}_1$ to compute the DP test statistic and $\tilde{\theta}_0$ to simulate the null. $P_{\tilde{\theta}_0}$ represents the distribution from which we will sample from to simulate the null distribution. When $(X, y) \sim P_{\tilde{\theta}_0}$ for $\theta_0 \in \Omega_0$ and we set $\tilde{\theta}_1 = \tilde{\theta}_1(X, Y)$ and sample $(X', y') \sim P_{\tilde{\theta}_0}$, then $\hat{\theta}_1((X', y'))$ has approximately the same distribution as $\hat{\theta}_1((X, y))$.

---

[5] Gaussian noise addition (for privacy) was chosen because the noise in the dependent variable is also assumed to be Gaussian. The use of the Gaussian (or truncated Gaussian) distribution for both privacy and sampling error is a convenient choice as it could result in a clearer, more compatible, theoretical analysis.

---
**Algorithm 3** Monte Carlo DP Test Framework.
---
1: **Data**: $X \in \mathbb{R}^{n \times p}; Y \in \mathbb{R}^n$
2: **Input**: $n$ (dataset size); $\rho$ (privacy-loss parameter); $\alpha$ (target significance); $T$ (test statistic)
3: $(\tilde{\theta}_0, \tilde{\theta}_1) = \texttt{DPStats}(X, Y, n, \rho)$
4: **if** $\tilde{\theta}_0 = \tilde{\theta}_1 = \perp$ **then**
5:     **return** Fail to Reject the null
6: **end if**
7: // non-DP test statistic applied to DP sufficient statistics
8: $\tilde{t} = T(\tilde{\theta}_1)$
9: Select $K > 1/\alpha$
10: **for** $k = 1 \dots K$ **do**
11:     $\forall i \in [n], {}^k X_i, {}^k y_i \sim P_{\tilde{\theta}_0}$
12:     ${}^k\tilde{\theta}_1, {}^k\tilde{\theta}_0 = \texttt{DPStats}({}^k X_i, {}^k y_i, n, \rho)$
13:     Obtain $t_k$ from $T({}^k\tilde{\theta}_1)$
14: **end for**
15: Sort $t_{(1)} \le \cdots \le t_{(K)}$
16: Compute threshold $t_{(r)}$ where $r = \lceil (K+1)(1-\alpha) \rceil$
17: **if** $\tilde{t} > t_{(r)}$ **then**
18:     **return** Reject the null
19: **else**
20:     **return** Fail to Reject the null
21: **end if**
---

## C.2 Testing a Linear Relationship

For testing a linear relationship in simple linear regression models, recall that in the non-private case, we had

$$T(X, Y, \hat{\beta}, \hat{\beta}^N, n, r, q) = \left( \frac{n-r}{r-q} \right) \frac{\|X\hat{\beta} - X\hat{\beta}^N\|^2}{\|Y - X\hat{\beta}\|^2}.$$

Accordingly, we define and compute $\tilde{T}_L(X, Y, \hat{\beta}, \hat{\beta}^N, n, r, q, \rho, \Delta) = \tilde{t}$, a private estimate of $T(X, Y, \hat{\beta}, \hat{\beta}^N, n, r, q)$. In Algorithm 4, we give the full $\rho$-zCDP procedure for computing all necessary sufficient statistics to compute $\tilde{T}_L(X, Y, \hat{\beta}, \hat{\beta}^N, n, r, q, \rho, \Delta)$.

The DP estimate of $S^2$, $\widetilde{S^2}$, can be computed as $\widetilde{S^2} = \frac{\sum_{i=1}^n (y_i - \tilde{\beta}_2 - \tilde{\beta}_1 x_i)^2]_0^{\Delta^2} + \mathcal{N}(0, \frac{\Delta^4}{2\rho})}{n-r}$. Another equivalent way to compute $\widetilde{S^2}$ is to compute $\widetilde{\overline{y^2}}$ privately and then, together with the other DP estimates, to compute $\widetilde{S^2}$. Note that under the null hypothesis, the DP estimate of $S^2$ is $\widetilde{S_0^2} = \frac{\sum_{i=1}^n (y_i - \tilde{\beta}_2)^2]_0^{\Delta^2} + \mathcal{N}(0, \frac{\Delta^4}{2\rho})}{n-r}$ which can also be equivalently computed by using $\tilde{\bar{y}}, \widetilde{\overline{y^2}}, \tilde{\beta}_2$. Also, we return $(\tilde{\theta}_0, \tilde{\theta}_1) = (\perp, \perp)$ if the computed DP sufficient statistics cannot be used to simulate the null distribution.

**Lemma C.1.** *For any $\rho, \Delta > 0$, Algorithm 4 satisfies $\rho$-zCDP.*

*Proof.* This follows from Proposition 1.6 in [12] (use of the Gaussian Mechanism). Next, we apply composition and post-processing (Lemmas 1.7 and 1.8 in [12]). The computation of the following statistics is each done to satisfy $\rho/5$-zCDP: $\tilde{\bar{x}}, \tilde{\bar{y}}, \widetilde{\overline{x^2}}, \widetilde{\overline{xy}}, \widetilde{\overline{y^2}}$. $\tilde{\beta}_1, \tilde{\beta}_2, \widetilde{S^2}, \widetilde{S_0^2}$ are post-processing of the other DP releases.

As a result, the entire procedure satisfies $\rho$-zCDP.

$\square$

**Instantiating Algorithm 3 for the Linear Tester:** If the procedure $\texttt{DPStats}_L$ returns $(\perp, \perp)$, then we fail to reject the null. Otherwise, we use the returned statistics $\tilde{\theta}_1 = (\tilde{\beta}_1, \tilde{\bar{x}}, \widetilde{\overline{x^2}}, \widetilde{S^2}, n)$ to

---

**Algorithm 4** $\rho$-zCDP procedure $\texttt{DPStats}_L$

---

1: **Data**: $X \in \mathbb{R}^{n \times 2}; Y \in \mathbb{R}^n$
2: **Input**: integer $n \geq 2$; $r, q$; $\rho > 0, \Delta > 0$
3: Set $\rho = \rho/5$ and compute the following:

- $\tilde{\bar{x}} = \frac{1}{n} \sum_{i=1}^n x_i]_{-\Delta}^{\Delta} + \mathcal{N}(0, \frac{2\Delta^2}{\rho n^2})$.
- $\tilde{\bar{y}} = \frac{1}{n} \sum_{i=1}^n y_i]_{-\Delta}^{\Delta} + \mathcal{N}(0, \frac{2\Delta^2}{\rho n^2})$.
- $\widetilde{\overline{x^2}} = \frac{1}{n} \sum_{i=1}^n x_i^2]_0^{\Delta^2} + \mathcal{N}(0, \frac{\Delta^4}{2\rho n^2})$.
- $\widetilde{\overline{xy}} = \frac{1}{n} \sum_{i=1}^n x_i y_i]_{-\Delta^2}^{\Delta^2} + \mathcal{N}(0, \frac{2\Delta^4}{\rho n^2})$.
- $\widetilde{\overline{y^2}} = \frac{1}{n} \sum_{i=1}^n y_i^2]_0^{\Delta^2} + \mathcal{N}(0, \frac{\Delta^4}{2\rho n^2})$.
- 

$$\tilde{\beta}_1 = \frac{\widetilde{\overline{xy}} - \tilde{\bar{x}}\tilde{\bar{y}}}{\widetilde{\overline{x^2}} - \tilde{\bar{x}}^2}, \quad \tilde{\beta}_2 = \frac{\tilde{\bar{y}} \cdot \widetilde{\overline{x^2}} - \tilde{\bar{x}} \cdot \widetilde{\overline{xy}}}{\widetilde{\overline{x^2}} - \tilde{\bar{x}}^2}.$$

- 

$$\widetilde{S_0^2} = \frac{n\widetilde{\overline{y^2}} - 2\tilde{\beta}_2 n\tilde{\bar{y}} + n\tilde{\beta}_2^2}{n - r}.$$

$$\widetilde{S^2} = \frac{n\widetilde{\overline{y^2}} - 2\tilde{\beta}_2 n\tilde{\bar{y}} - 2\tilde{\beta}_1 n\widetilde{\overline{xy}} + n\tilde{\beta}_2^2 + 2\tilde{\beta}_1 \tilde{\beta}_2 n\tilde{\bar{x}} + \tilde{\beta}_1^2 n\widetilde{\overline{xy}}}{n - r}.$$

4: $(\tilde{\theta}_0, \tilde{\theta}_1) = (\bot, \bot)$
5: **if** $\min(\widetilde{S_0^2}, (n\widetilde{\overline{x^2}} - n\tilde{\bar{x}}^2)/(n-1)) > 0$ **then**
6: $\quad \tilde{\theta}_0 = (\tilde{\beta}_2, \tilde{\bar{x}}, \widetilde{\overline{x^2}}, \widetilde{S_0^2}, n)$
7: $\quad \tilde{\theta}_1 = (\tilde{\beta}_1, \tilde{\bar{x}}, \widetilde{\overline{x^2}}, \widetilde{S^2}, n)$
8: **end if** $(\tilde{\theta}_0, \tilde{\theta}_1) = (\bot, \bot)$
9: **return** $(\tilde{\theta}_0, \tilde{\theta}_1)$

---

create the test statistic $T(\tilde{\theta}_1) = \frac{\tilde{\beta}_1^2 \cdot n \cdot (\widetilde{\overline{x^2}} - \tilde{\bar{x}}^2)}{\widetilde{S^2}}$ and use $\tilde{\theta}_0 = (\tilde{\bar{y}}, \tilde{\bar{x}}, \widetilde{\overline{x^2}}, \widetilde{S_0^2}, n)$ to simulate the null distributions (to decide to reject or fail to reject the null hypothesis). $P_{\tilde{\theta}_0}$ is instantiated as a normal distribution and used to generate ${}^k x_i$ distributed as $\mathcal{N}(\tilde{\bar{x}}, (n\widetilde{\overline{x^2}} - n\tilde{\bar{x}}^2)/(n-1))$ and ${}^k y_i$ as $\tilde{\beta}_2 + e_i$, $e_i \sim \mathcal{N}(0, \widetilde{S_0^2})$ for all $i \in [n]$.

### C.3 Testing Mixture Models

As we will show experimentally, the best framework for the mixture model test depends on the properties of the dataset. This can be seen as conditional inference [6].

### C.3.1 $F$-Statistic

In the non-private case, we can use the following test statistic for testing mixtures in simple linear regression models:

$$T(X, Y, \hat{\beta}, \hat{\beta}^N, n, r, q) = \left(\frac{n-r}{r-q}\right) \frac{\|X\hat{\beta} - X\hat{\beta}^N\|^2}{\|Y - X\hat{\beta}\|^2}$$

$$= \frac{n_1 \overline{x^2}_1 n_2 \overline{x^2}_2}{S^2 n\overline{x^2}} (\hat{\beta}_1 - \hat{\beta}_2)^2.$$

In Algorithm 5, we apply the Gaussian mechanism to calculate the DP sufficient statistics. $\widetilde{S_0^2}$ and $\widetilde{S^2}$ are DP estimates of the sampling error under the null and alternative hypothesis, respectively. In particular, $\widetilde{S_0^2}$ corresponds to an estimate of the sampling error when the groups have the same distributional properties.

---

**Algorithm 5** $\rho$-zCDP procedure $\texttt{DPStats}_M$

---

1: **Data**: $X \in \mathbb{R}^{n \times 2}, Y \in \mathbb{R}^n$
2: **Input**: integer $n_1, n \geq 2$; $r, q$; $\rho > 0, \Delta > 0$
3: Set $\rho = \rho/8$ and $n_2 = n - n_1$. Then compute the following:

- $\tilde{x}_1 = \frac{1}{n_1}\sum_{i=1}^{n_1} x_i]_{-\Delta}^{\Delta} + \mathcal{N}(0, \frac{2\Delta^2}{\rho n_1^2})$, $\tilde{x}_2 = \frac{1}{n_2}\sum_{i=n_1+1}^{n} x_i]_{-\Delta}^{\Delta} + \mathcal{N}(0, \frac{2\Delta^2}{\rho n_2^2})$, $\tilde{x} = n_1/n \cdot \tilde{x}_1 + n_2/n \cdot \tilde{x}_2$.

- $\widetilde{x^2}_1 = \frac{1}{n_1}\sum_{i=1}^{n_1} x_i^2]_0^{\Delta^2} + \mathcal{N}(0, \frac{\Delta^4}{2\rho n_1^2})$, $\widetilde{x^2}_2 = \frac{1}{n_2}\sum_{i=n_1+1}^{n} x_i^2]_0^{\Delta^2} + \mathcal{N}(0, \frac{\Delta^4}{2\rho n_2^2})$, $\widetilde{x^2} = n_1/n \cdot \widetilde{x^2}_1 + n_2/n \cdot \widetilde{x^2}_2$.

- $\widetilde{xy}_1 = \frac{1}{n_1}\sum_{i=1}^{n_1} x_i y_i]_{-\Delta^2}^{\Delta^2} + \mathcal{N}(0, \frac{2\Delta^4}{\rho n_1^2})$, $\widetilde{xy}_2 = \frac{1}{n_2}\sum_{i=n_1+1}^{n} x_i y_i]_{-\Delta^2}^{\Delta^2} + \mathcal{N}(0, \frac{2\Delta^4}{\rho n_2^2})$, $\widetilde{xy} = n_1/n \cdot \widetilde{xy}_1 + n_2/n \cdot \widetilde{xy}_2$.

- $\widetilde{y^2}_1 = \frac{1}{n_1}\sum_{i=1}^{n_1} y_i^2]_0^{\Delta^2} + \mathcal{N}(0, \frac{\Delta^4}{2\rho n_1^2})$, $\widetilde{y^2}_2 = \frac{1}{n_2}\sum_{i=n_1+1}^{n} y_i^2]_0^{\Delta^2} + \mathcal{N}(0, \frac{\Delta^4}{2\rho n_2^2})$, $\widetilde{y^2} = n_1/n \cdot \widetilde{y^2}_1 + n_2/n \cdot \widetilde{y^2}_2$.

-
$$\tilde{\beta}_1 = \frac{\widetilde{xy}_1}{\widetilde{x^2}_1}, \quad \tilde{\beta}_2 = \frac{\widetilde{xy}_2}{\widetilde{x^2}_2}, \quad \tilde{\beta} = n_1/n \cdot \tilde{\beta}_1 + n_2/n \cdot \tilde{\beta}_2.$$

-
$$\widetilde{S_0^2} = \frac{n\widetilde{y^2} + n\tilde{\beta}^2 - 2n\widetilde{xy}\tilde{\beta}}{n - r}.$$

$$\widetilde{S^2} = \frac{n_1\widetilde{y^2}_1 + n_1\tilde{\beta}_1^2 - 2n_1\widetilde{xy}_1\tilde{\beta}_1 + n_2\widetilde{y^2}_2 + n_2\tilde{\beta}_2^2 - 2n_2\widetilde{xy}_2\tilde{\beta}_2}{n - r}.$$

4: $(\tilde{\theta}_0, \tilde{\theta}_1) = (\perp, \perp)$
5: **if** $\min(\widetilde{S_0^2}, (n\widetilde{x^2} - n\tilde{x}^2)/(n-1)) > 0$ **then**
6: $\quad \tilde{\theta}_0 = (\tilde{\beta}_1, \tilde{x}, \widetilde{x^2}, \widetilde{S_0^2}, n_1, n_2, n)$
7: $\quad \tilde{\theta}_1 = (\tilde{\beta}_1, \tilde{\beta}_2, \widetilde{x^2}_1, \widetilde{x^2}_2, \widetilde{x^2}, \widetilde{S^2}, n_1, n_2, n)$
8: **end if**
9: **return** $(\tilde{\theta}_0, \tilde{\theta}_1)$

---

**Lemma C.2.** *For any $\rho, \Delta > 0$, Algorithm 5 satisfies $\rho$-zCDP.*

*Proof.* This follows from Proposition 1.6 in [12] via the use of the Gaussian Mechanism to ensure zCDP.

The composition and post-processing properties of zCDP (Lemmas 1.7 and 1.8 in [12]) can then be applied. The computation of the following statistics is each done to satisfy $\rho/8$-zCDP: $\tilde{x}_1, \tilde{x}_2, \widetilde{x^2}_1, \widetilde{x^2}_2, \widetilde{xy}_1, \widetilde{xy}_2, \widetilde{y^2}_1, \widetilde{y^2}_2$. The other statistics computed are post-processed DP releases.

As a result, the entire procedure satisfies $\rho$-zCDP.

$\square$

**Instantiating Algorithm 3 for the $F$-statistic Mixture Tester:** If the procedure $\texttt{DPStats}_M$ returns $(\perp, \perp)$, then we fail to reject the null. Otherwise, we use the returned statistics $\tilde{\theta}_1 = (\tilde{\beta}_1, \tilde{\beta}_2, \widetilde{x^2}_1, \widetilde{x^2}_2, \widetilde{x^2}, \widetilde{S^2}, n_1, n_2, n)$ to create the test statistic and use $\tilde{\theta}_0 = (\tilde{\beta}_1, \tilde{x}, \widetilde{x^2}, \widetilde{S_0^2}, n_1, n_2, n)$ to simulate the null distributions. $P_{\tilde{\theta}_0}$ is instantiated as a normal distribution and used to generate $^k x_i$ distributed as $\mathcal{N}(\tilde{x}, (n\widetilde{x^2} - n\tilde{x}_1^2)/(n-1))$ and to generate $^k y_i$ distributed as $\tilde{\beta}_1 {}^k x_i + e_i, e_i \sim \mathcal{N}(0, \widetilde{S_0^2})$ for either group 1 with size $n_1$ or group 2 with size $n - n_1$.

### C.3.2 Nonparametric Tests via Kruskal-Wallis

Couch, Kazan, Shi, Bray, and Groce [21] present DP analogues of nonparametric hypothesis testing methods (which require little or no distributional assumptions). They find that the DP variant of the Kruskal-Wallis test statistic is more powerful than the DP version of the traditional parametric statistics for testing if two groups have the same medians. Here, we reduce our problem of testing mixture models to their problem of testing if groups share the same median. The reduction is as follows: Given two datasets $(\mathbf{x}^1, \mathbf{y}^1)$ and $(\mathbf{x}^2, \mathbf{y}^2)$, each of size $n_1$ and $n_2$ respectively, we wish to test if the slopes are equal. We randomly match all pairs of points in $(\mathbf{x}^1, \mathbf{y}^1)$ and obtain at most $n_1/2$ slopes in $s_1$. We do the same for the second group $(\mathbf{x}^2, \mathbf{y}^2)$ to obtain $n_2/2$ slopes in $s_2$. Then we compute the mean of ranks of elements in $s_1$ and $s_2$ as $\bar{r}_1$ and $\bar{r}_2$ respectively. Next, we compute the Kruskal-Wallis absolute value test statistic $h$ from [21] and release a perturbed version satisfying zCDP. We can use the Monte Carlo testing framework in Algorithm 3 and use Algorithm 6 to compute the test statistics. Under the null, the slopes in $s_1$ and $s_2$ would have similar ranks so we choose uniform random numbers in some interval. We then decide to reject or fail to reject the null, based on the distribution of test statistics obtained via this process and its relation to the statistic computed on the observed data.

**Lemma C.3.** *For any $\rho > 0$ and even $n$, Algorithm 6 satisfies $\rho$-zCDP.*

*Proof.* Algorithm 6 randomly pairs all $n_1$ pairs of points in group 1 (to obtain slopes $s_1$ of size $n_1/2$) and pairs all $n_2$ pairs in group 2 (to obtain slopes $s_2$ of size $n_2/2$). Note that this is a 1-Lipschitz transformation so that the differential privacy guarantees are preserved (by Definition 16 and Lemma 17 of [3]).

Then we proceed to use the DP Kruskal-Wallis absolute value test statistic with sensitivity of 8 (as shown in Theorem 3.4 of [21]). By Proposition 1.6 in [12], via the use of the Gaussian Mechanism, the procedure satisfies $\rho$-zCDP. □

---

**Algorithm 6** $\rho$-zCDP procedure DPKW

---

1: **Data**: $X \in \mathbb{R}^{n \times 2}$; $Y \in \mathbb{R}^n$
2: **Input**: Even $n_1, n \in \mathbb{N}$; $\rho > 0$
3: Let $x_1, \ldots, x_n$ be the observed 1-D independent variables from $X$
4: Let $\tau : [n] \to [n]$ be a randomly chosen permutation
5: $s_1 = \{\}$
6: **for** $i = 1 \ldots n_1/2$ **do**
7:     $s_1 = s_1 \bigcup \{\frac{Y_{\tau(n_1/2+i)} - Y_{\tau(i)}}{x_{\tau(n_1/2+i)} - x_{\tau(i)}}\}$
8: **end for**
9: $n_2 = n - n_1$
10: $s_2 = \{\}$
11: $e = n_1 + n_2/2$
12: **for** $i = n_1 + 1 \ldots e$ **do**
13:     $s_2 = s_2 \bigcup \{\frac{Y_{\tau(e+i)} - Y_{\tau(i)}}{x_{\tau(e+i)} - x_{\tau(i)}}\}$
14: **end for**
15: Let $r : \mathbb{R}^m \to [m]$ be rank-computing function on any $m$ elements
16: Compute $s$ by appending (in an order-preserving manner) $s_2$ to $s_1$
17: Compute $\bar{r}_1$, mean of ranks of $s_1$ in $r(s)$
18: Compute $\bar{r}_2$, mean of ranks of $s_2$ in $r(s)$
19: Compute $h = \frac{4(n-1)}{n^2} \left( n_1 |\bar{r}_1 - \frac{n+1}{2}| + n_2 |\bar{r}_2 - \frac{n+1}{2}| \right)$
20: **return** null, $h + \mathcal{N}(0, 8^2/(2\rho))$

---

**Instantiating Algorithm 3 for the Kruskal-Wallis Mixture Tester:** We use the returned statistic $\tilde{\theta}_1 = (h)$ as the sole statistic. In this case, $T$ is the identity function. $\tilde{\theta}_0$ is taken to be null. $P_{\tilde{\theta}_0}$ generates ${}^k x_i$ and ${}^k y_i$ (for the 2 groups) independently and uniformly at random in a fixed interval (say $[-5, 5]$). Although this distribution may be very different from the actual data distribution, the distribution of ranks of the slopes will be identical to that under the null, ensuring that $T(({}^k x, {}^k y))$ has the right distribution.

# D Differentially Private $F$-Statistic

In this section, we will show that the DP $F$-statistic converges, in distribution, to the asymptotic distribution of the $F$-statistic. The focus will be on showing results for Algorithm 4 but a similar route can be used to obtain analogous results for Algorithm 5. Recall that Algorithm 4 is an instantiation of the DP $F$-statistic for testing a linear relationship while Algorithm 5 is for testing mixtures.

$T_n$ is the non-private $F$-statistic while $\tilde{T}_n$ is the DP $F$-statistic constructed from DP sufficient statistics obtained via Algorithm 4. The main theorem in this section is Theorem D.1, which shows the convergence, in distribution, of $\tilde{T}_n$ to the asymptotic distribution of $T_n$, the chi-squared distribution. As a corollary, the statistical power of $\tilde{T}_n$ converges to the statistical power of $T_n$. While Theorem D.1 is specialized to the simple linear regression setting (i.e., $p = 2$), it can easily be extended to multiple linear regression.

**Theorem D.1.** *Let $\sigma_e > 0$, $r = p = 2, q = 1$, and $\beta \in \mathbb{R}^p$. For every $n \in \mathbb{N}$ with $n > r$, let $X_n \in \mathbb{R}^{n \times p}$ be the design matrix where the first column is an all-ones vector and the second column is $(x_1, \ldots, x_n)^T$. Let $\Delta = \Delta_n > 0$ be a sequence of clipping bounds, $\rho = \rho_n > 0$ be a sequence of privacy parameters, and $\eta_n^2 = \frac{\|X_n\beta - X_n\beta^N\|^2}{\sigma_e^2}$. Under the general linear model (GLM), $Y_n \sim \mathcal{N}(X_n\beta, \sigma_e^2 I_{n \times n})$. Let $\tilde{\beta}$ and $\tilde{\beta}^N$ be the DP least-squares estimate of $\beta$, obtained in Algorithm 4, under the alternative and null hypotheses, respectively. Let $\tilde{T} = \tilde{T}_n$ be the DP $F$-statistic computed from DP sufficient statistics via Algorithm 4 and Equation (7). Suppose the following conditions hold:*

1. *$\exists c_x, c_{x^2} \in \mathbb{R}$ such that $\bar{x} \to c_x$, $\overline{x^2} \to c_{x^2}$, and $c_{x^2} > c_x^2$,*

2. *$\exists \eta \in \mathbb{R}$ such that $\eta_n^2 \to \eta^2$,*

3. *$\frac{\Delta_n^2}{\rho_n n}, \frac{\Delta_n^4}{\rho_n n} \to 0$,*

4. *$\mathbb{P}[\exists i \in [n], y_i \notin [-\Delta_n, \Delta_n]] \to 0$ and $\forall i \in [n], x_i \in [-\Delta_n, \Delta_n]$.*

*Then we obtain the following results:*

1. *Under the null hypothesis: $\tilde{\beta}^N = \tilde{\beta}_n^N \xrightarrow{P} \beta$,*

2. *Under the alternative hypothesis: $\tilde{\beta} = \tilde{\beta}_n \xrightarrow{P} \beta$,*

3. *$\tilde{T} = \tilde{T}_n \xrightarrow{D} \frac{\chi_{r-q}^2(\eta^2)}{r-q}$.*

The condition that $\mathbb{P}[\exists i \in [n], y_i \notin [-\Delta_n, \Delta_n]] \to 0$ (Condition 4 in Theorem D.1), holds by a Gaussian tail bound (Claim D.4), if $\exists k > 0$ such that for all $i \in [n], \Delta_n \geq |\beta_1 x_i + \beta_2| + \sigma_e\sqrt{\log 2n^k}$.

First, we will prove convergence results for sufficient statistics used to construct the non-private $F$-statistic $T_n$, in our setting. Then we will show convergence results for DP sufficient statistics used to construct the private $F$-statistic $\tilde{T}_n$. Finally, we will combine these previous results to show Theorem D.1.

## D.1 Convergence of Non-private Sufficient Statistics

In Equation (7), the non-private $F$-statistic is given as

$$T = T_n = \frac{n-r}{r-q} \cdot \frac{\|X\hat{\beta} - X\hat{\beta}^N\|^2}{\|Y - X\hat{\beta}\|^2} = \frac{n-r}{r-q} \cdot \frac{\|X_n\hat{\beta} - X_n\hat{\beta}^N\|^2}{\|Y_n - X_n\hat{\beta}\|^2}.$$

We start by writing this $F$-statistic, in an equivalent form, in terms of quantities that we will show are convergent:

**Lemma D.2.** *Suppose that $\sigma_e > 0$, $p \in \mathbb{N}$, and $\beta \in \mathbb{R}^p$. Let $X = X_n \in \mathbb{R}^{n \times p}$ be the full-rank design matrix (as in Equation (4)) and $Y = Y_n \sim \mathcal{N}(X_n\beta, \sigma_e^2 I_{n \times n})$. Also, let $\hat{\beta}$ and $\hat{\beta}^N$ be the non-private least-squares estimate of $\beta$ under the alternative and null hypotheses, respectively.*

*Define the following quantities:*

$$\hat{E}_n = \left(\frac{X_n^T X_n}{n}\right)^{1/2} \in \mathbb{R}^{p\times p}, \quad \hat{F}_n = \frac{X_n^T Y_n}{n} \in \mathbb{R}^p, \quad \hat{G}_n = \frac{Y_n^T Y_n}{n} \in \mathbb{R}.$$

*Then the test statistic $T_n$ from Equation (7) can be re-written as*

$$T_n = \frac{n-r}{r-q} \cdot \frac{\|\sqrt{n}\hat{E}_n(\hat{\beta} - \hat{\beta}^N)\|^2}{n(\hat{\beta}^T \hat{E}_n^2 \hat{\beta} - 2\hat{\beta}^T \hat{F}_n + \hat{G}_n)}, \tag{10}$$

*for any $n, r, q \in \mathbb{N}$ such that $q < r$.*

*Proof of Lemma D.2.* First, note that $\hat{E}_n \in \mathbb{R}^{p\times p}$ (i) exists because $X_n^T X_n$ is positive definite, (ii) is unique since its square is positive definite [33].

$$\begin{aligned}
\|X_n\hat{\beta} - X_n\hat{\beta}^N\|^2 &= (\hat{\beta} - \hat{\beta}^N)^T X_n^T X_n(\hat{\beta} - \hat{\beta}^N) \\
&= \sqrt{n}(\hat{\beta} - \hat{\beta}^N)^T \hat{E}_n^T \sqrt{n}\hat{E}_n(\hat{\beta} - \hat{\beta}^N) \\
&= \|\sqrt{n}\hat{E}_n(\hat{\beta} - \hat{\beta}^N)\|^2.
\end{aligned}$$

Next,

$$\begin{aligned}
\|Y_n - X_n\hat{\beta}\|^2 &= (Y_n - X_n\hat{\beta})^T(Y_n - X_n\hat{\beta}) \\
&= Y_n^T Y_n - Y_n^T X_n\hat{\beta} - \hat{\beta}^T X_n^T Y_n + \hat{\beta}^T X_n^T X_n\hat{\beta} \\
&= Y_n^T Y_n + \hat{\beta}^T X_n^T X_n\hat{\beta} - 2\hat{\beta}^T X_n^T Y_n \\
&= n(\hat{\beta}^T \hat{E}_n^2 \hat{\beta} - 2\hat{\beta}^T \hat{F}_n + \hat{G}_n).
\end{aligned}$$

$\square$

It will be easier to use Equation (10) as an equivalent form of the $F$-statistic to prove convergence results. An analogous representation will be used to prove convergence results for the DP $F$-statistic.

In the case of testing a linear relationship (as in Section A.1) in simple linear regression (i.e., where $p = 2$ and the columns of $X$ are the all-ones vector and $(x_1, \ldots, x_n)^T$),

$$X^T X = \begin{pmatrix} n & n\bar{x} \\ n\bar{x} & n\overline{x^2} \end{pmatrix}, \quad X^T Y = \begin{pmatrix} n\bar{y} \\ n\overline{xy} \end{pmatrix}, \quad Y^T Y = \sum_{i=1}^n y_i^2,$$

$$\hat{\beta} = \begin{pmatrix} \hat{\beta}_2 \\ \hat{\beta}_1 \end{pmatrix}, \quad \hat{\beta}^N = \begin{pmatrix} \bar{y} \\ 0 \end{pmatrix},$$

$$\widehat{\sigma_x^2} = \overline{x^2} - \bar{x}^2.$$

In this case, it can be verified that $\hat{E}_n, \hat{F}_n, \hat{G}_n$ is:

$$\begin{aligned}
\hat{E}_n &= \frac{1}{\sqrt{\overline{x^2} + 1 + 2\sqrt{\overline{x^2} - \bar{x}^2}}} \begin{pmatrix} 1 + \sqrt{\overline{x^2} - \bar{x}^2} & \bar{x} \\ \bar{x} & \overline{x^2} + \sqrt{\overline{x^2} - \bar{x}^2} \end{pmatrix}, \\
&= \frac{1}{\sqrt{\overline{x^2} + 1 + 2\sqrt{\widehat{\sigma_x^2}}}} \begin{pmatrix} 1 + \sqrt{\widehat{\sigma_x^2}} & \bar{x} \\ \bar{x} & \overline{x^2} + \sqrt{\widehat{\sigma_x^2}} \end{pmatrix}, \\
\hat{F}_n &= \frac{X^T Y}{n} = \begin{pmatrix} \bar{y} \\ \overline{xy} \end{pmatrix}, \quad \hat{G}_n = \frac{Y^T Y}{n} \overset{\text{def}}{=} \overline{y^2}.
\end{aligned} \tag{11}$$

Next, we proceed to show non-private convergence results that will be pivotal to our final result. We will crucially rely on the Gaussian tail bound, the normality of $\hat{\beta}, \hat{\beta}^N$, and Corollary D.5.

**Lemma D.3.** *For every sequence of clipping bounds $\Delta = \Delta_n > 0$ and sequence of privacy parameters $\rho = \rho_n > 0$, under the conditions of Theorem D.1:*

*(1) $\exists c_y \in \mathbb{R}$ such that $\bar{y} \xrightarrow{P} c_y$,*

*(2) $\exists c_a, c_{xy} \in \mathbb{R}$ such that $\overline{xy} \xrightarrow{P} c_{xy}$, $\overline{xy} - \bar{x} \cdot \bar{y} \xrightarrow{P} c_a$,*

*(3) $\exists c_b \neq 0$ such that $\widehat{\sigma_x^2} \xrightarrow{P} c_b$,*

*(4) $\exists$ unique positive-definite $C^{1/2} \in \mathbb{R}^{2 \times 2}$ such that $\hat{E}_n \to C^{1/2}$,*

*(5) $\hat{F}_n \xrightarrow{P} \begin{pmatrix} c_y & c_{xy} \end{pmatrix}^T$,*

*(6) $\exists c_{y^2} \in \mathbb{R}$ such that $\hat{G}_n = \frac{Y^T Y}{n} \xrightarrow{P} c_{y^2}$,*

*(7) Normality of $\hat{\beta}$: $\hat{\beta} \sim \mathcal{N}\left(\beta, \sigma_e^2 (X_n^T X_n)^{-1}\right)$; Consistency of $\hat{\beta}$: $\hat{\beta} \xrightarrow{P} \beta$.*

To prove Lemma D.3, we will make use of the following tools: the Gaussian tail bound and Slutsky's Theorem, which we state below:

**Claim D.4** (Gaussian Tail Bound)**.** *Let $Z$ be a standard normal random variable with mean 0 and variance 1. i.e., $Z \sim \mathcal{N}(0, 1)$. Then*

$$\mathbb{P}[|Z| > t] \leq 2 \exp(-t^2/2),$$

*for every $t > 0$.*

By the Gaussian tail bound, any Gaussian random variable (such as the DP estimates) converges, in probability, to the asymptotic distributions of the estimates without Gaussian noise added as long as the variance goes to 0 (Corollary D.5):

**Corollary D.5.** *Let $N_n \sim \mathcal{N}(0, \sigma_n^2)$ where $\sigma_n \to 0$, then $N_n \xrightarrow{P} 0$.*

Corollary D.5 follows from the definition of convergence in probability and the Gaussian tail bound (Claim D.4).

Next, we introduce Slutsky's Theorem which will be crucial to combining individual convergence results to show more general results:

**Theorem D.6** (Slutsky's Theorem, see [31])**.** *Let $\{W_n\}, \{Z_n\}$ be a sequence of random vectors and $W$ be a random vector. If $W_n \xrightarrow{D} W$ and $Z_n \xrightarrow{P} c$ for a constant $c \in \mathbb{R}$, then as $n \to \infty$:*

*1. $W_n \cdot Z_n \xrightarrow{D} Wc$,*

*2. $W_n + Z_n \xrightarrow{D} W + c$,*

*3. $W_n / Z_n \xrightarrow{D} W/c$ as long as $c \neq 0$.*

*Proof of Lemma D.3.* (1): By definition, for all $i \in [n]$, $y_i \sim \beta_2 + \beta_1 x_i + \mathcal{N}(0, \sigma_e^2)$. Then, $\bar{y} \sim \beta_2 + \beta_1 \bar{x} + \mathcal{N}(0, \frac{\sigma_e^2}{n})$. By Slutsky's Theorem and Corollary D.5, $\bar{y} \xrightarrow{P} \beta_2 + \beta_1 c_x \overset{\text{def}}{=} c_y$. As a result, $\bar{y} \xrightarrow{P} c_y \in \mathbb{R}$.

(2): Also,

$$\overline{xy} = \frac{1}{n} \sum_{i=1}^n x_i y_i$$

$$\sim \frac{1}{n} \sum_{i=1}^n x_i \mathcal{N}\left(\beta_2 + \beta_1 x_i, \sigma_e^2\right)$$

$$= \frac{1}{n} \sum_{i=1}^n \left(\beta_2 x_i + \beta_1 x_i^2\right) + \mathcal{N}\left(0, \frac{1}{n^2} \sum_{i=1}^n \sigma_e^2 x_i^2\right).$$

From the assumptions of Theorem D.1, $\frac{1}{n}\sum_{i=1}^{n}(\beta_2 x_i + \beta_1 x_i^2) \to \beta_2 c_x + \beta_1 c_{x^2}$ and $\frac{\sigma_e^2}{n^2}\sum_{i=1}^{n} x_i^2 = \frac{\sigma_e^2}{n}\overline{x^2} \to 0$. Then by Slutsky's Theorem and Corollary D.5, $\overline{xy} \xrightarrow{P} \beta_2 c_x + \beta_1 c_{x^2} \overset{\text{def}}{=} c_{xy}$ and $\overline{xy} - \bar{x}\cdot\bar{y} \xrightarrow{P} c_{xy} - c_x c_y \overset{\text{def}}{=} c_a$.

(3): By Slutsky's Theorem and the assumptions in Theorem D.1 that $\overline{x^2} \to c_{x^2}$, $\bar{x} \to c_x$, $c_{x^2} \neq c_x^2$, we have that $\exists c_b \neq 0$ such that $\widehat{\sigma_x^2} \to c_{x^2} - c_x^2 \overset{\text{def}}{=} c_b$.

(4): By Equation (11), we have

$$\hat{E}_n = \frac{1}{\sqrt{\overline{x^2} + 1 + 2\sqrt{\widehat{\sigma_x^2}}}} \begin{pmatrix} 1 + \sqrt{\widehat{\sigma_x^2}} & \bar{x} \\ \bar{x} & \overline{x^2} + \sqrt{\widehat{\sigma_x^2}} \end{pmatrix} \tag{12}$$

$$\to \frac{1}{\sqrt{c_{x^2} + 1 + 2\sqrt{c_{x^2} - c_x^2}}} \begin{pmatrix} 1 + \sqrt{c_{x^2} - c_x^2} & c_x \\ c_x & c_{x^2} + \sqrt{c_{x^2} - c_x^2} \end{pmatrix} \overset{\text{def}}{=} C^{1/2}. \tag{13}$$

(5): Next,

$$\hat{F}_n = \frac{X^T Y}{n} = \begin{pmatrix} \bar{y} \\ \overline{xy} \end{pmatrix} \xrightarrow{P} \begin{pmatrix} c_y \\ c_{xy} \end{pmatrix}.$$

(6): By the weak law of large numbers (Lemma F.1), $\frac{\chi_n^2}{n} \xrightarrow{P} 1$. Then,

$$\begin{aligned}
\hat{G}_n &= \frac{\sum_{i=1}^{n} y_i^2}{n} \\
&\sim \frac{1}{n}\sum_{i=1}^{n}\left(\beta_2 + \beta_1 x_i + \mathcal{N}(0, \sigma_e^2)\right)^2 \\
&= \beta_2^2 + 2\bar{x}\beta_1\beta_2 + 2\beta_2 \mathcal{N}\left(0, \frac{\sigma_e^2}{n}\right) + \beta_1^2 \overline{x^2} + 2\beta_1\bar{x}\mathcal{N}\left(0, \frac{\sigma_e^2}{n}\right) + \sigma_e^2 \frac{\chi_n^2}{n} \\
&\xrightarrow{P} \beta_2^2 + 2c_x\beta_1\beta_2 + \beta_1^2 c_{x^2} + \sigma_e^2 \\
&\overset{\text{def}}{=} c_{y^2},
\end{aligned}$$

via the use of Slutsky's Theorem, Corollary D.5, and assumptions that $\bar{x} \to c_x$, $\overline{x^2} \to c_{x^2}$.

(7): First, we recall that $\hat{\beta}$, the non-private OLS estimate, is Gaussian and centered at $\beta$:

$$\hat{\beta} \sim \mathcal{N}\left(\beta, \sigma_e^2 (X^T X)^{-1}\right) = \mathcal{N}\left(\beta, \sigma_e^2 \hat{E}_n^{-2}/n\right),$$

This follows from Equations (3.9) and (3.10) in [32] for any design matrix $X \in \mathbb{R}^{n\times 2}$.

Since $\hat{E}_n \to C^{1/2}$, it follows that $\hat{E}_n^{-2}/n \to 0$ so that by Corollary D.5, $\hat{\beta} \xrightarrow{P} \beta$.

$\square$

Now, we will show that the DP statistics converge, either in probability or distribution, to the distributions of their corresponding non-DP statistics.

## D.2 Convergence of Differentially Private Sufficient Statistics

The DP $F$-statistic is constructed via Algorithm 4 and Equation (7). We start by rewriting the DP $F$-statistic analogously to Lemma D.2:

**Lemma D.7.** *Suppose that $\sigma_e > 0$, $p \in \mathbb{N}$, and $\beta \in \mathbb{R}^p$. Let $X = X_n \in \mathbb{R}^{n\times p}$ be the full-rank design matrix (as in Equation (4)) and $Y = Y_n \sim \mathcal{N}(X_n\beta, \sigma_e^2 I_{n\times n})$. Also, let $\tilde{\bar{x}}, \tilde{\bar{y}}, \widetilde{\overline{x^2}}, \widetilde{\overline{xy}}, \widetilde{\overline{y^2}}, \tilde{\beta}_1$, and $\tilde{\beta}_2$ be as computed in Algorithm 4.*

*Define the following quantities:*

$$\widetilde{\sigma_x^2} \overset{\text{def}}{=} \widetilde{\widetilde{x^2}} - \tilde{x}^2, \tag{14}$$

$$\tilde{E}_n = \left(\frac{\widetilde{X^T X}}{n}\right)^{1/2} = \frac{1}{\sqrt{\widetilde{\widetilde{x^2}} + 1 + 2\sqrt{\widetilde{\sigma_x^2}}}} \begin{pmatrix} 1 + \sqrt{\widetilde{\sigma_x^2}} & \tilde{x} \\ \tilde{x} & \widetilde{\widetilde{x^2}} + \sqrt{\widetilde{\sigma_x^2}} \end{pmatrix},$$

$$\tilde{F}_n = \frac{\widetilde{X^T Y}}{n} = \begin{pmatrix} \tilde{\tilde{y}} \\ \widetilde{xy} \end{pmatrix}, \quad \tilde{G}_n = \widetilde{\widetilde{y^2}}, \tag{15}$$

$$\tilde{\beta} = \begin{pmatrix} \tilde{\beta}_2 \\ \tilde{\beta}_1 \end{pmatrix}, \quad \tilde{\beta}^N = \begin{pmatrix} \tilde{\tilde{y}} \\ 0 \end{pmatrix}. \tag{16}$$

*where we take $\sqrt{\widetilde{\sigma_x^2}}$ to be the square root of $\widetilde{\sigma_x^2}$ with non-negative real and imaginary parts.*

*Furthermore, if $\tilde{T}_n = T(\tilde{\theta}_1)$ is the test statistic obtained via statistics computed in Algorithm 4 and via Equation (7), then $\tilde{T}_n$ can be re-written as*

$$\tilde{T} = \tilde{T}_n = \frac{n-r}{r-q} \cdot \frac{\|\sqrt{n}\tilde{E}_n(\tilde{\beta} - \tilde{\beta}^N)\|^2}{n(\tilde{\beta}^T \tilde{E}_n^2 \tilde{\beta} - 2\tilde{\beta}^T \tilde{F}_n + \tilde{G}_n)}, \tag{17}$$

*for any $n, r, q \in \mathbb{N}$ such that $q < r$.*

*Proof of Lemma D.7.*

$$(\tilde{\beta} - \tilde{\beta}^N)^T \widetilde{X_n^T X_n}(\tilde{\beta} - \tilde{\beta}^N) = \sqrt{n}(\tilde{\beta} - \tilde{\beta}^N)^T \tilde{E}_n^T \sqrt{n}\tilde{E}_n(\tilde{\beta} - \tilde{\beta}^N)$$
$$= \|\sqrt{n}\tilde{E}_n(\tilde{\beta} - \tilde{\beta}^N)\|^2.$$

Next,

$$\widetilde{Y_n^T Y_n} - 2\tilde{\beta}^T \widetilde{X_n^T Y_n} + \tilde{\beta}^T \widetilde{X_n^T X_n}\tilde{\beta} = n\tilde{G}_n - 2n\tilde{\beta}^T \tilde{F}_n + n\tilde{\beta}^T \hat{E}_n^2 \tilde{\beta}$$
$$= n(\tilde{\beta}^T \tilde{E}_n^2 \tilde{\beta} - 2\tilde{\beta}^T \tilde{F}_n + \tilde{G}_n).$$

$\square$

We now introduce two helper lemmas that are useful for showing later results. The first uses a hybrid-type argument to show a $1/f(n)$ rate of convergence of a ratio of random variables. The second can be used to show that if the difference of two random variables converge to 0, then as long as they converge to a non-zero constant, the difference of their square root converge to 0.

**Lemma D.8.** *Let $A_n, B_n, \widetilde{A}_n, \widetilde{B}_n$ be random variables such that:*

1. *For constants $c_1, c_2 \in \mathbb{R}, c_2 \neq 0$:* $A_n \xrightarrow{P} c_1, B_n \xrightarrow{P} c_2,$

2. *For function $f(n)$:* $f(n)(\widetilde{A}_n - A_n) \xrightarrow{P} 0$ *and* $f(n)(\widetilde{B}_n - B_n) \xrightarrow{P} 0.$

*Then:*

$$f(n)\left(\frac{\widetilde{A}_n}{\widetilde{B}_n} - \frac{A_n}{B_n}\right) \xrightarrow{P} 0.$$

*Proof of Lemma D.8.* We use a hybrid-type argument. We write

$$f(n)\left(\frac{\widetilde{A}_n}{\widetilde{B}_n} - \frac{A_n}{B_n}\right) = f(n)\left(\frac{\widetilde{A}_n}{\widetilde{B}_n} - \frac{\widetilde{A}_n}{B_n} + \frac{\widetilde{A}_n}{B_n} - \frac{A_n}{B_n}\right).$$

Then,

$$f(n)\left(\frac{\widetilde{A}_n}{\widetilde{B}_n} - \frac{\widetilde{A}_n}{B_n}\right) = f(n)\left(\frac{\widetilde{A}_n B_n - \widetilde{A}_n \widetilde{B}_n}{\widetilde{B}_n B_n}\right)$$

$$= \widetilde{A}_n f(n)\left(\frac{B_n - \widetilde{B}_n}{\widetilde{B}_n B_n}\right)$$

$$\xrightarrow{P} 0,$$

since $\widetilde{A}_n \xrightarrow{P} c_1$, $f(n)(B_n - \widetilde{B}_n) \xrightarrow{P} 0$, and by Slutsky's Theorem $B_n \widetilde{B}_n \xrightarrow{P} c_2^2 \neq 0$.

Also,

$$f(n)\left(\frac{\widetilde{A}_n}{B_n} - \frac{A_n}{B_n}\right) = f(n)\left(\frac{\widetilde{A}_n - A_n}{B_n}\right) \xrightarrow{P} 0,$$

since $f(n)(\widetilde{A}_n - A_n) \xrightarrow{P} 0$, $B_n \xrightarrow{P} c_2 \neq 0$ so that the result follows by a routine application of Slutsky's Theorem.

As a result, $f(n)\left(\frac{\widetilde{A}_n}{\widetilde{B}_n} - \frac{A_n}{B_n}\right) \xrightarrow{P} 0$.

$\square$

**Lemma D.9.** *Let $A_n, \widetilde{A}_n$ be random variables such that:*

1. *For constant $c \in \mathbb{R}$, $c \neq 0$:* $A_n \xrightarrow{P} c$,

2. *For function $f(n)$:* $f(n)(\widetilde{A}_n - A_n) \xrightarrow{P} 0$.

*Then:*

$$f(n)(\widetilde{A}_n^{1/2} - A_n^{1/2}) \xrightarrow{P} 0.$$

*Proof of Lemma D.9.* Throughout, we take square roots in which both the real and imaginary parts are non-negative.

Recall that by difference of two squares:

$$a^{1/2} - b^{1/2} = \frac{a - b}{a^{1/2} + b^{1/2}},$$

for any $a, b \in \mathbb{C}$.

Then by Slutsky's Theorem:

$$f(n)(\widetilde{A}_n^{1/2} - A_n^{1/2}) = \frac{f(n)(\widetilde{A}_n - A_n)}{\widetilde{A}_n^{1/2} + A_n^{1/2}}$$

$$\xrightarrow{P} 0,$$

where $\widetilde{A}_n^{1/2}, A_n^{1/2} \xrightarrow{P} c^{1/2}$.

$\square$

We will show that the DP regression coefficients converge to the true coefficients. i.e., $\tilde{\beta} \xrightarrow{P} \beta$. But we begin with showing convergence of the constituent DP sufficient statistics.

**Lemma D.10.** *For every sequence of clipping bounds $\Delta = \Delta_n > 0$ and sequence of privacy parameters $\rho = \rho_n > 0$, in Algorithm 4, under the conditions of Theorem D.1:*

(1) $\sqrt{n}|\tilde{\bar{x}} - \bar{x}| \xrightarrow{P} 0$, $\tilde{\bar{x}} \xrightarrow{P} c_x$,

(2) $\sqrt{n}|\tilde{\bar{y}} - \bar{y}| \xrightarrow{P} 0$, $\tilde{\bar{y}} \xrightarrow{P} c_y$,

(3) $\sqrt{n}|\widetilde{\overline{x^2}} - \overline{x^2}| \xrightarrow{P} 0$, $\widetilde{\overline{x^2}} \xrightarrow{P} c_{x^2}$,

(4) $\sqrt{n}|\widetilde{\overline{xy}} - \overline{xy}| \xrightarrow{P} 0$, $\widetilde{\overline{xy}} \xrightarrow{P} c_{xy}$,

(5) $\sqrt{n}|\tilde{\bar{x}}^2 - \bar{x}^2| \xrightarrow{P} 0$,

(6) $\sqrt{n}|\tilde{\bar{x}}\tilde{\bar{y}} - \bar{x} \cdot \bar{y}| \xrightarrow{P} 0$,

(7) $\exists c_a \in \mathbb{R}$, $\widetilde{\overline{xy}} - \tilde{\bar{x}}\tilde{\bar{y}} \xrightarrow{P} c_a$,

(8) $\exists c_b \neq 0$, $\widetilde{\sigma_x^2} \xrightarrow{P} c_b$,

(9) $\sqrt{n}(\widetilde{\sigma_x^2} - \widehat{\sigma_x^2}) \xrightarrow{P} 0$,

(10) $\exists C^{1/2} \in \mathbb{R}^{2 \times 2}$ such that $\sqrt{n}(\tilde{E}_n - \hat{E}_n) \xrightarrow{P} 0$, $\tilde{E}_n \xrightarrow{P} C^{1/2}$,

(11) $\tilde{F}_n \xrightarrow{P} (c_y \quad c_{xy})^T$,

(12) $\exists c_{y^2} \in \mathbb{R}$ such that $\tilde{G}_n \xrightarrow{P} c_{y^2}$,

*where the constant scalars and matrix $c_x, c_y, c_{x^2}, c_{xy}, c_{y^2}, c_a, c_b, C^{1/2}$ are the same as the ones defined in Lemma D.3.*

*Proof of Lemma D.10.* Define

1. $\breve{x} = \frac{1}{n} \sum_{i=1}^n x_i|_{-\Delta_n}^{\Delta_n}$,

2. $\breve{y} = \frac{1}{n} \sum_{i=1}^n y_i|_{-\Delta_n}^{\Delta_n}$,

3. $\overset{\smile}{\overline{x^2}} = \frac{1}{n} \sum_{i=1}^n x_i^2|_0^{\Delta_n^2}$,

4. $\overset{\smile}{\overline{xy}} = \frac{1}{n} \sum_{i=1}^n x_i y_i|_{-\Delta_n^2}^{\Delta_n^2}$, and

5. $\overset{\smile}{\overline{y^2}} = \sum_{i=1}^n y_i^2|_0^{\Delta_n^2}$.

Then $\tilde{\bar{x}} = \breve{x} + N_1$, $\tilde{\bar{y}} = \breve{y} + N_2$, $\widetilde{\overline{x^2}} = \overset{\smile}{\overline{x^2}} + N_3$, $\widetilde{\overline{xy}} = \overset{\smile}{\overline{xy}} + N_4$, $\widetilde{\overline{y^2}} = \overset{\smile}{\overline{y^2}} + N_5$ where $N_1, N_2 \sim \mathcal{N}(0, \frac{2\Delta^2}{\rho n^2})$, $N_3 \sim \mathcal{N}(0, \frac{\Delta^4}{2\rho n^2})$, $N_4 \sim \mathcal{N}(0, \frac{2\Delta^4}{\rho n^2})$, and $N_5 \sim \mathcal{N}(0, \frac{\Delta^4}{2\rho})$.

By conditions of Theorem D.1, $\frac{\Delta_n^2}{\rho_n n} \to 0$, $\frac{\Delta_n^4}{\rho_n n} \to 0$ so that by Corollary D.5, $\sqrt{n}|N_1| \xrightarrow{P} 0$, $\sqrt{n}|N_2| \xrightarrow{P} 0$, $\sqrt{n}|N_3| \xrightarrow{P} 0$, $\sqrt{n}|N_4| \xrightarrow{P} 0$, and $\frac{\sqrt{n}}{n}|N_5| \xrightarrow{P} 0$ since $\sqrt{n}\mathcal{N}(0, \frac{2\Delta^2}{\rho n^2}) \sim \mathcal{N}(0, \frac{2\Delta^2}{\rho n})$, $\sqrt{n}\mathcal{N}(0, \frac{\Delta^4}{2\rho n^2}) \sim \mathcal{N}(0, \frac{\Delta^4}{2\rho n})$, and $\frac{\sqrt{n}}{n}\mathcal{N}(0, \frac{2\Delta^4}{\rho}) \sim \mathcal{N}(0, \frac{2\Delta^4}{\rho n})$.

(1): By assumption, for all $i \in [n]$, $x_i \in [-\Delta_n, \Delta_n]$. Thus, $\bar{x} = \breve{x}$ so that $\tilde{\bar{x}} = \bar{x} + N_1$. Then,

$$\sqrt{n}|\tilde{\bar{x}} - \bar{x}| \leq \sqrt{n}|N_1| + \sqrt{n}|\breve{x} - \bar{x}| \xrightarrow{P} 0$$

by Slutsky's Theorem. As a corollary, $\tilde{\bar{x}} \xrightarrow{P} c_x$ by Lemma B.4 and the assumption in Theorem D.1 that $\bar{x} \to c_x$.

(2): The proof that $\sqrt{n}|\bar{y} - \tilde{\bar{y}}| \xrightarrow{P} 0$ is very similar: observe that by the assumptions of Theorem D.1:

$$\mathbb{P}[|\bar{y} - \breve{y}| > 0] \leq \mathbb{P}[\exists i \in [n], y_i \notin [-\Delta_n, \Delta_n]] \tag{18}$$
$$\to 0. \tag{19}$$

Thus, $\sqrt{n}|\breve{y} - \bar{y}| \xrightarrow{P} 0$. Combining with $\sqrt{n}|N_2| \xrightarrow{P} 0$, by the triangle inequality, $\sqrt{n}|\bar{y} - \tilde{y}| \xrightarrow{P} 0$. As a corollary, $\tilde{y} \xrightarrow{P} c_y$ by Lemma B.4 and the assumption in Theorem D.1 that $\bar{x} \to c_x$.

(3): To show $\sqrt{n}|\widetilde{x^2} - \overline{x^2}| \xrightarrow{P} 0$, we proceed in an analogous way: using the assumption that $\mathbb{P}[\exists i \in [n], x_i \notin [-\Delta_n, \Delta_n]] = 0$, we obtain that $\mathbb{P}[\exists i \in [n], x_i^2 \notin [0, \Delta_n^2]] = 0$ so that $\sqrt{n}|\overline{x^2} - \overset{\smile}{\overline{x^2}}| \xrightarrow{P} 0$. Combining with $\sqrt{n}|N_3| \xrightarrow{P} 0$, by the triangle inequality, $\sqrt{n}|\widetilde{x^2} - \overline{x^2}| \xrightarrow{P} 0$. As a corollary, $\widetilde{x^2} \xrightarrow{P} c_{x^2}$ by Lemma B.4 and the assumption in Theorem D.1 that $\overline{x^2} \to c_{x^2}$.

(4): In a similar fashion, $\sqrt{n}|\widetilde{xy} - \overline{xy}| \xrightarrow{P} 0$: using the assumptions

$$\mathbb{P}[\exists i \in [n], x_i \notin [-\Delta_n, \Delta_n]] = 0, \quad \mathbb{P}[\exists i \in [n], y_i \notin [-\Delta_n, \Delta_n]] \to 0,$$

we have that

$$\begin{aligned}
&\mathbb{P}[\exists i \in [n], x_i y_i \notin [-\Delta_n^2, \Delta_n^2]] \\
&\leq \mathbb{P}[\exists i \in [n], x_i \notin [-\Delta_n, \Delta_n]] + \mathbb{P}[\exists i \in [n], y_i \notin [-\Delta_n, \Delta_n]] \\
&\to 0,
\end{aligned}$$

so that $\sqrt{n}|\overline{xy} - \overset{\smile}{\overline{xy}}| \xrightarrow{P} 0$. Combining with $\sqrt{n}|N_4| \xrightarrow{P} 0$, by the triangle inequality, $\sqrt{n}|\widetilde{xy} - \overline{xy}| \xrightarrow{P} 0$. By Lemma D.3, $\overline{xy} \xrightarrow{P} c_{xy}$. Then by Lemma B.4, $\widetilde{xy} \xrightarrow{P} c_{xy}$.

(5): Next we show $\sqrt{n}|\tilde{x}^2 - \bar{x}^2| \xrightarrow{P} 0$: $\sqrt{n}(\tilde{x}^2 - \bar{x}^2) = \sqrt{n}(\tilde{x} - \bar{x})(\tilde{x} + \bar{x})$. Since, $\tilde{x}, \bar{x} \xrightarrow{P} c_x$, we have $(\tilde{x} + \bar{x}) \xrightarrow{P} 2c_x$, $(\tilde{x} - \bar{x}) \xrightarrow{P} 0$ so that by Slutsky's Theorem, $\sqrt{n}|\tilde{x}^2 - \bar{x}^2| \xrightarrow{P} 0$

(6): In a similar fashion, $\sqrt{n}|\tilde{x}\tilde{y} - \bar{x}\cdot\bar{y}| \xrightarrow{P} 0$: $\tilde{x} = \breve{x} + N_1 = \bar{x} + N_1$, since $\forall i \in [n], x_i \in [-\Delta_n, \Delta_n]$. Then,

$$\begin{aligned}
\sqrt{n}(\tilde{x}\tilde{y} - \bar{x}\cdot\bar{y}) &= \sqrt{n}\left[(\bar{x} + N_1)\tilde{y} - \bar{x}\cdot\bar{y}\right] \\
&= \sqrt{n}N_1\tilde{y} + \bar{x}\sqrt{n}(\tilde{y} - \bar{y}) \\
&\xrightarrow{P} 0,
\end{aligned}$$

since $\sqrt{n}(\tilde{y} - \bar{y}) \xrightarrow{P} 0$, $\tilde{y} \xrightarrow{P} c_y$, $\sqrt{n}N_1 \xrightarrow{P} 0$.

(7): Next, we show that $\widetilde{xy} - \tilde{x}\tilde{y} \xrightarrow{P} c_a \in \mathbb{R}$: Follows by Slutsky's Theorem since $\widetilde{xy} \xrightarrow{P} c_{xy}$, $\tilde{x} \xrightarrow{P} c_x$, and $\tilde{y} \xrightarrow{P} c_y$.

(8): In a similar fashion, $\widetilde{\sigma_x^2} \xrightarrow{P} c_b$: This follows by Slutsky's Theorem since $\widetilde{x^2} \xrightarrow{P} c_{x^2}$, $\tilde{x} \xrightarrow{P} c_x$.

(9) $\sqrt{n}(\widetilde{\sigma_x^2} - \widehat{\sigma_x^2}) \xrightarrow{P} 0$ follows from parts (3) and (5).

(10): By Lemma D.9,

$$\sqrt{n}\left(\sqrt{\widetilde{\sigma_x^2}} - \sqrt{\widehat{\sigma_x^2}}\right) \xrightarrow{P} 0,$$

since $\sqrt{n}\left(\widetilde{\sigma_x^2} - \widehat{\sigma_x^2}\right) \xrightarrow{P} 0$ and $\widehat{\sigma_x^2} \xrightarrow{P} c_b \neq 0$ by Lemma D.3.

We have already established that $\sqrt{n}(\widetilde{x^2} - \overline{x^2}) \xrightarrow{P} 0$ and $\sqrt{n}\left(\sqrt{\widetilde{\sigma_x^2}} - \sqrt{\widehat{\sigma_x^2}}\right) \xrightarrow{P} 0$.

As a result, by Lemma D.9,

$$\sqrt{n}\left(\sqrt{\widetilde{x^2} + 1 + 2\sqrt{\widetilde{\sigma_x^2}}} - \sqrt{\overline{x^2} + 1 + 2\sqrt{\widehat{\sigma_x^2}}}\right) \xrightarrow{P} 0.$$

Then since $\hat{E}_n, \tilde{E}_n$ converge to constant matrices and $\sqrt{n}(\tilde{x} - \bar{x}) \xrightarrow{P} 0$, $\sqrt{n}(\widetilde{x^2} - \overline{x^2}) \xrightarrow{P} 0$, $\sqrt{n}\left(\sqrt{\widetilde{\sigma_x^2}} - \sqrt{\widehat{\sigma_x^2}}\right) \xrightarrow{P} 0$, it follows by Lemma D.8 that $\sqrt{n}(\tilde{E}_n - \hat{E}_n) \xrightarrow{P} 0$.

As a corollary, $\tilde{E}_n \xrightarrow{P} C^{1/2}$ since $\hat{E}_n \xrightarrow{P} C^{1/2}$ by Lemma D.3.

(11): Also, $\tilde{F}_n \xrightarrow{P} (c_y \quad c_{xy})^T$ since $\tilde{\bar{y}} \xrightarrow{P} c_y$ and $\widetilde{\overline{xy}} \xrightarrow{P} c_{xy}$.

(12): Finally, we show that $\tilde{G}_n \xrightarrow{P} c_{y^2} \in \mathbb{R}$: using the assumption that $\mathbb{P}[\exists i \in [n], y_i \notin [-\Delta_n, \Delta_n]] \to 0$, we can obtain that $\mathbb{P}[\exists i \in [n], y_i^2 \notin [0, \Delta_n^2]] \to 0$ so that $\sqrt{n}|\overline{y^2} - \widetilde{y^2}| \xrightarrow{P} 0$. Combining with $\frac{\sqrt{n}}{n}|N_5| \xrightarrow{P} 0$, by the triangle inequality, $\sqrt{n}|\overline{y^2} - \widetilde{y^2}| \xrightarrow{P} 0$ which implies that $\sqrt{n}|\tilde{G}_n - \hat{G}_n| \xrightarrow{P} 0$. Then by Lemma D.3 and Lemma B.4, $\tilde{G}_n \xrightarrow{P} c_{y^2}$.

$\square$

Lemma D.10 shows that the noise added to the non-DP estimates converges, in probability, to 0 and that the DP estimates of the regression parameters converge, in probability, to the true parameters. Next, we will show the $1/\sqrt{n}$ convergence rates of $\tilde{\beta}^N, \tilde{\beta}$. As a corollary, this implies the consistency of $\tilde{\beta}^N, \tilde{\beta}$.

**Lemma D.11.** *For every sequence of clipping bounds $\Delta = \Delta_n > 0$ and sequence of privacy parameters $\rho = \rho_n > 0$, in Algorithm 4, under the conditions of Theorem D.1:*

1. *$\sqrt{n}(\tilde{\beta}^N - \hat{\beta}^N) \xrightarrow{P} 0$,*

2. *$\sqrt{n}(\tilde{\beta} - \hat{\beta}) \xrightarrow{P} 0$.*

*Proof of Lemma D.11.* As previously defined,

$$\tilde{\beta}^N = \begin{pmatrix} \tilde{\bar{y}} \\ 0 \end{pmatrix}, \quad \hat{\beta}^N = \begin{pmatrix} \bar{y} \\ 0 \end{pmatrix}.$$

Then, $\sqrt{n}(\tilde{\beta}^N - \hat{\beta}^N) \xrightarrow{P} 0$ by Lemma D.10 since $\sqrt{n}|\tilde{\bar{y}} - \breve{y}| \xrightarrow{P} 0$.

We will show that $\sqrt{n}(\tilde{\beta} - \hat{\beta}) \xrightarrow{P} 0$. First, to show that $\sqrt{n}(\tilde{\beta}_1 - \hat{\beta}_1) \xrightarrow{P} 0$, we apply Lemma D.8 with $\tilde{A} = \widetilde{\overline{xy}} - \tilde{\bar{x}}\tilde{\bar{y}}$, $A = \overline{xy} - \bar{x} \cdot \bar{y}$, $\tilde{B} = \widetilde{\overline{x^2}} - \tilde{\bar{x}}^2 = \widetilde{\sigma_x^2}$, $B = \overline{x^2} - \bar{x}^2 = \widehat{\sigma_x^2}$ and $f(n) = \sqrt{n}$. Then $\tilde{\beta}_1 = \frac{\tilde{A}}{\tilde{B}}$ and $\hat{\beta}_1 = \frac{A}{B}$. By Lemma D.3, $B = \widehat{\sigma_x^2}$ converges, in probability, to a non-zero constant and $\sqrt{n}(\widetilde{\overline{xy}} - \overline{xy})$, $\sqrt{n}(\bar{x} \cdot \bar{y} - \tilde{\bar{x}}\tilde{\bar{y}}) \xrightarrow{P} 0$ by Lemma D.10. Also, by Lemma D.10 and Lemma D.3, if we define $\tilde{B} = \widetilde{\sigma_x^2}, B = \widehat{\sigma_x^2}, \tilde{A} = \widetilde{\overline{xy}} - \tilde{\bar{x}}\tilde{\bar{y}}$, then $\sqrt{n}(\tilde{B} - B) \xrightarrow{P} 0$ and $\sqrt{n}(\tilde{A} - A) \xrightarrow{P} 0$. Then by Lemma D.8, $\sqrt{n}(\tilde{\beta}_1 - \hat{\beta}_1) \xrightarrow{P} 0$. By similar arguments, $\sqrt{n}(\hat{\beta}_2 - \tilde{\beta}_2) \xrightarrow{P} 0$ so that $\sqrt{n}(\hat{\beta} - \tilde{\beta}) \xrightarrow{P} 0$.

$\square$

Lemma D.11 leads to the following corollary, showing consistency of the DP estimates of $\beta$ under the null or alternative hypothesis.

**Corollary D.12.** *For every sequence of clipping bounds $\Delta = \Delta_n > 0$ and sequence of privacy parameters $\rho = \rho_n > 0$, in Algorithm 4, under the conditions of Theorem D.1:*

1. *Under the null hypothesis: $\tilde{\beta}^N \xrightarrow{P} \beta$,*

2. *Under the alternative hypothesis: $\tilde{\beta} \xrightarrow{P} \beta$.*

*Proof of Corollary D.12.* By Lemma D.3, $\hat{\beta}_1 \xrightarrow{P} \beta_1$ and $\hat{\beta}_2 \xrightarrow{P} \beta_2$. Then, using Lemma D.11 and Lemma B.4: $\tilde{\beta}_1 \xrightarrow{P} \beta_1, \tilde{\beta}_2 \xrightarrow{P} \beta_2$.

Also, by Lemma D.3, $\bar{y} \xrightarrow{P} c_y$ so that under the null hypothesis, $\tilde{\beta}^N \xrightarrow{P} \beta$.

$\square$

### D.3 Convergence of Differentially Private $F$-Statistic

We now introduce the continuous mapping theorem, which is especially useful for combining individual convergence results to show, under certain conditions, more complex convergence results. The continuous mapping theorem can be used to map convergent sequences into another convergent sequence via a continuous function.

**Theorem D.13** (Continuous Mapping Theorem, see [31])**.** *Let $\{W_n\}$ be a sequence of random vectors and $W$ be a random vector taking values in the same metric space $\mathcal{X}$. Let $\mathcal{Y}$ be a metric space and $g : \mathcal{X} \to \mathcal{Y}$ be a measurable function.*

*Define $D_g = \{x \; : \; g$ is discontinuous at $x\}$. Suppose that $\mathbb{P}[W \in D_g] = 0$. Then:*

1. *$W_n \xrightarrow{P} W \Rightarrow g(W_n) \xrightarrow{P} g(W)$,*

2. *$W_n \xrightarrow{D} W \Rightarrow g(W_n) \xrightarrow{D} g(W)$,*

3. *$W_n \xrightarrow{a.s.} W \Rightarrow g(W_n) \xrightarrow{a.s.} g(W)$.*

We now state and prove a helper lemma that will be useful for showing that the numerators and denominators of the DP $F$-statistic converge to the right distribution.

**Lemma D.14.** *Let $A_n, B_n$ be random vectors such that there exists distribution $L$ where:*

1. *$A_n - B_n \xrightarrow{P} 0$,*

2. *$\|B_n\| \xrightarrow{D} L$ such that $\mathbb{P}[\|B_n\| = 0] = 0$.*

*Then,*

$$\|A_n\|^2 \xrightarrow{D} L^2.$$

*Proof of Lemma D.14.* Consider the unit vector $\frac{B_n}{\|B_n\|}$. Since $A_n - B_n \xrightarrow{P} 0$, we have that by definition of convergence in probability:

$$\frac{B_n}{\|B_n\|}(A_n - B_n) \xrightarrow{P} 0,$$

where $\|B_n\|$ is almost surely never 0.

First, let $W_n = (\|B_n\|, \|B_n\|)$. Then since $\|B_n\| \xrightarrow{D} L$, by the continuous mapping theorem (Theorem D.13), if $g((x, y)) = x \cdot y$, then $g(W_n) = \|B_n\|^2 \xrightarrow{D} L^2$.

Then, let $W_n = (\|B_n\|, \frac{B_n}{\|B_n\|}(A_n - B_n))$. Then since $\mathbb{P}[\|B_n\| = 0] = 0$, by the continuous mapping theorem (Theorem D.13), if $g((x, y)) = x \cdot y$, then $g(W_n) = B_n \cdot (A_n - B_n) \xrightarrow{D} 0$ which implies that

$$\langle A_n, B_n \rangle - \|B_n\|^2 = B_n \cdot (A_n - B_n) \xrightarrow{P} 0,$$

so that

$$2(\langle A_n, B_n \rangle - \|B_n\|^2) \xrightarrow{P} 0. \tag{20}$$

Also, by the continuous mapping theorem,

$$\|A_n\|^2 - 2\langle A_n, B_n \rangle + \|B_n\|^2 \tag{21}$$

$$= \|A_n - B_n\|^2 \tag{22}$$

$$\xrightarrow{P} 0. \tag{23}$$

Adding Equations (20) and (21) results in the following: $\|A_n\|^2 - \|B_n\|^2 \xrightarrow{P} 0$. Then by Lemma B.4, since $\|B_n\|^2 \xrightarrow{D} L^2$, we have that $\|A_n\|^2 \xrightarrow{D} L^2$.

$\square$

We now show that the main terms in the numerators and denominators of the DP $F$-statistic converge to the asymptotic distribution of their non-private counterparts.

**Lemma D.15.** *Let $\sigma_e > 0$, $r = p = 2, q = 1$, and $\beta \in \mathbb{R}^p$. For every $n \in \mathbb{N}$ with $n > r$, let $X_n \in \mathbb{R}^{n \times p}$ be the design matrix. For every sequence of clipping bounds $\Delta = \Delta_n > 0$ and sequence of privacy parameters $\rho = \rho_n > 0$, in Algorithm 4, under the conditions of Theorem D.1:*

$$\frac{n(\tilde{\beta}^T \tilde{E}_n^2 \tilde{\beta} - 2\tilde{\beta}^T \tilde{F}_n + \tilde{G}_n)}{n-r} \xrightarrow{P} \sigma_e^2, \quad \|\sqrt{n}\tilde{E}_n(\tilde{\beta} - \tilde{\beta}^N)\|^2 \xrightarrow{D} \mathcal{X}_{r-q}^2(\eta^2)\sigma_e^2.$$

*Proof of Lemma D.15.* By Lemma D.3 and Lemma D.10:

1. $\tilde{\beta}, \hat{\beta} \xrightarrow{P} \beta$,

2. $\tilde{E}_n, \hat{E}_n \to C^{1/2}$,

3. $\tilde{F}_n, \hat{F}_n \xrightarrow{P} (c_y \quad c_{xy})^T$,

4. $\tilde{G}_n, \hat{G}_n \xrightarrow{P} c_{y^2}$.

Furthermore,

$$\frac{n}{n-r} = \frac{1}{1 - r/n} \to 1.$$

As a result, by Slutsky's Theorem,

$$\left( \frac{n(\tilde{\beta}^T \tilde{E}_n^2 \tilde{\beta} - 2\tilde{\beta}^T \tilde{F}_n + \tilde{G}_n) - n(\hat{\beta}^T \hat{E}_n^2 \hat{\beta} - 2\hat{\beta}^T \hat{F}_n + \hat{G}_n)}{n-r} \right) \xrightarrow{P} 0.$$

By Theorem A.1,

$$\frac{\|Y_n - X_n \hat{\beta}\|^2}{n-r} \xrightarrow{P} \sigma_e^2.$$

By Lemma D.2, $\|Y - X\hat{\beta}\|^2 = n(\hat{\beta}^T \hat{E}_n^2 \hat{\beta} - 2\hat{\beta}^T \hat{F}_n + \hat{G}_n)$. As a result, by Lemma B.4,

$$\frac{n(\tilde{\beta}^T \tilde{E}_n^2 \tilde{\beta} - 2\tilde{\beta}^T \tilde{F}_n + \tilde{G}_n)}{n-r} \xrightarrow{P} \sigma_e^2.$$

Next, by Theorem A.1, $\|X_n \hat{\beta} - X_n \hat{\beta}^N\|^2 \sim \chi_{r-q}^2(\eta^2)\sigma_e^2$. Then by Lemma D.2, $\|\sqrt{n}\hat{E}_n(\hat{\beta} - \hat{\beta}^N)\|^2 \sim \chi_{r-q}^2(\eta^2)\sigma_e^2$. We will show that

$$\|\sqrt{n}\tilde{E}_n(\tilde{\beta} - \tilde{\beta}^N)\|^2 \xrightarrow{D} \chi_{r-q}^2(\eta^2)\sigma_e^2.$$

First, we define the random vectors

$$\tilde{H}_n = \tilde{E}_n \sqrt{n}(\tilde{\beta} - \tilde{\beta}^N), \quad \hat{H}_n = \hat{E}_n \sqrt{n}(\hat{\beta} - \hat{\beta}^N).$$

By Lemma D.11, $\sqrt{n}(\tilde{\beta} - \hat{\beta}) \xrightarrow{P} 0$ and $\sqrt{n}(\tilde{\beta}^N - \hat{\beta}^N) \xrightarrow{P} 0$. And since by Lemma D.3, $\hat{E}_n \xrightarrow{P} C^{1/2}$, we have that by Slutsky's Theorem, $\sqrt{n}\hat{E}_n(\tilde{\beta} - \hat{\beta}) \xrightarrow{P} 0$, and $\sqrt{n}\hat{E}_n(\tilde{\beta}^N - \hat{\beta}^N) \xrightarrow{P} 0$. Also, by Lemma D.10, $\sqrt{n}(\tilde{E}_n - \hat{E}_n) \xrightarrow{P} 0$ which implies that, by Slutsky's Theorem, and Lemma D.11, $\sqrt{n}(\tilde{E}_n - \hat{E}_n)\tilde{\beta} \xrightarrow{P} 0$ and $\sqrt{n}(\tilde{E}_n - \hat{E}_n)\tilde{\beta}^N \xrightarrow{P} 0$ so that $\sqrt{n}(\tilde{E}_n - \hat{E}_n)(\tilde{\beta} - \tilde{\beta}^N) \xrightarrow{P} 0$.

As a result,

$$\tilde{H}_n - \hat{H}_n$$
$$= \tilde{E}_n\sqrt{n}(\tilde{\beta} - \tilde{\beta}^N) - \hat{E}_n\sqrt{n}(\hat{\beta} - \hat{\beta}^N)$$
$$= \tilde{E}_n\sqrt{n}\tilde{\beta} - \hat{E}_n\sqrt{n}\hat{\beta} - \left[\tilde{E}_n\sqrt{n}\tilde{\beta}^N - \hat{E}_n\sqrt{n}\hat{\beta}^N\right]$$
$$= (\tilde{E}_n - \hat{E}_n)\sqrt{n}\tilde{\beta} + \hat{E}_n\sqrt{n}(\tilde{\beta} - \hat{\beta}) - \left[(\tilde{E}_n - \hat{E}_n)\sqrt{n}\tilde{\beta}^N + \hat{E}_n\sqrt{n}(\tilde{\beta}^N - \hat{\beta}^N)\right]$$
$$= (\tilde{E}_n - \hat{E}_n)\sqrt{n}(\tilde{\beta} - \tilde{\beta}^N) + \hat{E}_n\sqrt{n}(\tilde{\beta} - \hat{\beta}) - \left[\hat{E}_n\sqrt{n}(\tilde{\beta}^N - \hat{\beta}^N)\right]$$
$$\xrightarrow{P} 0.$$

By Lemma D.14, since $\|\hat{H}_n\| \xrightarrow{D} \chi_{r-q}(\eta^2)\sigma_e$ and $\tilde{H}_n - \hat{H}_n \xrightarrow{P} 0$, we have that

$$\|\tilde{H}_n\|^2 \xrightarrow{D} \chi^2_{r-q}(\eta^2)\sigma_e^2.$$

As a result, $\|\sqrt{n}\tilde{E}_n(\tilde{\beta} - \tilde{\beta}^N)\|^2 \xrightarrow{D} \chi^2_{r-q}(\eta^2)\sigma_e^2$. This completes the proof.

$\square$

Lemma D.15 shows the convergence of individual quantities that can now be combined to show the convergence of the DP test statistic $\tilde{T}$:

*Proof of Theorem D.1.* First, by Corollary D.12, under the null hypothesis: $\tilde{\beta}^N = \tilde{\beta}^N_n \xrightarrow{P} \beta$. And under the alternative hypothesis: $\tilde{\beta} = \tilde{\beta}_n \xrightarrow{P} \beta$.

By Lemma D.2,

$$T = T_n = \frac{n-r}{r-q} \cdot \frac{\|X\hat{\beta} - X\hat{\beta}^N\|^2}{\|Y - X\hat{\beta}\|^2} = \frac{n-r}{r-q} \cdot \frac{\|\sqrt{n}\hat{E}_n(\hat{\beta} - \hat{\beta}^N)\|^2}{n(\hat{\beta}^T\hat{E}_n^2\hat{\beta} - 2\hat{\beta}^T\hat{F}_n + \hat{G}_n)}.$$

And by Equation (17),

$$\tilde{T} = \tilde{T}_n = \frac{n-r}{r-q} \cdot \frac{\|\sqrt{n}\tilde{E}_n(\tilde{\beta} - \tilde{\beta}^N)\|^2}{n(\tilde{\beta}^T\tilde{E}_n^2\tilde{\beta} - 2\tilde{\beta}^T\tilde{F}_n + \tilde{G}_n)}.$$

From Theorem A.1, in the non-private case where $Y_n \sim \mathcal{N}(X_n\beta, \sigma_e^2 I_{n\times n})$, if $T = T_n$ is the test statistic from Equation (7), then

$$T_n \sim F_{r-q,n-r}(\eta_n^2), \quad \eta_n^2 = \frac{\|X_n\beta - X_n\beta^N\|^2}{\sigma_e^2}.$$

Also, by Theorem A.1, the asymptotic distribution of $T$ is a chi-squared distribution. i.e., $T = T_n \xrightarrow{D} \frac{\chi^2_{r-q}(\eta^2)}{r-q}$. Next, we show that the DP $F$-statistic also has asymptotic distribution of chi-squared.

By Lemma D.15,

$$\|\sqrt{n}\tilde{E}_n(\tilde{\beta} - \tilde{\beta}^N)\|^2 \xrightarrow{D} \mathcal{X}^2_{r-q}(\eta^2)\sigma_e^2,$$

and

$$\frac{n(\tilde{\beta}^T\tilde{E}_n^2\tilde{\beta} - 2\tilde{\beta}^T\tilde{F}_n + \tilde{G}_n)}{n-r} \xrightarrow{P} \sigma_e^2.$$

Let

$$W_n = \left(\frac{\|\sqrt{n}\tilde{E}_n(\tilde{\beta} - \tilde{\beta}^N)\|^2}{r-q}, \frac{n(\tilde{\beta}^T\tilde{E}_n^2\tilde{\beta} - 2\tilde{\beta}^T\tilde{F}_n + \tilde{G}_n)}{n-r}\right) = (A_n, B_n),$$

and

$$W = \left(\frac{\mathcal{X}^2_{r-q}(\eta^2)\sigma_e^2}{r-q}, \sigma_e^2\right).$$

By the condition that $\sigma_e > 0$ ($\mathbb{P}[\sigma_e = 0] = 0$) and the continuous mapping theorem (Theorem D.13), $\tilde{T}_n = g(W_n) = g(A_n, B_n) = A_n/B_n$ converges, in distribution, to $\frac{\chi^2_{r-q}(\eta^2)}{r-q}$ as $n \to \infty$.

$\square$

# E   Experimental Evaluation of Power and Significance

We will measure the effectiveness of our hypothesis tests via significance and power. In Section H, we describe our meta-procedures for collating the significance and power of our implementations of non-private and private statistical tests.

The power and significance of our differentially private tests are estimated on both semi-synthetic datasets based on the Opportunity Atlas [19, 20] and on synthetic datasets. The OI semi-synthetic datasets consists of simulated microdata for each census tract in some states in the U.S. The dependent variable $Y$ is the child national income percentile and the independent variable $X$ is the corresponding parent national income percentile. See [3] for more details on the properties of simulated data from the OI team. In the OI data, $X$ is lognormally distributed and the distribution of counts of individuals across tracts in a state follows an exponential distribution.

**General Parameter Setup for Synthetic Data:**   For experimental evaluation on synthetic datasets, we generated datasets with sizes between $n = 100$ and $n = 10,000$.

For both the linear relationship and mixture model tests on synthetic data below, we consider a subset of the following values of the privacy budget $\rho$: $\{0.1^2/2, 0.5^2/2, 1^2/2, 2^2/2, 3^2/2, 5^2/2, 10^2/2\}$.

We draw the independent variables $x_1, \ldots, x_n$ according to a few different distributions: Normal, Uniform, Exponential. We will detail the parameters used to generated variables from these distributions in the corresponding subsections.

For all tests below, the clipping parameter is either set to $\Delta = 2$ or $\Delta = 3$. For the synthetic data, the dependent variable $Y$ is generated using the linear or mixture model specification described in previous sections and by fixing or varying parameters (such as $\sigma_e$). For estimating the power and significance, we fix the target significance level to 0.05 and run Monte Carlo tests 2000 times. We estimate the power and significance as the fraction of times the null is rejected, given various settings of parameters that satisfy the alternative and null hypothesis, respectively.

## E.1   Testing a Linear Relationship on Synthetic Data

### E.1.1   $F$-statistic

We evaluate our DP linear relationship test on synthetically generated data from three different distributions: normal, uniform, and exponential. We also vary parameters such as: the slope of the linear model and the noise distribution of the dependent variable.

**Evaluating the Significance for Normally Distributed Independent Variables**: Generally, we see that the significance remains below the target significance level, on average, for all values of $\rho$. For the linear relationship tester, when the standard deviation of the dependent variable ($\sigma_e$) is small (Figure 4a), we see that the true significance level is well below the target significance of 0.05, which is fine (but conservative). We conjecture that this happens because when $\sigma_e$ is small: (i) we fail to reject when the noisy estimate of $\sigma_e$ is $\leq 0$; or (ii) the test statistic under the null distribution will be almost always 0 since under the null (even without privacy), the standard deviation of the test statistic is proportional to $\sigma_e$. In Figures 4a, 4b and 4c, we see the significance of the linear tester attains the target (of 0.05) as we vary the noise in the dependent variable $\sigma_e$.

**Evaluating Power for Varying the Noise in the Dependent Variable**: For Figures 5a, 5b, and 5c, we set the true slope to 1. We then vary the noise in the dependent variable. That is, for the general linear model, $Y \sim \mathcal{N}(X\beta, \sigma_e^2 I_{n \times n})$, we vary $\sigma_e$. The following values of $\sigma_e$ are considered: $\{0.001, 0.35, 1\}$.

In Figure 5a, we generally see that compared to higher values of $\sigma_e$ (Figures 5b and 5c), the power is relatively low. We believe this occurs because when $\sigma_e$ is small, its DP estimate is more likely to be

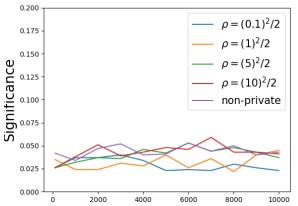 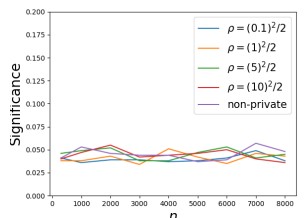 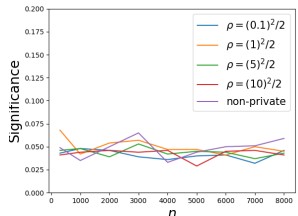

(a) Significance for testing a linear relationship. $x_i \sim \mathcal{N}(0.5, 1)$, $y_i \sim 0 \cdot x_i + \mathcal{N}(0, 0.001^2)$. $\Delta = 2$.

(b) Significance for testing a linear relationship. $x_i \sim \mathcal{N}(0.5, 1)$, $y_i \sim 0 \cdot x_i + \mathcal{N}(0, 0.35^2)$. $\Delta = 2$.

(c) Significance for testing a linear relationship. $x_i \sim \mathcal{N}(0.5, 1)$, $y_i \sim 0 \cdot x_i + \mathcal{N}(0, 1)$. $\Delta = 2$.

Figure 4

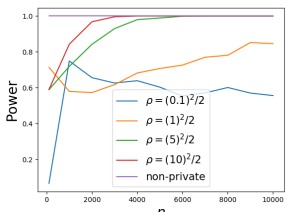 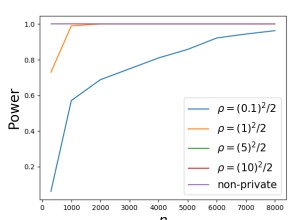 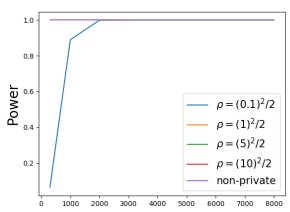

(a) Power for testing a linear relationship. $x_i \sim \mathcal{N}(0.5, 1)$, $y_i \sim 1 \cdot x_i + \mathcal{N}(0, 0.001^2)$. $\Delta = 2$.

(b) Power for testing a linear relationship. $x_i \sim \mathcal{N}(0.5, 1)$, $y_i \sim 1 \cdot x_i + \mathcal{N}(0, 0.35^2)$. $\Delta = 2$.

(c) Power for testing a linear relationship. $x_i \sim \mathcal{N}(0.5, 1)$, $y_i \sim 1 \cdot x_i + \mathcal{N}(0, 1)$. $\Delta = 2$.

Figure 5

less than 0, in which case we fail to reject the null (even when the alternative is true). This generally leads to a reduction in the power.

**Evaluating Power for Varying Slopes**: Figures 6a, 6b show the power of the linear test for slopes of 0.1, 1. We generally see that the larger the slope, the higher the power of the DP tests.

**Evaluating the Significance while Varying the Distribution of the Independent Variable**: For Figures 7a, 7b, and 7c, we set the standard deviation of the noise dependent variable to 0.35. We then

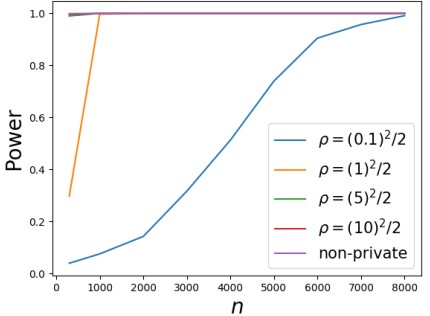 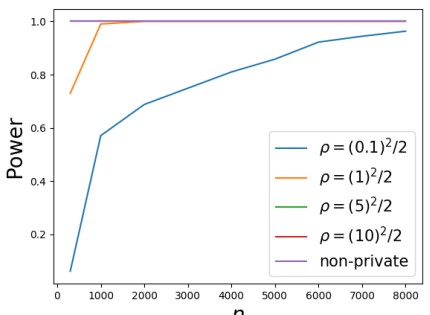

(a) Power for testing a linear relationship. $x_i \sim \mathcal{N}(0.5, 1)$, $y_i \sim 0.1 \cdot x_i + \mathcal{N}(0, 0.35^2)$. $\Delta = 2$.

(b) Power for testing a linear relationship. $x_i \sim \mathcal{N}(0.5, 1)$, $y_i \sim 1 \cdot x_i + \mathcal{N}(0, 0.35^2)$. $\Delta = 2$.

Figure 6

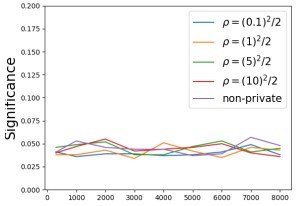
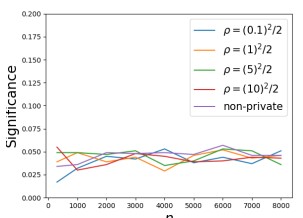
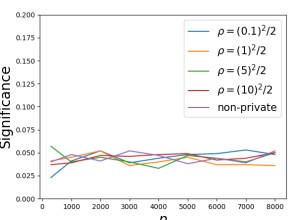

(a) Significance for testing a linear relationship. Normal Distribution on $X$.

(b) Significance for testing a linear relationship. Uniform Distribution on $X$.

(c) Significance for testing a linear relationship. Exponential Distribution on $X$.

Figure 7

vary the distribution of the independent variable — while maintaining the variance — to take on one of the following:

1. **Normal**: with mean 0.5 and variance 1/12.
2. **Uniform**: between 0 and 1 (variance of 1/12).
3. **Exponential**: with scale of $1/\sqrt{12}$.

We observe that the significance is still preserved even though, in our DP testers, the null distribution is simulated via a normal distribution.

### E.1.2   Differentially Private Bootstrap Confidence Intervals

Using the duality between confidence interval estimation and hypothesis testing, we can construct hypothesis tests for testing a linear relationship based on DP confidence interval procedures. Specifically, we compare the $F$-statistic linear relationship tester to the tester that uses DP confidence intervals. See Section G for more details on the experimental framework of the DP bootstrap confidence intervals. Algorithm 7 summarizes the approach for testing that builds on DP confidence intervals.

In Figure 8a, we present experimental results for the significance level of Algorithm 7 compared to Algorithm 3 instantiated with the DP $F$-statistic. As we see, Algorithm 7 achieves the target significance level. In Figure 8b, we also present experimental results for the power of Algorithm 7 compared to Algorithm 3. We see that Algorithm 7 has less power than Algorithm 3. This observation is more pronounced for less concentrated distributions (i.e., uniform) on the independent variable. See Figure 9b. This might be due to the, sometimes excessive, width of the confidence interval produced by the bootstrap interval (in order to ensure coverage under the null hypothesis).

Figures 8a, 8b, 9a, and 9b show results averaged out over 2000 trials. The dashed lines correspond to the bootstrap confidence interval approach (denoted CI) while the solid lines are for the $F$-statistic (denoted $F$-stat).

### E.2   Testing Mixture Models on Synthetic Data

### E.2.1   $F$-Statistic

We evaluate the $F$-statistic DP mixture model test on synthetically generated data. We vary parameters such as: the fraction of data in each group and the slopes used to generate data for each group. Let $\beta_1, \beta_2$ denote the slopes of the two groups.

**Evaluating the Significance**: Like in the DP linear model tester, we also see that we achieve the target significance levels, on average, for all values of $\rho$. In Figures 10a, 10b, and 10c, we vary the noise in the dependent variable. In Figures 11a, 11b, and 11c, we vary the fraction of group sizes, using either a 1/8, 1/4, or 1/2 fraction for the first group. We see that the more unbalanced splits tend to have lower significance levels.

**Power while Varying the Group Size Fraction**: Let $n$ be the total number of datapoints and $n_1, n_2$ be the number of points in groups 1 and 2 respectively. We vary the fraction of points in group

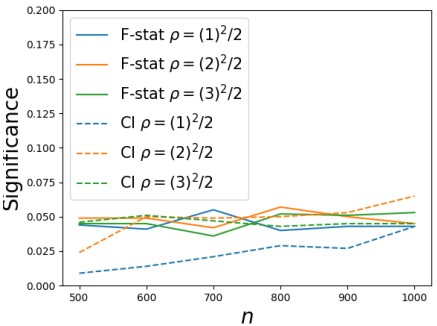
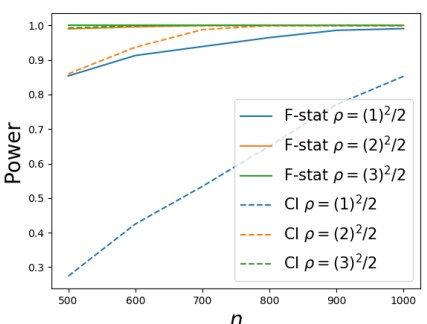

(a) Significance for $F$-statistic versus confidence interval approach. $x_i \sim \mathcal{N}(0.5, 1)$, $y_i \sim 0 \cdot x_i + \mathcal{N}(0, 0.35^2)$. $\Delta = 2$.

(b) Power for $F$-statistic versus confidence interval approach. $x_i \sim \mathcal{N}(0.5, 1)$, $y_i \sim 1 \cdot x_i + \mathcal{N}(0, 0.35^2)$. $\Delta = 2$.

Figure 8

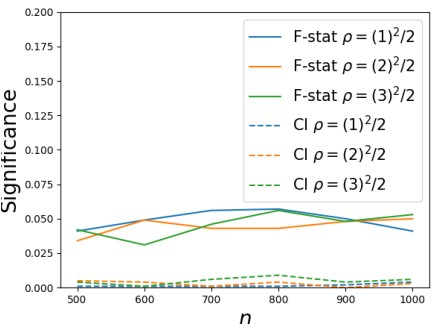
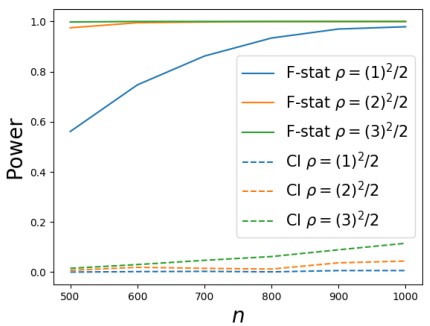

(a) Significance for $F$-statistic versus confidence interval approach. $x_i \sim \text{Unif}(0, 1)$, $y_i \sim 0 \cdot x_i + \mathcal{N}(0, 0.35^2)$. $\Delta = 2$.

(b) Power for $F$-statistic versus confidence interval approach. $x_i \sim \text{Unif}(0, 1)$, $y_i \sim 1 \cdot x_i + \mathcal{N}(0, 0.35^2)$. $\Delta = 2$.

Figure 9

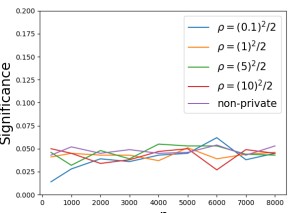
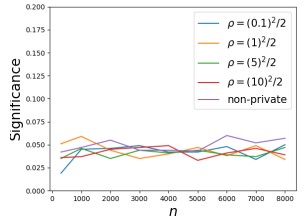
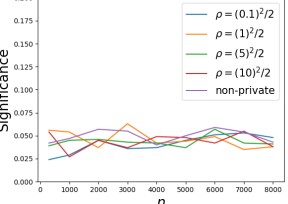

(a) Significance for testing mixtures. Equal-sized groups. $x_i \sim \mathcal{N}(0.5, 1)$, $y_i \sim 1 \cdot x_i + \mathcal{N}(0, 0.01^2)$ for Group 1. $y_i \sim 1 \cdot x_i + \mathcal{N}(0, 0.01^2)$ for Group 2.

(b) Significance for testing mixtures. Equal-sized groups. $x_i \sim \mathcal{N}(0.5, 1)$, $y_i \sim 1 \cdot x_i + \mathcal{N}(0, 0.35^2)$ for Group 1. $y_i \sim 1 \cdot x_i + \mathcal{N}(0, 0.35^2)$ for Group 2.

(c) Significance for testing mixtures. Equal-sized groups. $x_i \sim \mathcal{N}(0.5, 1)$, $y_i \sim 1 \cdot x_i + \mathcal{N}(0, 1)$ for Group 1. $y_i \sim 1 \cdot x_i + \mathcal{N}(0, 1)$ for Group 2.

Figure 10

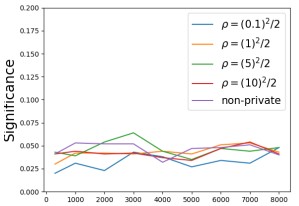

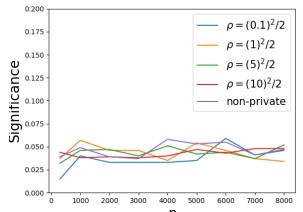

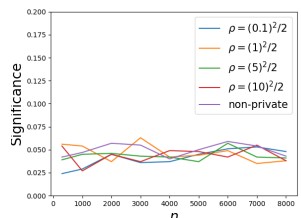

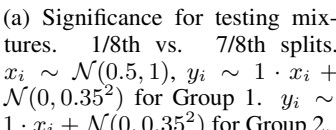

(a) Significance for testing mixtures. 1/8th vs. 7/8th splits. $x_i \sim \mathcal{N}(0.5, 1)$, $y_i \sim 1 \cdot x_i + \mathcal{N}(0, 0.35^2)$ for Group 1. $y_i \sim 1 \cdot x_i + \mathcal{N}(0, 0.35^2)$ for Group 2.

(b) Significance for testing mixtures. 1/4th vs. 3/4th splits. $x_i \sim \mathcal{N}(0.5, 1)$, $y_i \sim 1 \cdot x_i + \mathcal{N}(0, 0.35^2)$ for Group 1. $y_i \sim 1 \cdot x_i + \mathcal{N}(0, 0.35^2)$ for Group 2.

(c) Significance for testing mixtures. Equal-sized groups. $x_i \sim \mathcal{N}(0.5, 1)$, $y_i \sim 1 \cdot x_i + \mathcal{N}(0, 0.35^2)$ for Group 1. $y_i \sim 1 \cdot x_i + \mathcal{N}(0, 0.35^2)$ for Group 2.

Figure 11

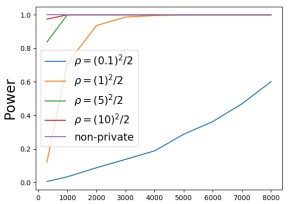

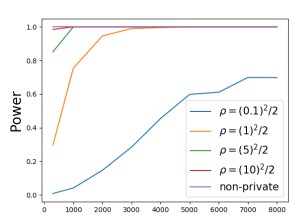

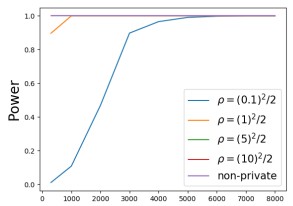

(a) Power for testing mixtures. 1/8th vs. 7/8th splits. $x_i \sim \mathcal{N}(0.5, 1)$, $y_i \sim -1 \cdot x_i + \mathcal{N}(0, 0.35^2)$ for Group 1. $y_i \sim 1 \cdot x_i + \mathcal{N}(0, 0.35^2)$ for Group 2.

(b) Power for testing mixtures. 1/4th vs. 3/4th splits. $x_i \sim \mathcal{N}(0.5, 1)$, $y_i \sim -1 \cdot x_i + \mathcal{N}(0, 0.35^2)$ for Group 1. $y_i \sim 1 \cdot x_i + \mathcal{N}(0, 0.35^2)$ for Group 2.

(c) Power for testing mixtures. Equal-sized groups. $x_i \sim \mathcal{N}(0.5, 1)$, $y_i \sim -1 \cdot x_i + \mathcal{N}(0, 1)$ for Group 1. $y_i \sim 1 \cdot x_i + \mathcal{N}(0, 1)$ for Group 2.

Figure 12

1: $n_1/n$. Setting the slopes of each group to $\beta_1 = -1$ and $\beta_2 = 1$, we vary this fraction so that $n_1/n \in \{1/8, 1/4, 1/2\}$. For Figure 12a, we set the group sizes to be equal. For Figure 12b, we set the group sizes to be $n/4, 3n/4$. And last, for Figure 12c, the group sizes are $n/8, 7n/8$.

Generally, the more even the group size fractions are, the higher the power of the DP test for testing mixtures in the general linear model.

**Power while Varying the Difference Between Slopes in Each Group**: Let $\beta_1, \beta_2$ correspond to the slopes for groups 1 and 2. We vary $|\beta_1 - \beta_2|$. Generally, we see that the larger $|\beta_1 - \beta_2|$ is, the higher the power of the test. In Figures 13a, 13b we vary the slope in the two groups and observe the aforementioned phenomena.

**Power while Varying the Noise in the Dependent Variable**: We also vary $\sigma_e$. We generally see that the smaller it is, the smaller the power. We conjecture that this happens because we err on the side of failing to reject the null if the DP estimate of $\sigma_e$ becomes $\leq 0$, which is more likely to happen if $\sigma_e$ is small. In Figures 14a, 14b,and 14c we see this phenomenon.

### E.2.2 Nonparametric Tests via Kruskal-Wallis

We now proceed to show results for comparing the mixture models based on Kruskal-Wallis (KW) to the parametric $F$-statistic method.

**Evaluating the Significance**: The KW methods, on average, achieve the target significance levels for all values of $\rho$ as illustrated in Figures 15a and 15b, where we vary the noise in the dependent variable.

**Evaluating the Power as we Increase the Difference in Slopes**: We see that the the KW method outperforms the $F$-statistic method on small datasets. But as the difference in slopes between the two

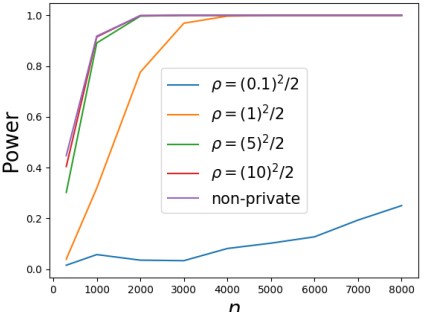
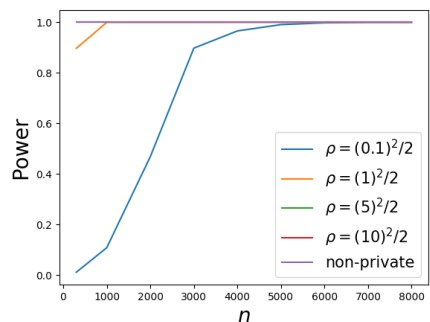

(a) Power for testing mixtures. Equal-sized groups. $x_i \sim \mathcal{N}(0.5, 1)$, $y_i \sim -0.1 \cdot x_i + \mathcal{N}(0, 0.35^2)$ for Group 1. $y_i \sim 0.1 \cdot x_i + \mathcal{N}(0, 0.35^2)$ for Group 2.

(b) Power for testing mixtures. Equal-sized groups. $x_i \sim \mathcal{N}(0.5, 1)$, $y_i \sim -1 \cdot x_i + \mathcal{N}(0, 0.35^2)$ for Group 1. $y_i \sim 1 \cdot x_i + \mathcal{N}(0, 0.35^2)$ for Group 2.

Figure 13

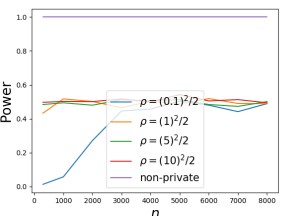
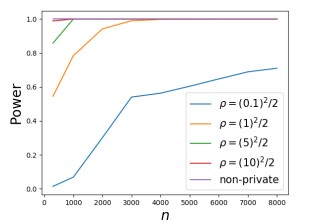
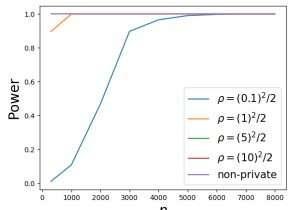

(a) Power for testing mixtures. Equal-sized groups. $x_i \sim \mathcal{N}(0.5, 1)$, $y_i \sim -1 \cdot x_i + \mathcal{N}(0, 0.01^2)$ for Group 1. $y_i \sim 1 \cdot x_i + \mathcal{N}(0, 0.01^2)$ for Group 2.

(b) Power for testing mixtures. Equal-sized groups. $x_i \sim \mathcal{N}(0.5, 1)$, $y_i \sim -1 \cdot x_i + \mathcal{N}(0, 0.35^2)$ for Group 1. $y_i \sim 1 \cdot x_i + \mathcal{N}(0, 0.35^2)$ for Group 2.

(c) Power for testing mixtures. Equal-sized groups. $x_i \sim \mathcal{N}(0.5, 1)$, $y_i \sim -1 \cdot x_i + \mathcal{N}(0, 1)$ for Group 1. $y_i \sim 1 \cdot x_i + \mathcal{N}(0, 1)$ for Group 2.

Figure 14

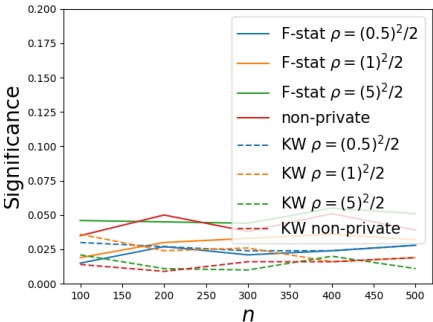
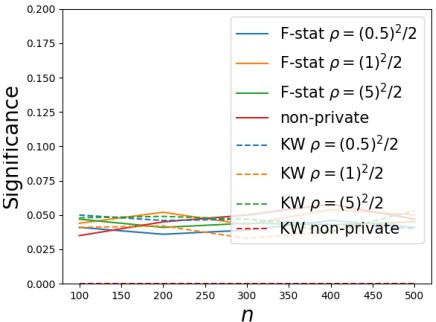

(a) Significance for testing mixtures. Equal-sized groups. $x_i \sim \mathcal{N}(0.5, 0.1)$, $y_i \sim 1 \cdot x_i + \mathcal{N}(0, 0.35^2)$ for Group 1. $y_i \sim 1 \cdot x_i + \mathcal{N}(0, 0.35^2)$ for Group 2.

(b) Significance for testing mixtures. Equal-sized groups. $x_i \sim \mathcal{N}(0.5, 1)$, $y_i \sim 1 \cdot x_i + \mathcal{N}(0, 1)$ for Group 1. $y_i \sim 1 \cdot x_i + \mathcal{N}(0, 1)$ for Group 2.

Figure 15

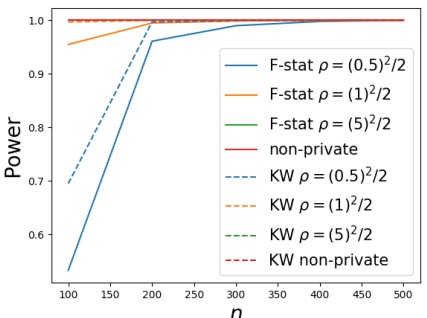
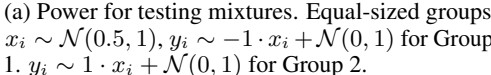
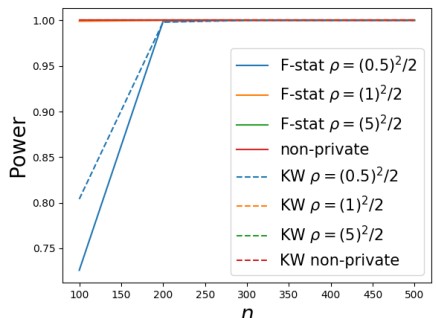

(a) Power for testing mixtures. Equal-sized groups. $x_i \sim \mathcal{N}(0.5, 1)$, $y_i \sim -1 \cdot x_i + \mathcal{N}(0, 1)$ for Group 1. $y_i \sim 1 \cdot x_i + \mathcal{N}(0, 1)$ for Group 2.

(b) Power for testing mixtures. Equal-sized groups. $x_i \sim \mathcal{N}(0.5, 1)$, $y_i \sim -1 \cdot x_i + \mathcal{N}(0, 1)$ for Group 1. $y_i \sim 5 \cdot x_i + \mathcal{N}(0, 1)$ for Group 2.

Figure 16

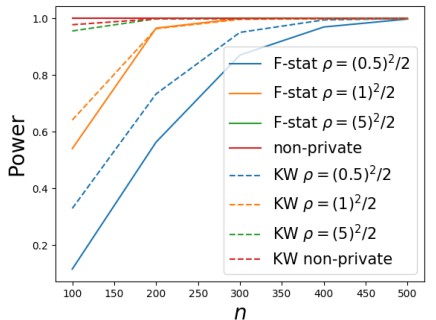
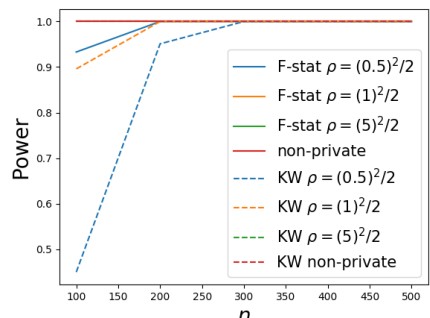

(a) Power for Kruskal-Wallis versus the $F$-statistic. $x_i \sim \mathcal{N}(0.5, 1)$, $y_i \sim -1 \cdot x_i + \mathcal{N}(0, 1)$ for Group 1. $y_i \sim 1 \cdot x_i + \mathcal{N}(0, 1)$ for Group 2.

(b) Power for Kruskal-Wallis versus the $F$-statistic. $x_i \sim \mathcal{N}(0.5, 10)$, $y_i \sim -1 \cdot x_i + \mathcal{N}(0, 1)$ for Group 1. $y_i \sim 1 \cdot x_i + \mathcal{N}(0, 1)$ for Group 2.

Figure 17

groups increases, the $F$-statistic method does better and begins to outperform the KW method. See Figures 16a and 16b.

**Evaluating the Power as we Increase the Variance of the Independent Variable**: In Figures 17a and 17b, we see that the $F$-statistic method outperforms the KW method when the variance of the independent variable is much larger (10x) than previously.

### E.3 Testing on Opportunity Insights Data

The Opportunity Insights (OI) team gave us simulated data for census tracts from the following states in the United States: Idaho, Illinois, New York, North Carolina, Texas, and Tennessee. The dependent and independent variables are the child and parent national income percentiles, respectively. For the linear tester, a rejection of the null hypothesis implies that there is a relationship between the parent and child income percentiles. For the mixture model tester, it implies that there is more than one linear relationship in the data which suggests that more granular data is needed for analysis on the data. The groups of data fed to the mixture model tester are conglomeration of one or more tracts.

Some of these states have a small number of datapoints. For example, within Illinois, there are tracts with just $n = 39$ datapoints. For the Illinois dataset, there are $n = 219,594$ datapoints that are subdivided into $3,108$ census tracts. The North Carolina and Texas datasets consists of datapoints subdivided into $2,156$ and $5,187$ census tracts respectively. We will focus on data from North

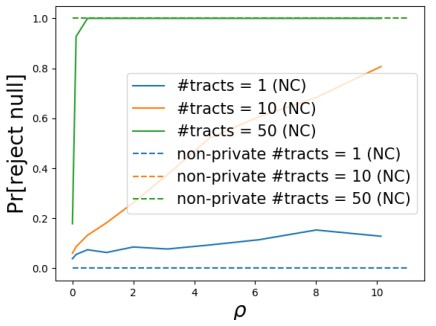
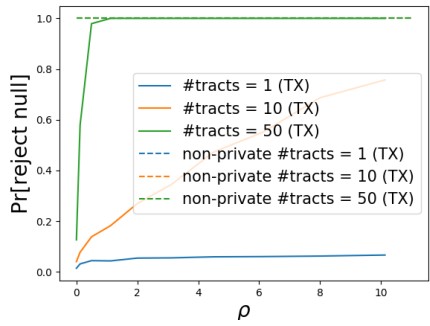

(a) $\mathbb{P}[\text{reject null}]$ for testing a linear relationship in NC. $\Delta = 2$.

(b) $\mathbb{P}[\text{reject null}]$ for testing a linear relationship in TX. $\Delta = 2$.

Figure 18

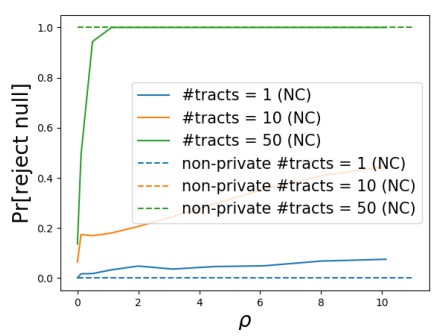
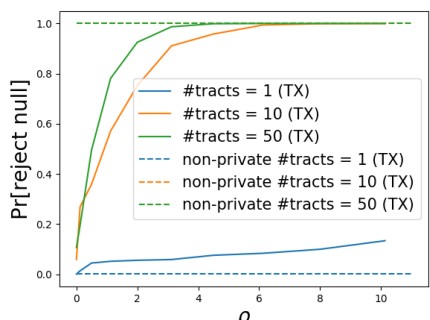

(a) $\mathbb{P}[\text{reject null}]$ for testing for mixtures in NC. $\Delta = 2$.

(b) $\mathbb{P}[\text{reject null}]$ for testing for mixtures in TX. $\Delta = 2$.

Figure 19

Carolina (NC), and Texas (TX) and experimentally evaluate $\mathbb{P}[\text{reject null}]$, the probability of rejecting the null hypothesis over the randomness of the DP algorithms. We run our tests on some census tracts in these states showing how these measures fair as the privacy parameter is relaxed. For the experiments below, from each state, we randomly and uniformly select: (i) a single tract; (ii) 10 randomly selected tracts and concatenate; and (iii) 50 randomly selected tracts and concatenate. Then we test for the presence of a (non-zero) linear relationship. The concatenation could result in hundreds or thousands of points.

Our tests are evaluated on the OI data. We have not included the test based on Kruskal-Wallis as our current implementation is, at the moment, relatively computationally inefficient to evaluate on such large datasets. See above synthetic data experiments for comparison of Kruskal-Wallis to the $F$-statistic method. Figures 18a, and 18b show the probability of rejecting the null as we increase the parameter $\rho$ when using the DP linear tester. Figures 19a, and 19b show the corresponding results for the $F$-stat based DP mixture model tester. We see that for the small-sized datasets tend to have a small chance of rejecting the null while larger ones have a higher chance.

### E.4 Testing on UCI Bike Dataset

We use the UCI bike dataset [28] with 17,389 instances. For this dataset, we test for a linear relationship between the "temp" (normalized temperature in Celsius) and "hr" (hour between 0 and 23) attributes. The null hypothesis is that there is no linear relationship between the "temp" and "hr" attributes. Without privacy, the linear relationship tester based on the $F$-statistic rejects the null. In Table 1, we show the probability of Algorithm 3 rejecting the null as we vary the

| $\rho$ | 0.005 | 0.125 | 0.5 | 1.125 | 2.0 | 3.125 | 4.5 | 6.125 | 8.0 | 10.125 | non-DP |
|---|---|---|---|---|---|---|---|---|---|---|---|
| $\mathbb{P}[\text{reject null} \mid 100\% \text{ data}]$ | 1.0 | 1.0 | 1.0 | 1.0 | 1.0 | 1.0 | 1.0 | 1.0 | 1.0 | 1.0 | 1.0 |
| $\mathbb{P}[\text{reject null} \mid 10\% \text{ data}]$ | 0.85 | 1.0 | 1.0 | 1.0 | 1.0 | 1.0 | 1.0 | 1.0 | 1.0 | 1.0 | 1.0 |

Table 1: $\mathbb{P}[\text{reject null}]$ for testing for a linear relationship between temperature and time (in hours).

privacy parameter. We can observe that for almost all—except for the smallest setting of $\rho$—privacy parameters, $\mathbb{P}[\text{reject null} \mid p\% \text{ data}]$ (probability of rejecting the null, given $p\%$ of the dataset) for the private test matches that of the non-private test.

While we show that our methods can run on real-world datasets, the synthetically generated datasets give a lot more information on the behavior of the tests.

## F $F$-Statistic for the General Linear Model

The proofs in this section rely on insights from [37]. In fact, Theorem F.2 can be seen as a special case of Theorem 14.11 in [37] where, under the null hypothesis, the projection onto $\omega_0$ results in $\beta^N$ and, under the alternative hypothesis, the projection onto $\omega$ results in $\beta$.

We present the main test statistic we will use for hypothesis testing. This statistic is equivalent to the generalized likelihood ratio test statistic and can be written as

$$T = \left(\frac{n-r}{r-q}\right) \frac{\|Y - X\hat{\beta}^N\|^2 - \|Y - X\hat{\beta}\|^2}{\|Y - X\hat{\beta}\|^2} = \left(\frac{n-r}{r-q}\right) \frac{\|X\hat{\beta} - X\hat{\beta}^N\|^2}{\|Y - X\hat{\beta}\|^2}, \quad (24)$$

where $\hat{\beta}^N, \hat{\beta}$ are the least squares estimates under the null and alternative hypothesis respectively.

The vectors $Y - X\hat{\beta}$ and $X\hat{\beta} - X\hat{\beta}^N$ can be shown to be orthogonal, so that $\|Y - X\hat{\beta}^N\|^2 = \|Y - X\hat{\beta}\|^2 + \|X\hat{\beta} - X\hat{\beta}^N\|^2$ by the Pythagorean theorem [37].

**Lemma F.1** (Weak Law of Large Numbers, see [37]). *Let $Y_1, \ldots, Y_n$ be i.i.d. random variables with mean $\mu$. Then*

$$\frac{1}{n}\sum_{i=1}^{n} Y_i = \bar{Y}_n \xrightarrow{P} \mu,$$

*provided that $\mathbb{E}[|Y_i|] < \infty$.*

**Theorem F.2.** *For every $n \in \mathbb{N}$ with $n > r$, let $X = X_n \in \mathbb{R}^{n \times p}$ be the design matrix. Under the general linear model $Y = Y_n \sim \mathcal{N}(X_n\beta, \sigma_e^2 I_{n \times n})$,*

$$T = T_n \sim F_{r-q,n-r}(\eta_n^2), \quad \eta_n^2 = \frac{\|X_n\beta - X_n\beta^N\|^2}{\sigma_e^2},$$

*where $F_{n,m}$ is the F-distribution with parameters $n$, $m$, $\beta^N = \mathbb{E}[\hat{\beta}^N]$, $q$ is the dimension of $\omega_0$, and $r$ is the dimension of $\omega$ with $0 \leq q < r$.*

*Furthermore,*

1.
$$\|Y_n - X_n\hat{\beta}\|^2 \sim \mathcal{X}_{n-r}^2 \sigma_e^2, \quad \|X_n\hat{\beta} - X_n\hat{\beta}^N\|^2 \sim \mathcal{X}_{r-q}^2(\eta_n^2)\sigma_e^2.$$

2. *If there exists $\eta \in \mathbb{R}$ such that $\frac{\|X_n\beta - X_n\beta^N\|^2}{\sigma_e^2} \to \eta^2$, then*

$$T = T_n \sim F_{r-q,n-r}(\eta_n^2) \xrightarrow{D} \frac{\chi_{r-q}^2(\eta^2)}{r-q}.$$

3. *We have*

$$\frac{\|Y_n - X_n\hat{\beta}\|^2}{n-r} \xrightarrow{P} \sigma_e^2.$$

*The values $\beta = \mathbb{E}[\hat{\beta}], \beta^N = \mathbb{E}[\hat{\beta}^N]$ are the expected values of our parameter estimates under the alternative and null hypotheses respectively.*

*Proof of Lemma F.2.* First, define $\Omega_0 = \{X\beta \; : \; \beta \in \mathbb{R}^p, \beta \in \omega_0\}$ (for null hypothesis) and $\Omega = \mathrm{span}\{c_1, \ldots, c_p\} = \{X\beta \; : \; \beta \in \mathbb{R}^p, \beta \in \omega\}$ (for alternative) where $c_1, \ldots, c_p$ are the columns of $X$. Write $Y = \sum_{i=1}^n Z_i v_i$, where $v_1, \ldots, v_n$ is an orthonormal basis chosen so that $v_1, \ldots, v_r$ spans $\Omega$ (so that $v_{r+1}, \ldots, v_n$ lies in $\Omega^\perp$) and $v_1, \ldots, v_q$ spans $\Omega_0$.

For all $i \in [n]$, we can find $Z_i \in \mathbb{R}^n$ by introducing an $n \times n$ matrix $O$ with columns $v_1, \ldots, v_n$. As a result, $O$ is an orthogonal matrix (i.e., $O^T O = OO^T = I$ since $O$ is a square matrix) such that $Z = O^T Y$ (or $Y = OZ$).

As before $Y = X\beta + e$, where $e \sim \mathcal{N}(0, \sigma_e^2 I_{n \times n})$. As a result, $Z = O^T(X\beta + e) = O^T X\beta + O^T e$. If we define $\tau = O^T X\beta$ and $e^* = O^T e$, then $Z = \tau + e^*$. And because $\mathbb{E}[e^*] = \mathbb{E}[O^T e] = O^T \mathbb{E}[e] = 0$ and $\mathrm{cov}(e^*) = \mathrm{cov}(O^T e) = O^T \mathrm{cov}(e) O = O^T(\sigma_e^2 I)O = \sigma_e^2 I$, $e^* \sim \mathcal{N}(0, \sigma_e^2 I_{n \times n})$. As a result,

$$Z \sim \mathcal{N}(\tau, \sigma_e^2 I_{n \times n}).$$

Next, since $c_1, \ldots, c_p$ denotes the columns of the design matrix $X$, $X\beta = \sum_{i=1}^p \beta_i c_i$ and

$$\tau = O^T X\beta = \begin{pmatrix} v_1^T \\ v_2^T \\ \vdots \\ v_n^T \end{pmatrix} \sum_{i=1}^p \beta_i c_i = \begin{pmatrix} \sum_{i=1}^p \beta_i v_1^T c_i \\ \sum_{i=1}^p \beta_i v_2^T c_i \\ \vdots \\ \sum_{i=1}^p \beta_i v_n^T c_i \end{pmatrix}.$$

And because $c_1, \ldots, c_p$ all lie in $\Omega$ and $v_{r+1}, \ldots, v_n$ in $\Omega^\perp$, we have $v_k^T c_i = 0$ for all $k > r$. Then, $\tau_{r+1} = \cdots = \tau_n = 0$.

Additionally, because $\tau = O^T X\beta$,

$$X\beta = O\tau = (v_1 \cdots v_n) \begin{pmatrix} \tau_1 \\ \tau_2 \\ \vdots \\ \tau_r \\ 0 \\ \vdots \\ 0 \end{pmatrix} = \sum_{i=1}^r \tau_i v_i.$$

*Essentially, we established a one-to-one relation between points $X\beta \in \Omega$ and $(\tau_1, \ldots, \tau_r) \in \mathbb{R}^r$.*

Now, since $Z \sim \mathcal{N}(\tau, \sigma_e^2 I_{n \times n})$, $Z_1, \ldots, Z_n$ are independent and $Z_i \sim \mathcal{N}(\tau_i, \sigma_e^2)$ for all $i \in [n]$. Furthermore, $\tau_{r+1} = \cdots = \tau_n = 0$.

Then since $X\beta = \sum_{i=1}^r \tau_i v_i$, $X\hat{\beta} = \sum_{i=1}^r Z_i v_i$ and $X\hat{\beta}^N = \sum_{i=1}^q Z_i v_i$.

As a result, we get

$$\|Y - X\hat{\beta}\|^2 = \|\sum_{i=r+1}^n Z_i v_i\|^2 = \sum_{i=r+1}^n \sum_{j=r+1}^n Z_i Z_j v_i^T v_j = \sum_{i=r+1}^n Z_i^2,$$

which follows since for all $i \neq j$, $v_i^T v_j = 0$ and for $i = j$, $v_i^T v_j = 1$. Also, since $\tau_{r+1} = \cdots = \tau_n = 0$, $Z_i \sim \sigma_e \mathcal{N}(\tau_i, 1)$, $\|Y - X\hat{\beta}\|^2 \sim \mathcal{X}_{n-r}^2 \sigma_e^2$.

Similarly,

$$\|Y - X\hat{\beta}^N\|^2 = \sum_{i=q+1}^n Z_i^2.$$

Then by Equation (24),

$$T = \frac{\frac{1}{r-q}\sum_{i=q+1}^r Z_i^2}{\frac{1}{n-r}\sum_{i=r+1}^n Z_i^2} = \frac{\frac{1}{r-q}\sum_{i=q+1}^r (Z_i/\sigma_e)^2}{\frac{1}{n-r}\sum_{i=r+1}^n (Z_i/\sigma_e)^2}.$$

The variables $Z_i$ are independent and because $Z_i/\sigma_e \sim \mathcal{N}(\tau_i/\sigma_e, 1)$, using properties of the (non-central) chi-squared distribution,

$$\sum_{i=q+1}^{r} \left(\frac{Z_i}{\sigma_e}\right)^2 \sim \chi^2_{r-q}(\eta_n^2), \quad \eta_n^2 = \sum_{i=q+1}^{r} \frac{\tau_i^2}{\sigma_e^2}.$$

As a corollary, $\|X\hat{\beta} - X\hat{\beta}^N\|^2 \sim \mathcal{X}^2_{r-q}(\eta_n^2)\sigma_e^2$.

Also, since $\tau_i = 0$ for $i = r+1, \ldots, n$, $Z_i/\sigma_e \sim \mathcal{N}(0,1)$ for $i = r+1, \ldots, n$. As a result, $\sum_{i=r+1}^{n}(Z_i/\sigma_e)^2 \sim \chi^2_{n-r}$.

By definition of the noncentral $F$-distribution, we have $T \sim F_{r-q,n-r}(\eta_n^2)$ where $\eta_n^2 = \sum_{i=q+1}^{r} \frac{\tau_i^2}{\sigma_e^2}$.

We know that $X\beta = \mathbb{E}[X\hat{\beta}] = \sum_{i=1}^{r} \tau_i v_i$ and $X\beta^N = \mathbb{E}[X\hat{\beta}^N] = \sum_{i=1}^{q} \tau_i v_i$. As a result, $X\beta - X\beta^N = \sum_{i=q+1}^{r} \tau_i v_i$ so that

$$\|X\beta - X\beta^N\|^2 = \sum_{i=q+1}^{r} \tau_i^2.$$

This completes the proof of the distribution of $T$.

In the limit, by Lemma F.3

$$T_n = F_{r-q,n-r}(\eta_n^2) \xrightarrow{D} \frac{\chi^2_{r-q}(\eta^2)}{r-q}.$$

Finally, we have established that

$$\frac{\|Y_n - X_n\hat{\beta}\|^2}{n-r} \sim \frac{\mathcal{X}^2_{n-r}\sigma_e^2}{n-r}.$$

Applying the weak law of large numbers (Lemma F.1), $\frac{\chi^2_{n-r}}{n-r} \xrightarrow{P} 1$. As a result,

$$\frac{\|Y_n - X_n\hat{\beta}\|^2}{n-r} \xrightarrow{P} \sigma_e^2.$$

$\square$

**Lemma F.3.** *Let $X \sim F_{n,m}(\lambda)$ and $Y = \lim_{m\to\infty} nX$. Then $Y \sim \chi^2_n(\lambda)$.*

*Proof.* By definition of the $F$-distribution, $X = \frac{N/n}{M/m}$, where $N \sim \mathcal{X}^2_n(\lambda)$ and $M \sim \mathcal{X}^2_m$ are independent random variables.

For mutually independent $\chi^2_1$ random variables $Y_1, \ldots, Y_m$,

$$M = \frac{Y_1 + \cdots + Y_m}{m}.$$

By the weak law of large numbers (Lemma F.1),

$$\frac{M}{m} \xrightarrow{P} \mathbb{E}[Y_1] = 1.$$

As a result, by Slutsky's Theorem (Theorem D.6),

$$\lim_{m\to\infty} nX = \lim_{m\to\infty} \frac{N}{M/m} = N \sim \mathcal{X}^2_n(\lambda).$$

$\square$

# G Duality Between Testing and Interval Estimation

In this section, we expand on the relationship between interval or region estimation (i.e., confidence interval estimation) and hypothesis testing. We present a general formulation, which can be specialized to linear regression. Previous work (e.g., [29]) examines the generation of confidence intervals for differentially private parametric inference. Our work focuses on hypothesis testing.

As before, for some unknown parameter $\theta \in \Omega$, $Z \sim P_\theta$ is the observed data. Also let $f : \Omega \to \mathbb{R}$ be a function on the parameter space (i.e., for linear regression, we can compute functions of the slope and/or intercept).

**Definition G.1** (Confidence Region). A (random) set $S(Z)$ is a $1 - \alpha$ confidence region for a (function of a) parameter $f(\theta)$ if
$$\mathbb{P}_\theta[f(\theta) \in S(Z)] \geq 1 - \alpha, \quad \forall \theta \in \Omega.$$

A confidence region is, essentially, a multi-dimensional generalization of a confidence interval.

**Definition G.2** (Acceptance Region). For every $f_0 \in \mathbb{R}$, $A(f_0)$ is the acceptance region for a nonrandomized level $\alpha$ test of
$$H_0 : f(\theta) = f_0 \ \text{ vs. } \ H_1 : f(\theta) \neq f_0.$$

$A(f_0)$ denotes the range of values that would lead to acceptance of the null hypothesis, when the null is true, where for the level $\alpha$ test,
$$\mathbb{P}_\theta[Z \in A(f(\theta))] \geq 1 - \alpha, \quad \forall \theta \in \Omega.$$

Define the following function:
$$S(z) = \{f_0 \ : \ \exists \theta_0 \text{ s.t. } f_0 = f(\theta_0) \text{ and } z \in A(f_0)\}.$$

Then $f(\theta) \in S(Z) \iff Z \in A(f(\theta))$ which implies that
$$\mathbb{P}_\theta(f(\theta) \in S(Z)) = \mathbb{P}_\theta(Z \in A(f(\theta))) \geq 1 - \alpha.$$

We have established that $S(Z)$ is, thus, a $1 - \alpha$ confidence region for $f$. Essentially, we have shown that **we can construct confidence regions from a family of nonrandomized tests**.

For any function $f : \Omega \to \mathbb{R}$ and $f_0 = f(\theta_0)$, we could seek to obtain a $1 - \alpha$ confidence region $S(Z)$ for the parameter $f_0$ (e.g., the mean or median).

Now, consider a test $\phi$ defined by
$$\phi(z) = \begin{cases} 1 & \text{if } f_0 \notin S(z) \\ 0 & \text{otherwise} \end{cases}.$$

Then if $f(\theta) = f_0$, then
$$\mathbb{E}_\theta \phi = \mathbb{P}_\theta[f_0 \notin S(Z)] \tag{25}$$
$$= \mathbb{P}_\theta[f(\theta) \notin S(Z)] \leq \alpha. \tag{26}$$

This test has level at most $\alpha$ for testing
$$H_0 : f(\theta) = f_0 \ \text{ vs. } \ H_1 : f(\theta) \neq f_0.$$

Then if the coverage probability for $S(Z)$ is exactly $1 - \alpha$, then
$$\mathbb{P}_\theta[f(\theta) \in S(Z)] = 1 - \alpha, \quad \forall \theta \in \Omega,$$
so that $\phi$ will have level of exactly $\alpha$.

To summarize, we have shown that we can: (1) Construct confidence regions from a family of nonrandomized tests. (2) Construct a family of nonrandomized tests from a $1 - \alpha$ confidence region.

Since (nontrivial) DP tests are randomized, to apply the duality framework, we could de-randomize by giving the DP procedure the random bits to be used for DP. Or we can also define the DP procedure as a family of nonrandomized tests.

We can still turn DP procedures for confidence region estimation into procedures for testing although, as we show below, the power of the corresponding tests is likely to be very low if the area of the confidence regions are too large.

**More Details on Experimental Evaluation of DP Confidence Intervals** We now proceed to construct a hypothesis test based on DP parametric bootstrap confidence intervals (e.g., using the work of [29]). Then we will experimentally compare to our linear relationship tester based on the DP $F$-statistic.

Suppose that $\theta$ is the set of parameters (e.g., standard deviation of the dependent and independent variables) and $f = f(\theta)$ is the estimation target (e.g., the slope in the dataset). The goal is to obtain a $1 - \alpha$ confidence interval $[\hat{a}_n, \hat{b}_n]$ for $f(\theta)$ via an end-to-end differentially private procedure. In other words, we want

$$\mathbb{P}\left[\hat{a}_n \leq f(\theta) \leq \hat{b}_n\right] = 1 - \alpha,$$

where the probability is taken over both $\theta$ and $f$.

Because of the randomized nature of (non-trivial) DP procedures, the finite-sample coverage of the interval might not exactly be close to $1 - \alpha$. Ferrando, Wang, and Sheldon [29] show the consistency of these intervals (in the large-sample, asymptotic regime).

Algorithm 7 follows the same framework as Algorithm 3, except that instead of simulating test statistics under the null hypothesis, the goal is to calculate a confidence interval for the slope. $P_{(\tilde{\theta}_0, \tilde{\theta}_1)}$ denotes the distribution from which we shall generate our bootstrap samples and from which a confidence interval can be estimated. For example, for taking bootstrap samples for the slope, $P_{(\tilde{\theta}_0, \tilde{\theta}_1)}$ would approximately be distributed as $\mathcal{N}(\tilde{\beta}_1, \frac{\widetilde{S^2}}{\widetilde{\text{nvar}}})$ where $\widetilde{\text{nvar}} = n \cdot \widetilde{x^2} - n \cdot \tilde{x}^2$ and $\widetilde{S^2}$ is as defined in Algorithm 4. Note that a crucial difference between tests based on the parametric bootstrap confidence intervals and our tests is the following: our tests only use $\tilde{\theta}_0$, a subset of the estimated DP statistics, to simulate the null distribution and decide to reject the null while the other approach uses $(\tilde{\theta}_0, \tilde{\theta}_1)$ to decide to reject the null.

The target slope is $b$. For example, if we seek to test for a linear relationship, we set $b = 0$ since under the null hypothesis, the slope will be 0. `DPStats` is a $\rho$-zCDP procedure for estimating DP sufficient statistics for a parametric model. In Algorithm 7, $(s_{(l)}, s_{(r)})$ is the parametric bootstrap confidence interval for the slopes under the null hypothesis.

---

**Algorithm 7** DP Test Framework via Parametric Bootstrap Confidence Intervals.

1: **Data**: $X \in \mathbb{R}^{n \times p}; Y \in \mathbb{R}^n$
2: **Input**: $n$ (dataset size); $\rho$ (privacy-loss parameter); $\alpha$ (target significance); $b$ (target slope)
3: $(\tilde{\theta}_0, \tilde{\theta}_1) = \text{DPStats}(X, Y, n, \rho)$
4: **if** $\tilde{\theta}_0 = \tilde{\theta}_1 = \perp$ **then**
5:     **return** Fail to Reject the null
6: **end if**
7: Select $K > 1/\alpha$
8: **for** $k = 1 \ldots K$ **do**
9:     Sample slope $s_k \sim P_{(\tilde{\theta}_0, \tilde{\theta}_1)}$
10: **end for**
11: Sort $s_{(1)} \leq \cdots \leq s_{(K)}$
12: Set $l = \lceil (K + 1)(\alpha/2) \rceil$
13: Set $r = \lceil (K + 1)(1 - \alpha/2) \rceil$
14: **if** $b \notin (s_{(l)}, s_{(r)})$ **then**
15:     **return** Reject the null
16: **else**
17:     **return** Fail to Reject the null
18: **end if**

---

Without privacy, by Lemma D.3, under the null hypothesis, we know that the slopes will be distributed as the following distribution: $\hat{\beta}_1 \sim \mathcal{N}\left(\beta, \frac{\sigma_\epsilon^2}{n \cdot \sigma_x^2}\right) \sim \mathcal{N}\left(0, \frac{\sigma_\epsilon^2}{n \cdot \sigma_x^2}\right)$. Even as $n$ increases and as we take fresh samples of $\hat{\beta}_1$, the parametric bootstrap confidence interval around $\hat{\beta}_1$ gets smaller and more concentrated around the true value 0. We expect to observe similar behavior when applying DP.

## H  Experimental Framework for Monte Carlo Evaluation

**Compute Resources Used**   We run all our experiments on a MacBook Pro (13-inch, 2018) with a 2.3GHz Quad-Core Intel Core i5 with 16GB Memory.[6] In Algorithm 8, we present a generic procedure showing how we obtain significance and power on our experimental evaluation of our DP tests. EstimateRejectionProb is a meta-procedure that uses DataSampler to sample a dataset $D$ either from the null or alternative distribution we are testing. Then it runs MonteCarloTester to decide whether to reject or fail to reject the null. MonteCarloTester can be any of the private Monte Carlo tests defined above or their non-private versions. The fraction of times (amongst $M$ trials) a reject decision is returned is estimated and can be used to calculate the significance or the power of the test.

DataSampler**:**   This procedure is used to sample from a user-specified distribution for testing the null or the alternative hypothesis. For example, for two groups $(X_1, Y_1)$ and $(X_2, Y_2)$ with slopes $\beta_1$ and $\beta_2$ respectively, the user could specify $\beta_1 = \beta_2$ as parameters to the data sampler. Other parameters include the size of the groups, the noise distribution in the dependent or independent variable, and so on.

MonteCarloTester**:**   Examples of this procedure are instantiations of Algorithm 3.

CompareAlgorithms**:**   For user-specified data samplers and Monte Carlo test procedures, this procedure (Algorithm 9) collates significance and power (to be plotted, for example).

---

**Algorithm 8** EstimateRejectionProb: Meta-Procedure for Estimating significance and power.

---
1: **Data**: DataSampler; MonteCarloTester; $M(\#trials)$
2: $r = 0$
3: **for** $m = 1, \ldots, M$ **do**
4:     $D \leftarrow$ DataSampler()
5:     **if** MonteCarloTester$(D)$ = Reject the null **then**
6:         $r = r + 1$
7:     **end if**
8: **end for**
9:
10: // Compute empirical probability that the test rejects
11:
12: **return** $r/M$

---

**Algorithm 9** CompareAlgorithms: Compares statistical performance of DP tests.

---
1: **Data**: DataSamplerList; MonteCarloTesterList; $M(\#trials)$
2: $R = []$
3: **for** DataSampler $\in$ DataSamplerList **do**
4:     **for** MonteCarloTester $\in$ MonteCarloTesterList **do**
5:         $e =$ EstimateRejectionProb(DataSampler, MonteCarloTester, $M$)
6:         Append (DataSampler, MonteCarloTester, $M, e$) to $R$
7:     **end for**
8: **end for**
9:
10: **return** $R$

---

[6] The code for reproducing our results can be found in the anonymous Github repository: `https://gitfront.io/r/user-4848858/LVCyokZuDhSj/dplr/`.