# OpenReview forum: "Hypothesis Testing for Differentially Private Linear Regression"
_NeurIPS.cc/2022/Conference — NeurIPS 2022 Accept_

### Official Review · Reviewer_f57B · 2022-07-04

**Rating:** 7
**Confidence:** 3
**Soundness:** 4 excellent
**Presentation:** 3 good
**Contribution:** 3 good

**Summary:**

This paper studies the problem of hypothesis testing for two problems in the generalised linear model: testing a linear relationship and testing for the presence of mixtures. Most of the techniques involves are based on privatising the $F$-statistic for the general linear model. These techniques are used through a Monte Carlo parametric bootstrap. Other techniques for testing presence of mixtures in this model are based on the DP non-parametric tests of a prior work by Couch et al. 2019.

The main results are that the DP $F$-statistic converges to the asymptotic distribution of its non-private counterpart, which implies the convergence of statistical power of DP $F$-statistic to that of the non-private $F$-statistic. The Monte Carlo parametric bootstrap is used to ensure an $\alpha$ target significance.

Asymptotic results for these problems are obtained in this paper. Also, experiments are performed to compare these results with either the non-private counterparts of these tests or other available techniques for this problem.

Edit: Updated score.

**Questions:**

I don't have a lot of questions, but just something for clarification.

1. The results in this paper are valid for the case when the ambient dimension is $2$, with the claim that they could be extended to arbitrary dimensions. Is that extension straightforward? Or would you require techniques that are significantly different mathematically from the ones being currently used in this paper?

2. Also, while generalising to higher dimensions, does the accuracy of the private tests still converge to that of the non-private ones?

3. In Line 12 of Algorithm 1, why do we need to use DPStats? I thought those points are artificial points, so do we need privacy here?

**Limitations:**

I think the authors have been honest about wanting to have non-asymptotic results for this problem. Even though I have mentioned that as a weakness above, I don't think it's a big deal.

I have mentioned my concerns about the writing above already. So, I don't have any other concerns. This results are quite self-contained and "optimal" already in the sense that the private convergence and the non-private convergence match.

Also, one more thing I would recommend is motivating the problem a bit more here. We know hypothesis testing and linear regression independently, but why is testing for this problem important? More like, why is privatising this is an important problem to solve? There are lot of problems that could need private solutions, but why is this an important problem to solve under DP constraints?

**Strengths And Weaknesses:**

Strengths:
1. Given that this problem has not been studied that extensively under DP constraints before (except in Sheffet 2017), I would say that this is a great starting point for this.
2. The work is also fairly technical and rigorous. The analyses of the significance levels theoretically is very thorough even if it is dealing with just asymptotic guarantees.
3. The convergence of statistical power of the DP $F$-statistic to that of its non-private counter part is a significant contribution. This means that the loss in accuracy due to privacy isn't much, which is usually the main target in most work on DP.
4. This work also addresses one of the main issues in the prior work on this: the prior work only starts to reject the null hypothesis when the sample size is very large. In other words, their tests could give accurate results even on "smaller" datasets.
5. One more thing I appreciate that the experiments have been performed both on synthetic and real-world datasets, as opposed to just on the former. It gives the impression that these tests are quite versatile in terms of the underlying data distributions.

Weaknesses:
1. I don't see this as a huge weakness (and is dependent on personal preferences), but getting non-asymptotic results would have made the paper a bit stronger. There are different schools of thoughts on this, of course. As I said though, this is not a big concern for me.
2. I have some issue with the writing of the paper. It looks like it was a beautifully written paper, but chopping it down to nine pages made it a little hard to read in the first shot. I'll be more specific about this. (A) The fluff text for Algorithm 1 didn't do much for me. I agree that a text version of the algorithms is supposed to help, but an intuition behind why certain decisions and steps are taken within them is probably a more helpful discussion. (B) Saying which algorithms have been restated in the appendices would have been nice. It gets confusing when algorithms are stated in the main body, but the discussions refer to the ones in the appendices.

---

> ### Author Response · Authors · 2022-07-31
> **Response #4 to Reviewer f57B**
>
> Thanks for the review. We will revise our paper, accordingly, to address your concerns. Below, we now respond to some of your comments:
>
> 1. *“I don't see this as a huge weakness (and is dependent on personal preferences), but getting non-asymptotic results would have made the paper a bit stronger. There are different schools of thoughts on this, of course. As I said though, this is not a big concern for me.”*:
>
> - Thanks for the suggestion and we agree that non-asymptotic results would be helpful and are very important. However, the very first step to making a statistical test usable theoretically is showing its asymptotic properties (since not all test statistics have nice asymptotic structure). For future work, we hope we (or some other group) can get non-asymptotic results for the DP F-statistic and all the other test statistics we have presented.
>
> 2. *“I have some issue with the writing of the paper. It looks like it was a beautifully written paper...I'll be more specific about this. (A) The fluff text for Algorithm 1 didn't do much for me. I agree that a text version of the algorithms is supposed to help, but an intuition behind why certain decisions and steps are taken within them is probably a more helpful discussion. (B) Saying which algorithms have been restated in the appendices would have been nice. It gets confusing when algorithms are stated in the main body, but the discussions refer to the ones in the appendices.”*
>
> - Thanks for the suggestion. We could be more explicit about how DPStats is used within the parametric bootstrap procedure. We will also reduce the amount of times we refer to results/algorithms in the appendix, to a need-to-know basis.
>
> 3. *“The results in this paper are valid for the case when the ambient dimension is 2, with the claim that they could be extended to arbitrary dimensions. Is that extension straightforward? Or would you require techniques that are significantly different mathematically from the ones being currently used in this paper?”*
>
> - For the F-statistic, we believe that the extension to the higher dimensional cases (for any design matrix with more than two columns) is not as involved. In particular, as long as one can compute sufficient statistics (using a procedure like DPStats) on X, Y, then you can compute the DP F-statistic using the perturbed sufficient statistics. On the other hand, the median testing reduction we provide might become a lot more involved, since the median is available in only one dimension (although higher dimensional analogues exist). We will add more text in the main body about how to (easily) generalize the F-statistic.
>
> 4. *“Also, while generalising to higher dimensions, does the accuracy of the private tests still converge to that of the non-private ones?”*:
>
> - Indeed, in theory, the accuracy of the privacy tests still converges to that of the non-private tests under certain conditions (e.g., the gram matrix still converges to a constant matrix). However, there might be cases in higher dimensions where this might not be the case and warrants further/future study.
>
> 5. *“In Line 12 of Algorithm 1, why do we need to use DPStats? I thought those points are artificial points, so do we need privacy here?”*:
>
> - Line 12 still uses DPStats to “simulate” the noise due to privacy when generating the null distribution. In principle, one could replace DPStats with Stats (without DP) but the distribution generated under the null will be noisy statistics (due to sampling) that do not account for noise due to privacy. DPStats allows us to simulate both sampling and privacy noise.
>
> 6. *“Also, one more thing I would recommend is motivating the problem a bit more here. We know hypothesis testing and linear regression independently, but why is testing for this problem important? More like, why is privatising this is an important problem to solve? There are lot of problems that could need private solutions, but why is this an important problem to solve under DP constraints?”*
>
> - We will include more examples of why privatizing linear regression is a very important problem. Thanks for the suggestion. The book by Stock and Watson [1] on the introductory tools for econometrics has several chapters that summarize the importance of linear regression to econometrics (and by extension, economics and the quantitative social sciences). An an example given in the first chapter dedicated to linear regression, suppose that a school cuts down the size of its elementary school classes: what is the effect of the decision on the test scores of the students? This question can be modeled by making X signify class size and Y signify test scores. Testing for the relationship between X and Y constitutes a linear regression testing problem. Ordinary least squares, and more generally, the general linear model are staple in economics.
>
> [1] James Stock and Mark Watson. Introduction to Econometrics (3rd edition). Addison Wesley Longman, 2011.

---

### Official Review · Reviewer_khGd · 2022-07-11

**Rating:** 7
**Confidence:** 4
**Soundness:** 3 good
**Presentation:** 2 fair
**Contribution:** 3 good

**Summary:**

Provides a GLRT & Monte-carlo based styled DP test for two kinds of hypothesis tests in the setting of linear regression: a) Testing a Linear Relationship, b) Testing for Mixtures. They take a holistic approach covering various aspects such as asymptotic  convergence in distribution results for the test-statistic. They provide empirical power analysis and significance analysis results. They compare against non-parametric tests such as the Kruskal-Wallis test.

Typically, a classical way has been to understand the asymptotic distribution of a test-statistic conditioned on null hypothesis being true, followed by verifying its departure from that case based on observed test-statistic to have a guarantee on the performance of the test both in terms of its significance, power and worst-case characterizations. Note: power of a test depends on effect-size, sample size, significance level and so forth

**Questions:**

1.) line 269: is the notation a typo or intended in terms of a constant in using O(1) in log 2n^O(1)?

2.) To this Q, the authors mention the conclusion section covers limitations-although it does not.: (Did I miss it?)
     Did you describe the limitations of your work? [Yes] See conclusion and experiments
     425 section (our DP tests do not perform well for certain data distributions).

**Limitations:**

-some details in expt section on how the empirical power was computed would be useful.
-the word appendix is referred to only once in the main paper. it instead refers to section A or C etc.
Please say 'Section A in appendix' and so forth.
-The section on Overview of Techniques seems a bit too packed in a short space. It should ideally better help the navigation of the paper.

-reference to paper on test statistic introduced in eqn 1 is needed or it needs to be clarified that it was introduced in this submission. it is unclear-which case it is in the beginning of the paper to the reader.

-minor: The word general linear model (GLM), has been used multiple times in this submission to refer to linear regression with Gaussian assumptions. GLM is typically used for generalized linear models of which linear regression is just a special case. please change the terminology to avoid this overloading.
-A table of main notations would be quite helpful.

**Strengths And Weaknesses:**

Strengths:  The submission does quite a thorough job in covering various aspects of DP hyp-testing for linear regression. They provide DP-tests, study convergence of test-statistic, draw similarities to GLRT, study the power, provide parametric bootstrap based computational test and so forth.

Weakness: It is bit unfair to expect this, given the page limit for the main paper: That said, the organization of the paper's approach & results in the appendix could have been better covered in the main paper. This is purely a comment in terms of providing clarity in the reading.

---

> ### Author Response · Authors · 2022-07-31
> **Response #3 to Reviewer khGd**
>
> Thanks for the review. We will revise our paper, accordingly, to address your concerns. Below, we now respond to some of your comments:
>
> 1. *“line 269: is the notation a typo or intended in terms of a constant in using O(1) in log 2n^O(1)?”*
>
> - Indeed, we use O(1) to denote a constant but could be more explicit with the use of this notation. i.e., $\exists k$, a universal constant, in the expression $\log 2n^k$.
>
> 2. *“To this Q, the authors mention the conclusion section covers limitations-although it does not.: (Did I miss it?) Did you describe the limitations of your work? [Yes] See conclusion and experiments 425 section (our DP tests do not perform well for certain data distributions).”*
>
> - We state, in the conclusion, that our theoretical results are in the asymptotic regime (large-sample theory for theoretical statistics) but it will be desirable (for future work) to also have theory for the finite-sample regime. In the contributions section, we also state when the power of our tests are low (e.g., when the variance of the dependent variable is very small). We can explicitly incorporate more discussion of limitations to the conclusion section.
>
> 3. *“some details in expt section on how the empirical power was computed would be useful. “*
>
> -  This is already in the appendix (Section H) but we can move some of this text to the main body, as well.
>
> 4. *“the word appendix is referred to only once in the main paper. it instead refers to section A or C etc. Please say 'Section A in appendix' and so forth. “*
>
> - Indeed, we can make this suggested change.
>
> 5. *“reference to paper on test statistic introduced in eqn 1 is needed or it needs to be clarified that it was introduced in this submission. it is unclear-which case it is in the beginning of the paper to the reader.”*
>
> - This is the (non-private) F-statistic. We can add a more explicit reference for this statistic.
>
> 6. *“-minor: The word general linear model (GLM), has been used multiple times in this submission to refer to linear regression with Gaussian assumptions. GLM is typically used for generalized linear models of which linear regression is just a special case. please change the terminology to avoid this overloading. -A table of main notations would be quite helpful.”*
>
> - Our results apply to the general linear model (GLM1) and might apply to the generalized linear model (GLM2). We do not state any results in terms of GLM2, which is more general (loosens assumptions on the residuals, for example) than GLM1. We will also add, into the main body, more text about our use of notation.

---

> > ### Comment · Reviewer_khGd · 2022-08-07
> > **Satisfactory**
> >
> > These comments & modifications are satisfactory. Hence, wouldn't change my evaluation.

---

### Official Review · Reviewer_wjEQ · 2022-07-12

**Rating:** 6
**Confidence:** 3
**Soundness:** 4 excellent
**Presentation:** 3 good
**Contribution:** 3 good

**Summary:**

The paper provides a framework for differentially private hypothesis tests for testing a linear relationship (when also uncertainty estimates are reported) and testing for the presence of mixtures. The proposed tests for detecting presence of mixtures are the first DP tests tailored for this particular task. There is both strong experimental and theoretical evidence (DP F-statistics converges to non-DP one).

**Questions:**



The introduction says: "For the mixture model tester, we additionally adapt and evaluate a nonparametric method, a DP analogue of the Kruskal-Wallis test due to Couch, Kazan, Shi, Bray, and Groce [21]..". It remains very unclear without looking to the appendix what is meant by this.

Being non-expert in this topic, it sounds to me that there are absolutely no better fit among DP hypothesis testing methods you could reasonably compare against, am I correct?

One of first things coming to my mind: How can I trust that your adaptation of this median testing method is the best possible method to compare against, how can I trust this particular adaptation is reasonable? Why not have an adaption of some other DP hypothesis testing algorithm? In any case I think it would be good to have some more explicit motivation for these choices.

Also, you mention that you compare to the DP parametric bootstrap method for confidence intervals but the corresponding experiments are only in the appendix (without reference).

Few remarks:

Perhaps a typo in abstract: "which are uniformly most powerful unbiased in the non-private setting" ?

Figures 3a and 3b: some dashed lines missing? (non-private baselines)

The notation for clipping used in Algorithm 2 is not explained anywhere.

Notice: The abbreviation KW that you use in Experiments - section is not explained anywhere in the main text.

**Limitations:**

Yes (the methods are widely applicable: hypothesis testing for linear regression)

**Strengths And Weaknesses:**

Strengths:

- Strong theory, showing that the DP F-statistics provided by these test converge in distribution to those of the non-DP ones.
- Strong experimental evidence
- An extensive literature review on this line of research (DP hypothesis testing)

Weaknesses:

- I think the paper would need some polishing (see details below)
- Difficult to digest for a non-expert in this topic like me. Considering that problem is of quite general nature (DP hypothesis testing, linear models), I would imagine the presentation could be made more easily accessible to a wider audience. I am afraid that the results would be appreciated by only expert in this line of research.
- On a related note, I think it is good that there is a short introduction to hypothesis testing in the appendix, but I think it needs also polishing: for example F-statistics is not explained anywhere.

---

> ### Author Response · Authors · 2022-07-31
> **Response #2 to Reviewer wjEQ**
>
> Thanks for the review. We will revise our paper, accordingly, to address your concerns. Below, we now respond to some of your comments:
>
> 1. *“I think the paper would need some polishing (see details below)”*:
>
> - we can polish the paper further to incorporate your comments.
>
> 2. “Difficult to digest for a non-expert in this topic like me. Considering that problem is of quite general nature (DP hypothesis testing, linear models), I would imagine the presentation could be made more easily accessible to a wider audience. I am afraid that the results would be appreciated by only expert in this line of research.”:
>
> - In the introduction, we can further motivate the problem of linear regression, especially for non-experts.
>
> 3. “On a related note, I think it is good that there is a short introduction to hypothesis testing in the appendix, but I think it needs also polishing: for example F-statistics is not explained anywhere.”:
>
> - In line 218, we provide the F-statistic in a shortened form; but we agree that we could be more explicit in naming this equation as the F-statistic. We will update the manuscript accordingly.
>
> 4. *“The introduction says: "For the mixture model tester, we additionally adapt and evaluate a nonparametric method, a DP analogue of the Kruskal-Wallis test due to Couch, Kazan, Shi, Bray, and Groce [21]..". It remains very unclear without looking to the appendix what is meant by this.”*:
>
> - In Section C.3.2, we fully describe how we test for mixtures via the use of the DP Kruskal-Wallis test. Because of space limitations, we couldn’t include our full set of experimental evaluation. We can add some text in the main body to explicitly refer to Section C.3.2.
>
> 5. *“Being non-expert in this topic, it sounds to me that there are absolutely no better fit among DP hypothesis testing methods you could reasonably compare against, am I correct?”*:
>
> - There might be better fits (that we are yet to find and this question definitely warrants further research). But we have tried an exhaustive-enough list of reasonable tests: both parametric and non-parametric tests. The use of non-parametric tests (e.g., Couch et al. work) is motivated by work by the theoretical work of Dwork and Lei, which shows that DP nonparametric methods might outperform parametric methods even when the data is nicely behaved (e.g., normally distributed). This is the reason why for every testing problem we tackle, we provide both a parametric and non-parametric version. The F-statistic serves as the general parametric solution that seems to, mostly, outperform the non-parametric counterparts. We identify the regimes under which this observation occurs.
>
> 6. *“Also, you mention that you compare to the DP parametric bootstrap method for confidence intervals but the corresponding experiments are only in the appendix (without reference).”*
>
> - Indeed, because of space reasons, the results are in the appendix. We will add an explicit reference to where these appear in the appendix.
>
> 7. *“The notation for clipping used in Algorithm 2 is not explained anywhere.”*:
>
> - We can include the clipping notation in the main body also (in addition to the appendix, as is done already). Thanks for the suggestion.
>
> 8. *“Notice: The abbreviation KW that you use in Experiments - section is not explained anywhere in the main text.”*
>
> - We can add a description of the KW abbreviation which means Kruskal-Wallis. Thanks for the suggestion.

---

> > ### Comment · Reviewer_wjEQ · 2022-08-09
> > **Response**
> >
> > Thank you for the thorough response. I agree with reviewer XTXZ that the paper would benefit from a more detailed intro to F-statistics and a motivation for using F-statistics. I am keeping my score i.e. also supporting acceptance.

---

### Official Review · Reviewer_XTXZ · 2022-07-17

**Rating:** 6
**Confidence:** 4
**Soundness:** 4 excellent
**Presentation:** 4 excellent
**Contribution:** 3 good

**Summary:**

This paper designs differentially private hypothesis tests in the generalized linear model (GLM) for the problems of testing a linear relationship (where the null hypothesis is that the slope of the linear model is (say) beta=0) and testing for the presence of mixtures.
The paper studies a DP version of the F-statistic and shows that converges asymptotically to the distribution of its non-private counterpart, as do the regression coefficients. Using this statistic, in combination with the Monte Carlo bootstrapping framework as used in Gaboardi et al (2016), the authors construct DP hypothesis tests both for testing a linear relationship and for the presence of mixtures.
Three datasets are used to evaluate these testers experimentally. The first tester is being compared to its non-private counterpart and to a test based on DP parametric bootstrap method for confidence interval  and seems to perform better. For testing for the presence of mixtures, the authors provide additionally e DP nonparametric method based on Couch et al. which performs better than the one based on the F-statistic for smaller datasets.

**Questions:**

One thing that I missed is some discussion on the choice of the F-statistic for these problems. I think some references on how widely-used this statistic is or why it is a better choice than other approaches in the literature might be useful to a statistics non-expert.

Small typo:
In line 8 of algorithm 2, the assignment to theta_0,theta_1 should presumably not be there.


**Limitations:**

Both adequately discussed.

**Strengths And Weaknesses:**

Strengths:
- The paper clearly states and supports its claims and positions itself with respect to prior work.
-  The convergence properties of the proposed DP version of the F-statistic are fully analysed. The authors show that the linear regression coefficients converge to their non-private counterparts with optimal rate and the power of the differentially private F-statistic converges to the statistical power of the non-private F-statistic.
- The paper includes experimental evaluations that compare the performance of the tests to the non-private tests but also to other approaches (one different approach for each problem), which creates a more complete picture.

Weaknesses:
- I would say that there are not clear techniques that emerge from this work but analyzing the convergence of the statistic is not trivial and in general this is a minor point. It is a solid paper overall.

---

> ### Author Response · Authors · 2022-07-31
> **Response #1 to Reviewer XTXZ**
>
> Thanks for the review. We will revise our paper, accordingly, to address your concerns. Below, we now respond to some of your comments:
>
> 1. *“I would say that there are not clear techniques that emerge from this work but analyzing the convergence of the statistic is not trivial”*:
>
> -  **Wide Applicability of DP F-statistic**: The DP F-statistic (not previously applied to linear regression in the DP literature) offers a general recipe for performing tests for the general linear model (e.g., using the same framework for testing a linear relationship and testing for mixtures). Such unification provides practitioners (especially DP non-experts) with easy-to-use tools with strong theoretical properties — as we show through our asymptotic convergence theorems and lemmas — for linear regression.
>
> - **Analytical Tools**: We agree that the analytic tools we provide for the main theoretical result are non-trivial. And, moreover, the analysis could apply to other test statistics beyond the F-statistic (such as DP variations of the U-statistic).
>
> - **Novel Reductions**: In addition, we provide novel reductions for DP hypothesis testing. In particular, we cannot the Couch et al. work directly but rather give a novel reduction from our linear-regression mixture model testing problem to their median-equality testing problem.
>
> 2. *“One thing that I missed is some discussion on the choice of the F-statistic for these problems. I think some references on how widely-used this statistic is or why it is a better choice than other approaches in the literature might be useful to a statistics non-expert.”*
>
> - **UMP**: A level-$\alpha$ test is uniformly most powerful (UMP) if its power is at least that of any other level-$\alpha$ test. As explained in the abstract (and other parts of the paper), the non-private F-statistic is uniformly most powerful unbiased, for the general linear model, which justifies the choice of this test statistic. We could add additional text in the main body about the statistical power of the F-statistic.
>
> - **Generality for Testing**: For example, as noted in [1], unlike the t-statistic, the F-statistic can be used to test joint hypothesis about regression coefficients (e.g., are the slope and intercept both 0?). In our work, we apply the F-statistic to testing for mixtures and for a linear relationship. But one could use the F-statistic more broadly.
>
> - **Use in Social Sciences and Econometrics:** As [1] states: [linear regression] "... is the dominant method used in practice" and "is the common language... throughout economics, finance, and the social sciences more generally." "Using [linear regression] means that you are speaking the same language as other economists and statisticians." We will add additional text to illustrate the wide applicability of linear regression in the quantitative social sciences and econometrics.
>
> 3. *"Small typo: In line 8 of algorithm 2, the assignment to theta_0,theta_1 should presumably not be there."*
>
> - Thanks for the careful reading. You are right — we will remove this line.
>
>
> [1] James Stock and Mark Watson. Introduction to Econometrics (3rd edition). Addison Wesley Longman, 2011.

---

> > ### Comment · Reviewer_XTXZ · 2022-08-08
> > **Satisfactory**
> >
> > Thanks for your response. As I and other reviewers mentioned, motivating the use of the F-statistic and the problem in general in the main body would be a significant improvement to the manuscript and the changes you suggested sound good. Also, I originally did not think of the reduction itself as a part of the novelty of the paper, but I agree that it is a nice contribution! So overall, clarifying and drawing attention to the points made in this and the rest of the responses (re: motivation for the problem and the techniques) would be a good revision. I still support acceptance of this paper.

---

### Meta-Review · Area_Chair_NHGh · 2022-08-24

**Recommendation:** Accept
**Confidence:** Certain

**Metareview:**

The reviewers were all quite positive on the results, and agreed the paper should be accepted. The only point that came up in the discussion is a slight miscommunication between the reviewers and authors: the reviewers asked for better motivation for some aspects (e.g., the F-statistic), and it seemed like the authors said they would motivate linear regression better. All the reviewers understand linear regression is very important, so the authors are recommended to instead revise based on their specific comments.

**Award:**

No

---

### Decision · Program_Chairs · 2022-09-14

Accept